# XGenBoost: Synthesizing Small and Large Tabular Datasets with XGBoost

## Abstract

Tree ensembles such as XGBoost are often preferred for discriminative tasks in mixed-type tabular data, due to their inductive biases, minimal hyperparameter tuning, and training efficiency. We argue that these qualities, when leveraged correctly, can make for better generative models as well. As such, we present XGenBoost, a pair of generative models based on XGBoost: i) a Denoising Diffusion Implicit Model (DDIM) with XGBoost as score-estimator suited for smaller datasets, and ii) a hierarchical autoregressive model whose conditionals are learned via XGBoost classifiers, suited for large-scale tabular synthesis. The architectures follow from the natural constraints imposed by tree-based learners, e.g., in the diffusion model, combining Gaussian and multinomial diffusion to leverage native categorical splits and avoid one-hot encoding while accurately modelling mixed data types. In the autoregressive model, we use a fixed-order factorization, a hierarchical classifier to impose ordinal inductive biases when modelling numerical features, and de-quantization based on empirical quantile functions to model the non-continuous nature of most real-world tabular datasets. Through two benchmarks, one containing smaller and the other larger datasets, we show that our proposed architectures outperform previous neural- and tree-based generative models for mixed-type tabular synthesis at lower training cost.

## 1. Introduction

Data synthesis for mixed-type tabular data has become a popular research topic for purposes such as data augmentation, sharing sensitive research data, and federated learning (Jordon et al., 2022; Das et al., 2022; van Kesteren, 2024;

Goetz & Tewari, 2020). Most current state-of-the-art methods rely on deep neural networks, which require modern computing resources (GPUs) to ensure reasonable training times. Access to these resources is, unfortunately, distributed unfairly across the world (Lehdonvirta et al., 2024). At the same time, tree ensembles are often considered to be more suitable function approximators for mixed-type tabular data: having more suitable inductive biases, being more efficient to train, and requiring less extensive hyperparameter tuning (Grinsztajn et al., 2022). This begs the question whether tree ensembles should replace neural networks as function approximators in generative architectures for tabular data.

**Design Constraints** Effectively leveraging tree learners in generative architectures requires respecting the constraints they naturally impose. Firstly, most implementations are single-output only[1], such that we train **feature-specific models**. Secondly, they do not use mini-batch training. Learning invariance against different permutations of the same samples therefore requires **extending the training data beyond its original size**. This inherently limits scalability to large datasets in architectures which require learning such invariances, e.g., diffusion models and random-order autoregressive models. Lastly, they are one of the few methods which can natively handle categorical data by learning to split directly on categories. However, this is only useful when we can **omit one-hot encoding** in the generative architecture.

**Synthesizing Small Datasets** For small tabular datasets, extending the training data beyond its original size can be permissible, such that we can leverage powerful methods such as diffusion models. We employ a Denoising Diffusion Probabilistic Model (DDPM) (Ho et al., 2020), combining Gaussian and multinomial diffusion to accurately model mixed-type tabular data. Using separate numerical and categorical diffusion processes allows us to apply a single XGBoost score-estimator per feature (regressor or classifier), instead of a purely numerical diffusion process in one-hot encoded space (Jolicoeur-Martineau et al., 2024). Another advantage, when compared to similar neural-based diffusion models (Kotelnikov et al., 2023), is that per-feature score-

---

[1]Anonymous Institution, Anonymous City, Anonymous Region, Anonymous Country. Correspondence to: Anonymous Author <anon.email@domain.com>.

Preliminary work. Under review by the International Conference on Machine Learning (ICML). Do not distribute.

[1]The XGBoost library has a vector-leaf implementation, but this is still in progress, and most features are missing.

estimators omit the need for loss-scale weighting of the two separate diffusion processes. Then, we further extend the DDPM to a Denoising Diffusion Implicit Model (DDIM) (Song et al., 2021a) to allow high quality samples at fewer diffusion steps. This is useful as we also train timestep-specific score-estimators, and therefore prefer fewer total diffusion steps. We name this method XGenB-DF. We show that XGenB-DF generates high-fidelity synthetic data in a previous benchmark of 27 small tabular datasets (Jolicoeur-Martineau et al., 2024).

**Synthesizing Large Datasets** To allow scaling to large tabular datasets, we cannot use a method which requires massively extending the training set. Note that here, we define scaling in an **absolute** rather than relative sense. Deep generative models, for example, scale *relatively* well, but still require significant resources and training time for large datasets. We employ a fixed-order autoregressive model where conditionals are learned by XGBoost models. We use hierarchical XGBoost classifiers as probabilistic predictors for numerical features (McCarter, 2024). Categorical features are directly modelled by multi-class XGBoost classifiers. The architecture is tailor-made for large-scale tabular synthesis in several ways, i) a fixed-order factorization to omit the need for extending the training set, ii) resampling quantized features based on interpolated empirical quantile functions to model skewed or non-continuous distributions, and iii) constraining categorical cardinality to limit training time and privacy risk. We name this method XGenB-AR. We show that XGenB-AR generates high-fidelity synthetic data at much lower training cost in a previous benchmark for tabular diffusion models (Mueller et al., 2025). As an example of the scalability, XGenB-AR trains in $\approx 3$ minutes on the acsincome[2] dataset using 16 CPU cores, 64GB RAM.

XGenBoost is available through a simple scikit-learn style API at `ANONYMIZED`.

## 2. Related Work

The first methods for tabular synthesis mainly leveraged iterative imputation (Rubin, 1993). Other statistical methods such as copulas (Patki et al., 2016) and Bayesian networks (Zhang et al., 2017) were also introduced. As compute became cheaper and more powerful, deep generative models became more apt to tabular synthesis. GANs (Goodfellow et al., 2014) and VAEs (Kingma & Welling, 2013) showed great promise initially (Xu et al., 2019; Zhao et al., 2021). When diffusion models were adapted for tabular data, they quickly became the leading approach for synthesizing high-fidelity tabular data (Lee et al., 2023; Kim et al., 2023;

Kotelnikov et al., 2023; Zhang et al., 2024). Many of the current research efforts therefore focus on further improving neural tabular diffusion models (Mueller et al., 2025; Si et al., 2025; Shi et al., 2025).

Various tree ensemble-based generative architectures have also been developed. Adversarial Random Forests (ARF) leverage Random Forests in alternating rounds of generation and discrimination to estimate density and generate synthetic data (Watson et al., 2023). Unmasking Trees (UT) is a random-order hierarchical autoregressive model using XGBoost classifiers to model conditional distributions (McCarter, 2024). ForestDiffusion (FD) and its flow-based counterpart ForestFlow (FF) leverage XGBoost as score-estimator in a score-matching diffusion and conditional flow-matching model, respectively (Jolicoeur-Martineau et al., 2024). Unfortunately, none of these methods scale well to large tabular datasets. ARF is hampered by i) sequential rounds of adversarial learning and ii) reliance on Random Forest classifiers (Watson et al., 2023). UT extends the training dataset $K$ times ($K \approx 50$) to allow random-order synthesis, which massively increases computational demands for large datasets (McCarter, 2024). FD and FF similarly extend the training set $K$ times ($K \approx 50 - 100$) to accurately estimate the expectation in diffusion losses (Jolicoeur-Martineau et al., 2024).

FD adopts Gaussian diffusion (or Gaussian probability paths, in FF) in a numerically encoded feature space, i.e., categorical variables are one-hot encoded. This results in non-smooth densities that deviate from Gaussian diffusion assumptions. Categorical features can be more appropriately modelled by multinomial diffusion processes (Hoogeboom et al., 2021). Additionally, one-hot encoding is not well-aligned with the choice of XGBoost as score-estimator (see Section 1). As XGBoost is typically single-output, the number of required XGBoost regressors scales linearly with the total categorical cardinality. Also, training occurs in a substantially higher-dimensional space, despite XGBoost being able to natively model categorical splits.

UT is a random-order autoregressive model, which models numerical features through hierarchical classifiers consisting of meta-trees of binary XGBoost classifiers, and categorical features through multi-class XGBoost classifiers (McCarter, 2024). To enable random-order synthesis, they extend the training set $K \approx 50$ times, which is the main culprit for high computational demand in large datasets. Note that also, for a dataset of $D$ features there are $D2^{D-1}$ conditionals to be learned in random-order synthesis, and as typically $K << D2^{D-1}$, this is a lossy approximation. Furthermore, quantized numerical bins are resampled uniformly, which does not suit most skewed or non-continuous real-world datasets.

Tree ensembles such as Random Forests (Breiman, 2001),

---

[2]`https://fairlearn.org/main/user_guide/datasets/acs_income.html`

LightGBM (Ke et al., 2017), CatBoost (Prokhorenkova et al., 2018), and XGBoost (Chen, 2016) are among the strongest methods for discriminative tasks on tabular data, in terms of accuracy, robustness, and efficiency. Lately, neural tabular foundation models have been receiving more attention, often performing very well without any fine-tuning through in-context learning (Hollmann et al., 2023). Currently, however, these are often constrained to limited rows and columns. We mainly consider XGBoost, as it provides a good balance of accuracy and training efficiency, is apt at handling missing data, and has shown to be effective in similar previous works (Jolicoeur-Martineau et al., 2024; McCarter, 2024). Essentially, we use XGBoost as a representative instance of a broader set of methods (gradient-boosted decision trees, but perhaps in the future also tabular foundation models) which are particularly more suited than neural networks for mixed-type tabular data, and can be effectively used in our generative architectures. Other similar methods (LightGBM, CatBoost) may be equally suitable.

## 3. Methods

Let $\mathcal{D} = \{x^{(i)}\}_{i=1}^n$, $x^{(i)} \in \mathcal{X}$ be a mixed-type tabular dataset of i.i.d. samples from an unknown joint distribution $p(\cdot)$ over the product space $\mathcal{X} = \prod_{j=1}^d \mathcal{X}_j$. We can distinguish $\{1, \ldots, d\} = \mathcal{M} \dot{\cup} \mathcal{C}$ where $\mathcal{M}$ and $\mathcal{C}$ denote the sets of numerical and categorical features, respectively. For each feature $j \in \{1, \ldots, d\}$, the feature domain is

$$\mathcal{X}_j = \begin{cases} \mathbb{R}, & j \in \mathcal{M}, \\ \{1, \ldots, K_j\}, & j \in \mathcal{C}, \end{cases} \quad K_j \in \mathbb{N}. \quad (1)$$

The goal of tabular data synthesis is to learn a parametric distribution $p_\theta$, although non-parametric alternatives exist (Neto, 2025), such that $p_\theta(\cdot) \approx p(\cdot)$.

### 3.1. Synthesizing Small Datasets: Diffusion Models with XGBoost as Score-Estimator

We propose a diffusion model with XGBoost as score-estimator for small-scale tabular synthesis. Diffusion models learn the likelihood of the data by learning to reverse a forward Markov process $q(\mathbf{x}_{1:T}|\mathbf{x}_0) = \prod_{t=1}^T q(\mathbf{x}_t|\mathbf{x}_{t-1})$ (Ho et al., 2020). Here, $q(\mathbf{x}_t|\mathbf{x}_{t-1})$ is part of a fixed Markov chain that adds noise to the data. The reverse process defines a joint distribution over $\mathcal{X}$:

$$p_\theta(\mathbf{x}_{0:T}) = p(\mathbf{x}_T) \prod_{t=1}^T p_\theta(\mathbf{x}_{t-1}|\mathbf{x}_t). \quad (2)$$

Score-matching models with variance-preserving schedules, such as FD (Jolicoeur-Martineau et al., 2024), are very similar to Denoising Diffusion Probabilistic Models (DDPMs)

(Ho et al., 2020), up to the choice of sampling discretization, e.g., Euler-Maruyama versus ancestral sampling (Song et al., 2021b). We choose a DDPM specification as this can be reconciled with previous discrete diffusion paradigms (Hoogeboom et al., 2021). Score-matching explicitly tries to model the gradient of the density function, which is undefined for discrete data - although some solutions have been proposed (Meng et al., 2022).

Most often, $\theta$ is learned by a neural network. However, it can also be learned by XGBoost models (Jolicoeur-Martineau et al., 2024). Specifically, each feature is de-noised by a separate XGBoost model. We also train separate models for each timestep and (categorical) target label, to condition on timesteps and the target feature, respectively. Finally, to adequately estimate the expectation in the diffusion loss, we need to marginalize over various noise levels per sample. As XGBoost does not use mini-batch training, we need to duplicate the dataset $K$ times to ensure we compute the loss w.r.t. each sample for $K$ different noise levels[3].

**Gaussian Diffusion**  Assuming a Gaussian latent variable such that $p(\mathbf{x}_T) = \mathcal{N}(\mathbf{x}_T; \mathbf{0}, \mathbf{I})$, the forward noising process is defined by

$$q(\mathbf{x}_t|\mathbf{x}_{t-1}) = \mathcal{N}(\mathbf{x}_t; \sqrt{1 - \beta_t}\mathbf{x}_{t-1}, \beta_t\mathbf{I}), \quad (3)$$

where $\beta_t$ define the stepwise variance, which in our case follows a variance-preserving schedule. The reverse process is modelled by Gaussian transitions such that

$$p_\theta(\mathbf{x}_{t-1} \mid \mathbf{x}_t) = \mathcal{N}(\mathbf{x}_{t-1}; \mu_\theta(\mathbf{x}_t, t), \tilde{\beta}_t\mathbf{I}),$$

$$\mu_\theta(\mathbf{x}_t, t) = \frac{\sqrt{\bar{\alpha}_{t-1}}\beta_t}{1 - \bar{\alpha}_t}\mathbf{x}_\theta(\mathbf{x}_t, t) + \frac{\sqrt{\bar{\alpha}_t}(1 - \bar{\alpha}_{t-1})}{1 - \bar{\alpha}_t}\mathbf{x}_t, \quad (4)$$

where $\alpha_t = 1 - \beta_t$, $\bar{\alpha}_t = \prod_{s=1}^t \alpha_s$, $\tilde{\beta}_t = \frac{1 - \bar{\alpha}_{t-1}}{1 - \bar{\alpha}_t}\beta_t$, and $\mathbf{x}_\theta(\mathbf{x}_t, t)$ is the prediction of the original data $\mathbf{x}_0$ at timestep $t$ by a regression model parameterized by $\theta$. Equivalently, we can reparameterize the model to predict the noise instead (Ho et al., 2020)

$$\epsilon_\theta(\mathbf{x}_t, t) = \frac{1}{\sqrt{1 - \bar{\alpha}_t}}(\mathbf{x}_t - \sqrt{\bar{\alpha}_t}\mathbf{x}_\theta(\mathbf{x}_t, t)), \quad (5)$$

or a velocity-based objective, which is a linear combination of the data and the noise (Salimans & Ho, 2022)

$$v_\theta(\mathbf{x}_t, t) = \sqrt{\bar{\alpha}_t}\epsilon_\theta(\mathbf{x}_t, t) - \sqrt{1 - \bar{\alpha}_t}\mathbf{x}_\theta(\mathbf{x}_t, t). \quad (6)$$

---

[3]Similar to a newer version of FD, we implemented XGBoost's data iterator, iterating the dataset $K$ times instead of duplicating. However, this still trains many models on huge datasets, and only solves the memory demand, not the scaling issue.

We use the velocity-based reparametrization as it tends to yield smoother loss trajectories (Gao et al., 2024).

Next, we note that we have to use a relatively low number of timesteps $T$, as we train a separate XGBoost model for each timestep. DDIMs are a generalization of DDPMs to non-Markovian diffusion processes, which yield *deterministic* generative processes capable of producing high quality samples much faster (Song et al., 2021a). DDIMs have the same training objective as DDPMs, and are thus an easily implementable solution to our low $T$ restriction. The (now deterministic) reverse process can be defined as:

$$\mathbf{x}_{t-1} = \sqrt{\bar{\alpha}_{t-1}}\mathbf{x}_\theta(\mathbf{x}_t, t) + \sqrt{1 - \bar{\alpha}_{t-1}}\epsilon_\theta(\mathbf{x}_t, t). \quad (7)$$

Whereas Jolicoeur-Martineau et al. (2024) propose a flow-matching model FF which outperforms FD, we stay within the diffusion paradigm. Flow-matching and diffusion are equivalent in many ways, especially when considering similar prediction objectives and samplers (Gao et al., 2024). In this respect, flow-matching and diffusion are especially similar when considering a velocity-prediction objective (Salimans & Ho, 2022) and a deterministic sampler, such as found in DDIMs (Song et al., 2021a). Following the success of FF, we find this to be effective for our method as well.

**Multinomial Diffusion** Multinomial diffusion can be more suitable for modelling categorical data. In this case we assume a uniform categorical latent variable $p(x_T) = \text{Cat}(x_T; \frac{1}{K}\mathbf{1})$, such that the forward noising process adds uniform noise over the $K$ categories (Hoogeboom et al., 2021):

$$q(x_t|x_{t-1}) = \text{Cat}(x_t; (1 - \beta_t)x_{t-1} + \beta_t/K). \quad (8)$$

Correspondingly, the reverse process is modelled by categorical transitions such that

$$p_\theta(x_{t-1}|x_t) = \text{Cat}\left(x_{t-1}; \frac{\pi_\theta(\mathbf{x}_t, t)}{\sum_{k=1}^K \pi_{k,\theta}(\mathbf{x}_t, t)}\right)$$
$$\pi_\theta(\mathbf{x}_t, t) = \left[\alpha_t\mathbf{x}_t + \frac{(1 - \alpha_t)}{K}\right] \odot \quad (9)$$
$$\left[\bar{\alpha}_{t-1}\mathbf{x}_\theta(\mathbf{x}_t, t) + \frac{(1 - \bar{\alpha}_{t-1})}{K}\right]$$

where $\mathbf{x}_\theta(\mathbf{x}_t, t)$ is the prediction of the original data $\mathbf{x}_0$ at timestep $t$ by a classifier parameterized by $\theta$.

Hoogeboom et al. (2021)'s original multinomial diffusion specification performs a single step to add uniform noise over binary-encoded categorical features, and sample from the noised distribution (Equation (8)). This is equivalent to a two-step process where we first decide for which samples to retain $x_t$, and for which to resample uniformly from $K$, allowing us to omit one-hot encoding.

**Combining Gaussian and Multinomial Diffusion** We model numerical features through Gaussian diffusion and categorical features through multinomial diffusion. TabD-DPM (Kotelnikov et al., 2023) is a previous neural-based diffusion model combining Gaussian and multinomial diffusion. In such methods, numerical and categorical likelihoods contribute to a shared loss, while individual terms may lie on a different scale, causing gradients to be dominated by high-variance or high-cardinality features. Some works go to great lengths to weight the different terms in the loss function (Mueller et al., 2025). Our method circumvents the need for loss-weighting as we train separate XGBoost models per feature.

At each timestep $t$, we parameterize $\theta_t^j$ by an XGBoost regressor if $j \in \mathcal{M}$ or by an XGBoost classifier if $j \in \mathcal{C}$, conditioning on all features of the previous timestep $\mathbf{x}_{t+1}$. Numerical features are gaussianized by a quantile transformation followed by a $z$-scale normalization such that $x_j \sim N(0,1)$, $j \in \mathcal{M}$. To prevent OOD values we clip numerical features to the range of the training data after synthesis. Categorical features are integer-encoded, and we use XGBoost's native functionality to split directly on categories.

To prevent memorization, we apply a dropout procedure where we randomly mask numerical input features with probability $p = 0.1$ by setting them to the mean $\mu_{x_j} = 0.0$, $j \in \mathcal{M}$ at each timestep. We noticed that this reduces memorization without massively deteriorating fidelity. See Appendix B for more details and an ablation of various dropout levels. Increasing dropout forces the model to produce more typical samples, which can be useful in terms of privacy risk. Generating *realistic* but *atypical* samples risks resampling rare identifiable samples at the edge of the distribution. The dropout rate provides a tunable parameter to increase privacy protection at the cost of sample diversity.

### 3.2. Synthesizing Large Datasets: Autoregressive Models with XGBoost as Conditional Learner

To yield a model scalable to larger datasets, we must avoid massively extending the training set. We suggest modelling the likelihood of the data through autoregressive factorization. Fix a permutation of the features $\pi$ of $\{1, \ldots, d\}$. The autoregressive factorization defines a joint distribution over $\mathcal{X}$ via the chain rule:

$$p_\theta(x) = \prod_{t=1}^d p_\theta\big(x_{\pi(t)} \mid x_{\pi(<t)}\big), \quad (10)$$
$$x_{\pi(<t)} := \big(x_{\pi(1)}, \ldots, x_{\pi(t-1)}\big).$$

A (series of) predictor(s) can now parameterize $\theta$. Sampling proceeds sequentially $\tilde{x}_{\pi(t)} \sim p_{\hat{\theta}}\big(\cdot \mid \tilde{x}_{\pi(<t)}\big)$, for $t = 1, \ldots, d$. Since we use teacher-forcing, this approach

is only autoregressive at inference-time, and learning the conditionals can be parallelized.

The permutation order $\pi$ is fixed as the original order of the dataset. We tried various approaches, such as an ordering based on increasing/decreasing mutual information, but this did not yield a consistent improvement on all evaluation dimensions.

As a starting point, for numerical features $j \in \mathcal{M}$, we fit a linearly interpolated empirical quantile function $F_{\pi(0)}^{-1}$ : $[0, 1] \to \mathbb{R}$ on the first feature $\tilde{x}_{\pi(0)}$. Interpolated quantile functions can model non-continuous or skewed distributions efficiently. We then resample $u \sim \mathrm{Unif}(0, 1), \quad \tilde{x}_j = F_{\pi(0)}^{-1}(u), \quad j \in \mathcal{M}$. For categorical features $j \in \mathcal{C}$ we sample from the empirical marginal distribution of $\tilde{x}_{\pi(0)}$.

**Learning conditionals** To avoid mode collapse, it is vital to be able to sample from the posterior of $p_\theta\big(x_{\pi(t)} \mid x_{\pi(<t)}\big)$. For categorical features this can be done naturally by sampling from (temperature-scaled) predicted probabilities of a classifier. For numerical features, standard mean-estimating regression does not provide posterior probabilities, thus we need to find an appropriate method for probabilistic prediction. Several approaches based on tree ensembles exist, such as quantile regression, conditional diffusion (Beltran Velez et al., 2024), and hierarchical classification (McCarter, 2024). We opt for the latter as a natural extension to the multi-class classification approach.

For numerical features $j \in \mathcal{M}$ let $Q_j : \mathbb{R} \to \{1, \ldots, B_j\}$ denote a quantization map into $B_j$ bins. For categorical features $x_j \in \mathcal{C}$ we assume they are already integer-encoded, and thus $Q_j$ is simply the identity function. We define the autoregressive model over encoded target variables $p_\theta(x) = \prod_{t=1}^{d} p_\theta\big(Q_{\pi(t)}(x_{\pi(t)}) \mid x_{\pi(<t)}\big)$, where the conditional distributions are modelled by (hierarchical) classifiers. Sampling proceeds by drawing a bin $\tilde{b} \sim p_{\theta,\tau}(\cdot \mid x_{\pi(<t)})$.

For $j \in \mathcal{M}$, $Q_j$ is a rank-based binning approach. We implement this as a quantile-based binning where we add small amounts of rank-preserving noise[4] to ensure each bin contains the same number of samples. This yields balanced classification tasks in the hierarchical classifier, and prevents the risk of resampling sparse bins with potentially identifiable values, as is possible in uniform-binning or data-driven binning (e.g., k-means) approaches.

De-quantization should respect the non-continuous nature of most real-world tabular datasets. For each bin $b$, we fit a linearly interpolated empirical quantile function $F_{j,b}^{-1} : [0, 1] \to \mathbb{R}$ using the training samples assigned to that bin. Given a sampled bin $\tilde{b}$, we resample $u \sim \mathrm{Unif}(0, 1), \quad \tilde{x}_j = F_{j,\tilde{b}}^{-1}(u), \quad j \in \mathcal{M}$.

---

[4]Noise which does not alter the ordering of the data.

Appendix E provides an ablation of several quantization and de-quantization strategies. We find our de-quantization strategy to be especially effective at improving fidelity when compared to more naive approaches such as uniformly sampling from bins.

**Hierarchical Classification** UT's approach to probabilistic prediction recursively partitions numerical features into binary bins $H$ times, thus quantizing into $2^H$ bins (McCarter, 2024). Or analogously, builds a height-$H$ meta-tree of binary classifiers. Each node of the meta-tree trains a binary XGBoost classifier to learn the partitioning of the current datapoints, i.e., which were previously partitioned to said node. To obtain a bin prediction, we traverse the meta-tree by sampling split decisions from the binary classifiers.

We can compare this approach to a more naive method, where we directly model quantized numerical features using multi-class classifiers. The hierarchical approach has at least two advantages: i) it imposes an ordinal inductive bias, as early nodes route towards collections of bins which correspond to similar regions in the feature space, ii) and provides more model capacity per datapoint when the task is harder, i.e., bins are more fine-grained. Appendix D provides an ablation of the multi-class versus hierarchical classification approach, showing that the hierarchical classification approach more accurately preserves the multivariate structure.

**Limiting categorical cardinality** High-cardinality categorical features can explode training time in XGenB-AR, since multi-class XGBoost classifiers build a separate tree for each category per boosting round. We set a global constraint on categorical cardinality $K_{max}$. A naive method to enforce this constraint is to merge infrequent categories into a single category such that each feature $j \in \mathcal{C}$ has at most $K_{max}$ categories. We can record the empirical distribution of the original categories in the merged category and resample after synthesis.

We opt for a more informed approach. We cluster mean-vector embeddings of samples with infrequent categories into $K_c$ clusters, while retaining the $K_{max} - K_c$ most frequent categories as-is. Mean-vector embeddings are defined as the mean over numerical features scaled to $[0, 1]$ and categorical features one-hot encoded $\in \{0, \frac{1}{2}\}$. We use a KMediods clustering on L1 distances. The preprocessing scheme and clustering are chosen to resemble clustering based on Gower-like distances (Gower, 1971), see Appendix G.3 for more details on the rationale behind this choice. We record the empirical distributions of the original categories in each of the clustered categories, and resample after synthesis. Appendix C provides an ablation of the aforementioned naive approach versus the clustering approach for limiting categorical cardinality. Results indicate that the clustering

approach more accurately preserves multivariate structure by conditioning on (a summary of) the joint feature structure associated with each category.

An additional advantage of merging infrequent categories is that it helps prevent overfitting to rare, potentially identifiable, categories. Overall, it helps the model behave more predictably when applied out-of-the-box to datasets with potentially large cardinality.

## 4. Experiments

We separate the evaluation into two benchmarks, i.e., the benchmark from Jolicoeur-Martineau et al. (2024) and the benchmark from Mueller et al. (2025). Jolicoeur-Martineau et al. (2024)'s benchmark contains mostly smaller datasets and will be called the Small Benchmark, whereas Mueller et al. (2025)'s benchmark contains mostly larger datasets and will be called the Big Benchmark.

We omit XGenB-DF from the Big Benchmark, as it is prohibitively expensive to train for some of the larger datasets. We *do* include XGenB-AR in the Small Benchmark - whereas it was developed with large-scale tabular synthesis in mind, it can also be used for smaller datasets.

**Baseline models.** We compare XGenBoost against a variety of methods for tabular synthesis. Both the Small Benchmark and the Big Benchmark contain SMOTE, ARF, CT-GAN, TVAE, and TabDDPM as baseline models. SMOTE is an oversampling technique (not a model) which interpolates observations from the training set in data space (Chawla et al., 2002). ARF is an adversarial generative model based on Random Forests (Watson et al., 2023). CTGAN is a popular GAN-based (Goodfellow et al., 2014) method for tabular synthesis (Xu et al., 2019). TVAE is the VAE-based (Kingma & Welling, 2013) counterpart of CTGAN from the same article. TabDDPM is a diffusion model for mixed-type tabular data, combining Gaussian and multinomial diffusion (Kotelnikov et al., 2023).

For the Small Benchmark, we also include FD, FF, and UT (Jolicoeur-Martineau et al., 2024; McCarter, 2024). Due to the scaling issues mentioned in Section 2, these are not included in the Big Benchmark. For the Big Benchmark, we also include TabSyn, which is a latent diffusion model for mixed-type tabular data (Zhang et al., 2024). We do not include this for the Small Benchmark, as TabSyn requires training two deep generative models sequentially (VAE and diffusion model), which we do not believe to be sensible for small datasets.

Appendix F provides additional details regarding baseline models and hyperparameters, and Appendix H regarding hardware used for training.

**Datasets.** Both the Small Benchmark and the Big Benchmark contain tabular datasets with a mix of categorical (including binary) and numerical (including integer) features, and a variety of prediction tasks (classification/regression). Appendix A provides more details on the datasets contained in both benchmarks. For both benchmarks, we drop rows with missing values in numerical features, and split the data in train, validation, and test sets in 3:1:1 proportion. Although XGenBoost can model missing data, other baselines cannot; their quality depends on the imputation method when synthesizing in a missing-data setting, which we believe to be an unfair comparison.

**Evaluation metrics.** We evaluate generated synthetic data on its three archetypal dimensions, i.e., fidelity, utility, and privacy (Jordon et al., 2022). For fidelity, we investigate similarity in marginal densities (Shape) and bivariate correlations (Trend) (Zhang et al., 2024), distinguishability by a classifier (Detection, lower scores are better) (Zhang et al., 2024; Mueller et al., 2025), and distributional precision ($\alpha$-Precision) and recall ($\beta$-Recall) scores (Alaa et al., 2022). For utility, we evaluate Machine Learning Efficacy (MLE), i.e., the extent to which predictive models trained on synthetic data generalize to real data (Xu et al., 2019; Kotelnikov et al., 2023; Zhang et al., 2024). This is sometimes also called the Train Synthetic Test Real (TSTR) approach (Esteban et al., 2017). For privacy, we compute the Distance to Closest Record (DCR) scores, to investigate whether generated data is similarly close to the training data and an independent test set (Zhang et al., 2024). Low DCR scores indicate that synthetic data is often closer to the training set, which is a sign of overfitting, which in turn can be a sign of privacy leakage. We acknowledge, however, that DCR is a somewhat debatable measure for privacy risk. See Appendix G for more details on the considered evaluation metrics, including a discussion on the usage of DCR as a privacy measure.

We generate synthetic datasets at the same size of the real training data. For the Detection score metric, we query an additional synthetic set at the size of the real test set to evaluate generalization of the classifier to the real and synthetic test sets. We cap the size of real and synthetic sets at 50 000 rows to limit evaluation time. We use 20 different seeds which affect the sampling process of the generator and the subsampling process of the real dataset (if applicable).

### 4.1. Results

Table 1 and Table 2 provide the average ranks of each generator across the datasets of the Small Benchmark and Big Benchmark, respectively, for the considered metrics. Appendix I provides detailed results. For the MLE metric, ranks are based on ROCAUC for classification datasets and $R^2$ for regression datasets. Other MLE metrics (F1

and RMSE) can be found in Appendix I. Generators which could not be run for a specific dataset receive the highest possible rank. This is the case for SMOTE in the covertype and acsincome dataset in the Big Benchmark (too costly, see also Mueller et al. (2025)), TabDDPM in the diabetes and acsincome dataset in the Big Benchmark (NaNs during training, see also Mueller et al. (2025)), and SMOTE in the ecoli dataset in the Small Benchmark ($< 2$ samples in one of the target classes).

**Small Benchmark.** Table 1 shows that XGenBoost consistently outperforms other generators in terms of fidelity (Shape, Trend, Detection). Here, XGenB-AR is especially apt at preserving marginal distributions (Shape), whereas XGenB-DF is better at preserving correlations (Trend), and therefore also machine learning utility (MLE). XGenB-DF can model correlations at a higher level of granularity than XGenB-AR, as the latter quantizes numerical features.

XGenB-DF's DDPM specifications tend to yield more diverse samples - due to its stochastic nature - resulting in higher $\beta$-Recall scores. Unfortunately, this can go hand in hand with more privacy risk, measured as worse DCR scores. High $\beta$-Recall indicates that realistic samples are being generated at the edge of the distribution. Analogously, this may seem like copying outliers from the training data, which increases reidentification risk. Furthermore, XGenB-DF's velocity-prediction reparameterizations yield slightly worse fidelity, but also decreased privacy risk. It provides a smoother loss trajectory at low noise levels, and therefore less risk of overfitting (Gao et al., 2024).

Overall, we find a DDIM sampler and a velocity-prediction reparametrization to yield the most balanced results. This specification of XGenB-DF outperforms all baseline models in terms of fidelity (Shape, Trend, Detection), and all baselines except SMOTE in terms of utility (MLE). In terms of privacy (DCR), it outperforms other high-fidelity methods SMOTE, FF, and UT. Due to the inherent fidelity-privacy tradeoff in synthetic data (Achterberg et al., 2025), other methods may achieve better privacy scores if they produce samples with much worse fidelity (CTGAN, TVAE, etc.).

**Big Benchmark.** Table 2 shows that XGenB-AR outperforms other generators in terms of fidelity, utility, and training time. In terms of privacy risk, XGenB-AR outranks SMOTE and ARF, while scoring similar to deep generative models such as TabDDPM and TVAE. This indicates that XGenB-AR generates synthetic data with increased fidelity while maintaining privacy protection, at much lower training cost.

**Scalability** Figure 1 shows the training times in minutes of XGenBoost in the Small and Big Benchmark. For most datasets, XGenB-DF trains within a reasonable time - a few

minutes - on our modest hardware (see Appendix H). Factors which negatively influence runtime are i) large numbers of features, as seen in datasets libras, connectionist bench sonar, qsar biodegradation, and ionosphere, ii) large numbers of rows, as seen in datasets california and bean, and iii) large numbers of categorical target labels, as seen in datasets libras and bean.

XGenB-AR trains within a reasonable time on all datasets of the Big Benchmark - even for datasets with many rows (acsincome), or many rows *and* many features (covertype). On the other hand, XGenB-AR's sequential inference over features invites scepticism regarding inference-scalability to datasets with many features. However, for the datasets in the Big Benchmark, we found that inference time is on-par with that of diffusion models such as TabDDPM and TabSyn (see Table 2). This holds even for datasets with many features (covertype, news), see Table 30. However, we recognize that inference time may become an issue when datasets have hundreds or thousands of features.

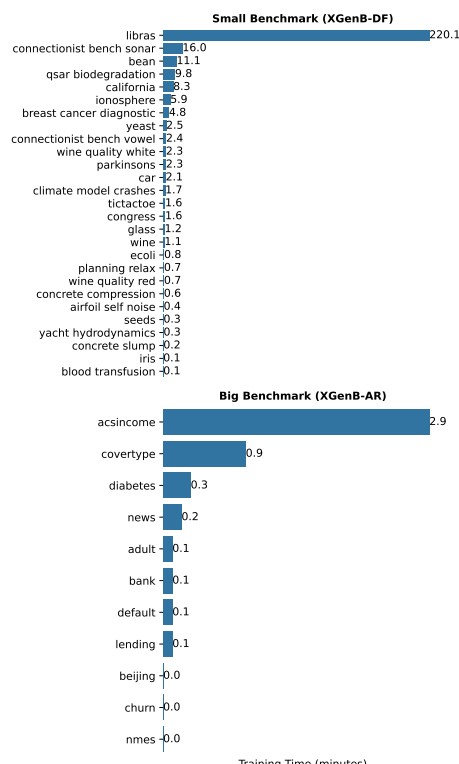

*Figure 1.* Training times in minutes of XGenB-DF (v-DDIM specification) and XGenB-AR in the Small and Big Benchmark, respectively.

## 5. Limitations

None of XGenBoost's models have currently been adapted for imputation. For the autoregressive model, we can relax the autoregressive factorization to a Besag pseudo-

*Table 1.* Small Benchmark: metric ranks ($\mu \pm \sigma$) across the 27 datasets of the Small Benchmark. Best results per metric are in **bold**, second best underlined.

| metric | Shape | Trend | Detection | $\alpha$-Precision | $\beta$-Recall | MLE | DCR | Training Time | Inference Time |
|---|---|---|---|---|---|---|---|---|---|
| SMOTE | $6.654_{\pm 2.756}$ | $5.154_{\pm 3.233}$ | $6.885_{\pm 2.805}$ | $7.769_{\pm 3.491}$ | $3.462_{\pm 3.289}$ | $\underline{4.808}_{\pm 3.805}$ | $10.923_{\pm 2.637}$ | $nan_{\pm nan}$ | $2.769_{\pm 1.177}$ |
| ARF | $7.852_{\pm 2.641}$ | $9.704_{\pm 1.938}$ | $9.222_{\pm 2.044}$ | $5.333_{\pm 3.063}$ | $10.630_{\pm 0.742}$ | $10.037_{\pm 2.457}$ | $3.278_{\pm 1.734}$ | $\underline{2.074}_{\pm 1.072}$ | $3.926_{\pm 0.829}$ |
| UT | $7.222_{\pm 2.532}$ | $7.148_{\pm 2.627}$ | $7.037_{\pm 2.993}$ | $6.741_{\pm 3.046}$ | $7.407_{\pm 1.474}$ | $7.370_{\pm 2.452}$ | $8.093_{\pm 3.000}$ | $5.407_{\pm 0.888}$ | $12.481_{\pm 1.221}$ |
| FD | $9.407_{\pm 1.623}$ | $5.407_{\pm 2.859}$ | $8.407_{\pm 3.165}$ | $7.000_{\pm 3.563}$ | $8.481_{\pm 2.190}$ | $5.333_{\pm 3.026}$ | $6.648_{\pm 2.269}$ | $10.630_{\pm 1.573}$ | $7.185_{\pm 0.483}$ |
| FF | $7.444_{\pm 2.326}$ | $5.000_{\pm 2.617}$ | $6.556_{\pm 3.055}$ | $6.926_{\pm 2.401}$ | $\underline{3.185}_{\pm 2.573}$ | $5.037_{\pm 2.579}$ | $9.278_{\pm 2.419}$ | $10.074_{\pm 0.958}$ | $5.593_{\pm 0.636}$ |
| CTGAN | $12.407_{\pm 0.694}$ | $12.296_{\pm 0.609}$ | $12.296_{\pm 0.775}$ | $11.815_{\pm 1.962}$ | $12.630_{\pm 0.451}$ | $12.667_{\pm 0.555}$ | $\underline{2.685}_{\pm 2.284}$ | $3.741_{\pm 0.594}$ | $\mathbf{1.630}_{\pm 0.629}$ |
| TVAE | $11.593_{\pm 0.694}$ | $11.444_{\pm 1.013}$ | $11.481_{\pm 0.849}$ | $9.963_{\pm 2.835}$ | $11.259_{\pm 0.984}$ | $10.704_{\pm 2.233}$ | $\mathbf{2.537}_{\pm 2.406}$ | $2.852_{\pm 0.602}$ | $\underline{2.074}_{\pm 0.917}$ |
| TabDDPM | $9.111_{\pm 3.836}$ | $10.148_{\pm 3.890}$ | $8.519_{\pm 3.877}$ | $8.556_{\pm 4.484}$ | $8.074_{\pm 4.164}$ | $7.185_{\pm 4.707}$ | $7.352_{\pm 4.090}$ | $7.741_{\pm 3.108}$ | $11.111_{\pm 2.342}$ |
| XGenB-AR | $\mathbf{2.278}_{\pm 1.883}$ | $6.556_{\pm 2.926}$ | $\mathbf{3.593}_{\pm 2.606}$ | $\mathbf{3.037}_{\pm 2.426}$ | $8.630_{\pm 1.006}$ | $7.556_{\pm 2.636}$ | $5.241_{\pm 1.772}$ | $\mathbf{1.370}_{\pm 0.492}$ | $5.185_{\pm 0.962}$ |
| XGenB-DF (x-DDPM) | $5.093_{\pm 2.094}$ | $5.074_{\pm 2.336}$ | $5.222_{\pm 2.309}$ | $\underline{4.741}_{\pm 3.083}$ | $\mathbf{3.037}_{\pm 1.285}$ | $\mathbf{4.593}_{\pm 2.531}$ | $9.926_{\pm 2.348}$ | $9.370_{\pm 1.925}$ | $10.444_{\pm 1.281}$ |
| XGenB-DF (v-DDPM) | $4.259_{\pm 1.873}$ | $4.741_{\pm 2.068}$ | $4.074_{\pm 2.448}$ | $7.148_{\pm 3.072}$ | $3.815_{\pm 1.469}$ | $4.926_{\pm 2.055}$ | $8.796_{\pm 2.435}$ | $9.148_{\pm 1.134}$ | $10.296_{\pm 0.869}$ |
| XGenB-DF (x-DDIM) | $3.926_{\pm 2.336}$ | $\mathbf{4.000}_{\pm 2.465}$ | $4.185_{\pm 2.149}$ | $4.815_{\pm 2.923}$ | $4.667_{\pm 1.519}$ | $5.370_{\pm 2.662}$ | $8.278_{\pm 2.318}$ | $7.741_{\pm 1.559}$ | $9.296_{\pm 1.171}$ |
| XGenB-DF (v-DDIM) | $\underline{3.519}_{\pm 2.260}$ | $\underline{4.037}_{\pm 2.328}$ | $\underline{3.741}_{\pm 2.280}$ | $6.963_{\pm 2.752}$ | $5.370_{\pm 1.621}$ | $5.111_{\pm 2.722}$ | $7.889_{\pm 1.631}$ | $7.852_{\pm 1.610}$ | $9.074_{\pm 1.174}$ |

*Table 2.* Big Benchmark: metric ranks ($\mu \pm \sigma$) across the 11 datasets of the Big Benchmark. Best results per metric are in **bold**, second best underlined.

| metric | Shape | Trend | Detection | $\alpha$-Precision | $\beta$-Recall | MLE | DCR | Training Time | Inference Time |
|---|---|---|---|---|---|---|---|---|---|
| SMOTE | $3.889_{\pm 0.782}$ | $3.111_{\pm 0.782}$ | $3.444_{\pm 0.726}$ | $5.667_{\pm 1.323}$ | $\mathbf{1.778}_{\pm 1.563}$ | $\mathbf{2.111}_{\pm 1.364}$ | $6.778_{\pm 0.441}$ | - | $4.556_{\pm 0.882}$ |
| ARF | $3.818_{\pm 0.982}$ | $6.364_{\pm 1.206}$ | $4.727_{\pm 0.467}$ | $2.636_{\pm 1.027}$ | $4.545_{\pm 1.508}$ | $2.909_{\pm 1.514}$ | $5.727_{\pm 0.786}$ | $3.818_{\pm 0.874}$ | $6.000_{\pm 0.775}$ |
| CTGAN | $6.000_{\pm 1.000}$ | $5.273_{\pm 0.905}$ | $6.455_{\pm 0.688}$ | $5.636_{\pm 1.206}$ | $6.091_{\pm 0.701}$ | $6.182_{\pm 0.982}$ | $\mathbf{2.727}_{\pm 1.348}$ | $3.273_{\pm 0.467}$ | $\underline{2.364}_{\pm 0.674}$ |
| TVAE | $6.000_{\pm 0.775}$ | $4.909_{\pm 0.539}$ | $6.364_{\pm 0.505}$ | $5.273_{\pm 1.104}$ | $5.364_{\pm 1.433}$ | $5.455_{\pm 1.036}$ | $3.000_{\pm 1.414}$ | $\underline{2.091}_{\pm 0.302}$ | $\mathbf{1.364}_{\pm 0.674}$ |
| TabDDPM | $3.889_{\pm 1.691}$ | $4.000_{\pm 1.323}$ | $3.667_{\pm 1.658}$ | $3.889_{\pm 1.764}$ | $3.333_{\pm 2.062}$ | $4.333_{\pm 1.732}$ | $3.222_{\pm 1.787}$ | $4.778_{\pm 0.441}$ | $7.000_{\pm 0.000}$ |
| TabSyn | $\underline{2.364}_{\pm 0.809}$ | $\underline{1.818}_{\pm 0.751}$ | $\underline{2.364}_{\pm 0.924}$ | $\underline{2.364}_{\pm 1.286}$ | $2.727_{\pm 0.905}$ | $3.000_{\pm 1.549}$ | $\underline{2.818}_{\pm 1.779}$ | $5.818_{\pm 0.405}$ | $4.909_{\pm 0.701}$ |
| XGenB-AR | $\mathbf{1.000}_{\pm 0.000}$ | $\mathbf{1.364}_{\pm 0.505}$ | $\mathbf{1.818}_{\pm 1.168}$ | $\mathbf{1.818}_{\pm 0.982}$ | $\underline{2.636}_{\pm 0.924}$ | $\underline{2.727}_{\pm 1.421}$ | $3.091_{\pm 1.136}$ | $\mathbf{1.000}_{\pm 0.000}$ | $3.455_{\pm 0.688}$ |

likelihood objective (Besag, 1975) and turn the model into a dependency network as described in Heckerman et al. (2000). This allows for any-order synthesis, and thus imputation, through Gibbs sampling. This approach may scale better to large datasets than the training set extension and random-order synthesis approach of UT (McCarter, 2024). The diffusion model can be used for imputation through an inpainting method, e.g., REPAINT (Lugmayr et al., 2022), similar to FD (Jolicoeur-Martineau et al., 2024).

XGenBoost's autoregressive and diffusion models can be prone to overfitting, which may increase privacy risk. In the diffusion model this may present as memorizing training samples at early timesteps, and in the autoregressive model, as preserving the exact rank-order structure from the training data. We take several measures against this. For the diffusion model, we can increase the dropout rate to make the model more robust against memorizing sample coordinates. In the autoregressive model, we can reduce the capacity of the boosters using standard hyperparameters to reduce the sharpness of learned conditionals, or decrease quantization granularity by reducing the meta-tree height.

## 6. Conclusion and Discussion

We presented XGenBoost, a pair of generative architectures for mixed-type tabular synthesis based on XGBoost. We present a Denoising Diffusion Implicit Model (DDIM) using XGBoost as score-estimator, which is especially suited to smaller tabular datasets. We also present a hierarchical autoregressive model, using XGBoost as conditional learner, which scales well to larger datasets. We show that such architectures, when designed according to the constraints and strengths of tree learners, can yield models capable of outperforming deep generative models in terms of both quality and computational costs. We hereby challenge the current paradigm which often adapts effective architectures from other fields such as NLP and computer vision, and rather, design tabular synthesizers based on methods which are known to have suitable inductive biases for mixed-type tabular data. Additionally, since XGenBoost contains separate methods for small-scale and large-scale tabular synthesis, it is applicable to a broad set of use cases. This data-first approach is aligned with real-world applications where users will choose an applicable model based on the characteristics of their dataset.

There are various potential extensions to XGenBoost's architectures. For XGenB-DF we can consider diffusion-specific extensions such as adaptive noise schedules, or different (especially discrete) diffusion kernels. For XGenB-AR we can consider, e.g., a more informed sequence order. Lastly, XGenB-DF and XGenB-AR can be combined into a hybrid approach, where XGenB-DF is used to first synthesize a set of low-risk or complex features with very high fidelity, and XGenB-AR subsequently synthesizes more private or simpler features at reduced privacy risk and lower training cost, conditioned on features synthesized by XGenB-DF. We leave these extensions for future research.

## 7. Impact Statement

This paper presents methods which generate synthetic observations similar to those found in real-world datasets. Proper care should be taken when using synthetic data to draw any inferences, as there are no guarantees that inferences drawn from synthetic data generalize to the real-world. Results should always be compared to results from real-world data.

Synthetic data can be a useful tool to share otherwise sensitive data, as observations do not pertain to real individuals. However, information from the real data may leak through to the synthetic data, potentially disclosing sensitive information. Malicious third parties can potentially combine synthetic data with quasi-identifiers available to them to draw inferences on real individuals. When using the methods from this work to share sensitive data, always perform a rigorous privacy evaluation first, consisting of (at least) various attribute and membership disclosure attacks. Here it is vital to make realistic assumptions about the data and resources available to a potential attacker.

One of the main goals of this work is democratizing access to strong methods for tabular synthesis. Previous performant methods often rely on modern computing resources, which are known to be unfairly distributed across the world (Lehdonvirta et al., 2024). Our presented methods run in reasonable times on modest CPU set-ups, even for datasets with millions of rows and dozens of features. We hereby ensure access to these methods, which can be beneficial for a large variety of use cases, for not only the privileged, but the entire world. Additionally, reducing computational requirements reduces energy demand, which can be beneficial from both a financial and sustainability perspective.

Lastly, this work designs methods by taking a data-first approach, designing generative architectures using learners which are already known to have suitable inductive priors for the type of data considered. This is in contrast to many other efforts which adapt strong methods from other fields and data types, e.g., NLP and computer vision, to the tabular data domain. We believe such data-first approaches have the potential to yield equally strong (or stronger) methods at reduced computational load. As these approaches already encode suitable inductive priors, these no longer have to be learned by generalistic learners, thereby saving computation.

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

## A. Datasets

Table 3 and Table 4 list the datasets contained in the Small Benchmark and Big Benchmark, respectively. The Small Benchmark was originally presented by Jolicoeur-Martineau et al. (2024), which was in turn inspired by Muzellec et al. (2020). The Big Benchmark was originally presented by Mueller et al. (2025). All datasets are openly available. All datasets except iris and nmes are licensed under a Creative Commons license; the iris dataset is licensed under BSD 3-Clause License, and the nmes dataset has no known license.

*Table 3.* Small Benchmark: overview of datasets.

| Dataset | Task | N | No. of features | | No. of categories | |
| --- | --- | --- | --- | --- | --- | --- |
| | | | categorical | numerical | min | max |
| airfoil self noise (Brooks et al., 2014) | Regression | 1503 | 0 | 6 | 0 | 0 |
| bean (Koklu & Ozkan, 2020) | Classification | 13611 | 1 | 16 | 7 | 7 |
| blood transfusion (Yeh et al., 2009) | Classification | 748 | 1 | 4 | 2 | 2 |
| breast cancer diagnostic (Street et al., 1993) | Classification | 569 | 1 | 30 | 2 | 2 |
| california (Pace & Barry, 1997) | Regression | 20640 | 0 | 9 | 0 | 0 |
| car (Bohanec & Rajkovic, 1988) | Classification | 1728 | 7 | 0 | 3 | 4 |
| climate model crashes (Lucas et al., 2013) | Classification | 540 | 1 | 18 | 2 | 2 |
| concrete compression (Yeh, 1998) | Regression | 1030 | 0 | 9 | 0 | 0 |
| concrete slump (Yeh, 2009) | Regression | 103 | 0 | 8 | 0 | 0 |
| congress (dat, 1987) | Classification | 435 | 17 | 0 | 2 | 3 |
| connectionist bench sonar (Sejnowski & Gorman) | Classification | 208 | 1 | 60 | 2 | 2 |
| connectionist bench vowel (Deterding et al.) | Classification | 990 | 1 | 10 | 11 | 11 |
| ecoli (Horton & Nakai, 1996) | Classification | 336 | 1 | 7 | 8 | 8 |
| glass (German, 1987) | Classification | 214 | 1 | 9 | 6 | 6 |
| ionosphere (Sigillito et al., 1989) | Classification | 351 | 2 | 32 | 2 | 2 |
| iris (Unwin & Kleinman, 2021) | Classification | 150 | 1 | 4 | 3 | 3 |
| libras (Dias et al., 2009) | Classification | 360 | 1 | 90 | 15 | 15 |
| parkinsons (Little et al., 2007) | Classification | 195 | 1 | 22 | 2 | 2 |
| planning relax (Bhatt, 2012) | Classification | 182 | 1 | 12 | 2 | 2 |
| qsar biodegradation (Mansouri et al., 2013) | Classification | 1055 | 4 | 38 | 2 | 2 |
| seeds (Charytanowicz et al., 2010) | Classification | 210 | 1 | 7 | 3 | 3 |
| tictactoe (Aha, 1991) | Classification | 958 | 10 | 0 | 2 | 3 |
| wine (Aeberhard et al., 1994) | Classification | 178 | 1 | 13 | 3 | 3 |
| wine quality red (Cortez et al., 2009) | Regression | 1599 | 0 | 11 | 0 | 0 |
| wine quality white (Cortez et al., 2009) | Regression | 4898 | 0 | 12 | 0 | 0 |
| yacht hydrodynamics (Gerritsma et al., 2013) | Regression | 308 | 0 | 7 | 0 | 0 |
| yeast (Nakai, 1996) | Classification | 1484 | 1 | 8 | 10 | 10 |

## B. Dropout: Preventing Memorization in Diffusion Models

At low noise levels, the input data to each XGBoost model essentially consists of $N$ clusters of slightly noised points originating from the same sample, where each cluster contains $K$ points - $K$ being the number of times we extend the training dataset. This can lead to a risk of overfitting to these clusters locally, which in a broader sense, implies memorization of the originating sample.

To mitigate memorization, we apply a dropout procedure to the numerical features, similar as may be used in neural networks. That is, we randomly mask the input features with probability $p$. As a masking token we use the mean $\mu_x = 0.0$. Other viable candidates are using a missing token, as XGBoost can natively handle missing data, or random replacement; we tried both approaches. However, as XGBoost essentially treats missing tokens as unique (OOD) values, it can learn spurious correlations, which deteriorates fidelity. Random replacement can also inject spurious correlations in an unpredictable manner. We noticed the best results when using the mean as masking token. This way, dropout does not necessarily deteriorate fidelity, but rather, controls the amount of high-fidelity samples which are generated at either the mode or the edges of the distribution. It allows a trade-off between sample diversity and privacy risk.

*Table 4.* Big Benchmark: overview of datasets.

| Dataset | Task | N | No. of features | | No. of categories | |
| --- | --- | --- | --- | --- | --- | --- |
| | | | categorical | numerical | min | max |
| acsincome (Ding et al., 2021) | Regression | 1 664 500 | 8 | 3 | 2 | 529 |
| adult (Becker & Kohavi, 1996) | Classification | 48 842 | 9 | 6 | 2 | 42 |
| bank (Moro et al., 2014) | Classification | 41 188 | 11 | 10 | 2 | 12 |
| beijing (Liang et al., 2015) | Regression | 41 757 | 1 | 11 | 4 | 4 |
| churn (Keramati et al., 2014) | Classification | 3 150 | 5 | 9 | 2 | 5 |
| covertype (Blackard, 1998) | Regression | 581 012 | 43 | 12 | 2 | 2 |
| default (Yeh & Lien, 2009) | Classification | 30 000 | 10 | 14 | 2 | 9 |
| diabetes (Strack et al., 2014) | Classification | 101 766 | 28 | 9 | 2 | 790 |
| lending (Lending Club, 2015) | Regression | 9 182 | 10 | 34 | 2 | 4355 |
| news (Fernandes et al., 2015) | Regression | 39 644 | 14 | 46 | 2 | 2 |
| nmes (Deb & Trivedi, 1997) | Regression | 4406 | 8 | 11 | 2 | 4 |

Figure 2 shows distributions of metric scores for dropout levels $\in [0.0, 0.3]$. The most notable changes are the increasing $\alpha$-Precision and decreasing $\beta$-Recall scores, clearly showing the trade-off between generating high fidelity samples at the mode or at the edge of the distribution. The DCR score also increases, as seen by an increase of the low-end of the distribution, and more probability mass at the top. The Detection score remains similar, showing that overall fidelity is retained.

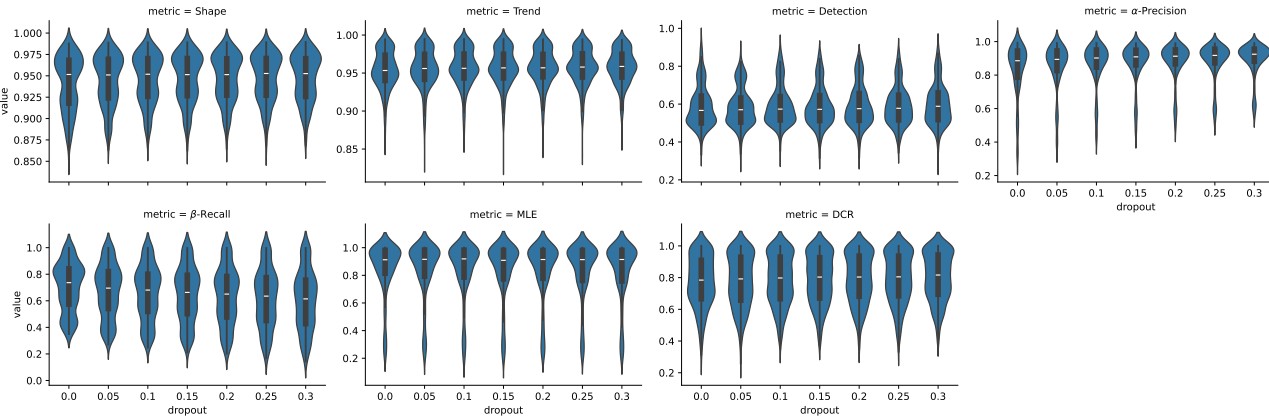

*Figure 2.* Violinplot of metric scores for various levels of dropout in XGenB-DF, evaluated for 20 sampling seeds per dataset in the Small Benchmark. MLE uses ROCAUC and $R^2$ for classification and regression, respectively.

Finally, note that this procedure is different from native XGBoost parameters which allow to subsample columns. Those subsampling procedures sample *the same columns for all samples*, either at each tree, level, or node, whereas we subsample *different columns per sample*. Our goal is to make the model invariant to sample-specific correlations, not to globally regularize the model.

## C. Merging High-Cardinality Categorical Features

Table 5 provides a comparison of the naive method for merging high-cardinality categorical features with the cluster-based method. The clustering approach outperforms the naive approach in terms of Detection score for 3/4 datasets. Especially for the acsincome dataset, the improvement is considerable. The fidelity gains can mostly be attributed to improved preservation of correlations (Trend) and more realistic diverse samples ($\beta$-Recall).

For some datasets, e.g., the adult dataset, the cluster-approach may not improve results compared to the naive approach. Whether the clustering approach improves results depends on whether the collection of infrequent categories can meaningfully be clustered into similar groups. If samples containing the different infrequent categories are similar - in terms of their

mean-vector embeddings - then clustering will not yield better results than a naive merging approach.

*Table 5.* Ablation: effect of merging infrequent categories in high-cardinality categorical features through a clustering approach versus a naive approach - clustering all infrequent categories into a single category. Metric scores ($\mu \pm \sigma$) over 20 sampling seeds for datasets containing high-cardinality categorical features ($K > 32$). MLE uses ROCAUC and $R^2$ for classification and regression, respectively.

| dataset | acsincome | | adult | | diabetes | | lending | |
|---|---|---|---|---|---|---|---|---|
| model | Clustering | Naive | Clustering | Naive | Clustering | Naive | Clustering | Naive |
| metric | | | | | | | | |
| Shape | $0.987_{\pm 0.001}$ | $0.987_{\pm 0.001}$ | $0.994_{\pm 0.001}$ | $0.994_{\pm 0.001}$ | $0.995_{\pm 0.000}$ | $0.995_{\pm 0.000}$ | $0.984_{\pm 0.001}$ | $0.984_{\pm 0.001}$ |
| Trend | $0.946_{\pm 0.002}$ | $0.942_{\pm 0.002}$ | $0.980_{\pm 0.003}$ | $0.980_{\pm 0.003}$ | $0.978_{\pm 0.002}$ | $0.977_{\pm 0.002}$ | $0.952_{\pm 0.002}$ | $0.952_{\pm 0.002}$ |
| Detection | $0.689_{\pm 0.003}$ | $0.764_{\pm 0.003}$ | $0.614_{\pm 0.004}$ | $0.614_{\pm 0.004}$ | $0.774_{\pm 0.002}$ | $0.781_{\pm 0.003}$ | $0.652_{\pm 0.015}$ | $0.660_{\pm 0.012}$ |
| $\alpha$-Precision | $0.992_{\pm 0.003}$ | $0.993_{\pm 0.002}$ | $0.997_{\pm 0.001}$ | $0.996_{\pm 0.001}$ | $0.996_{\pm 0.001}$ | $0.996_{\pm 0.001}$ | $0.972_{\pm 0.007}$ | $0.977_{\pm 0.007}$ |
| $\beta$-Recall | $0.436_{\pm 0.002}$ | $0.413_{\pm 0.002}$ | $0.478_{\pm 0.002}$ | $0.478_{\pm 0.003}$ | $0.389_{\pm 0.003}$ | $0.389_{\pm 0.003}$ | $0.479_{\pm 0.010}$ | $0.474_{\pm 0.007}$ |
| MLE | $0.577_{\pm 0.218}$ | $0.575_{\pm 0.225}$ | $0.811_{\pm 0.112}$ | $0.810_{\pm 0.113}$ | $0.630_{\pm 0.057}$ | $0.634_{\pm 0.060}$ | $0.528_{\pm 0.471}$ | $0.527_{\pm 0.472}$ |
| DCR | $0.998_{\pm 0.002}$ | $0.998_{\pm 0.003}$ | $0.984_{\pm 0.007}$ | $0.981_{\pm 0.005}$ | $0.991_{\pm 0.005}$ | $0.987_{\pm 0.005}$ | $0.950_{\pm 0.015}$ | $0.948_{\pm 0.010}$ |

## D. Benefit of Hierarchical Classification

Table 6 provides a comparison between using multi-class XGBoost classifiers to model numerical features in XGenB-AR, or using a hierarchical classification approach similar to that of UT (McCarter, 2024).

Results indicate that the hierarchical classifier outranks the multi-class approach in terms of preserving correlation structure (Trend), overall multivariate structure (Detection), and diversity ($\beta$-Recall). The average improvement is especially considerable for $\beta$-Recall. This, and other fidelity gains, can mostly be attributed to the ordinal inductive bias imposed by the hierarchical classifier, which is absent in the multi-class classifier. Specifically, the hierarchical classifier localizes prediction errors to semantically similar bins, whereas the multi-class classifier may sample distant - and thus "worse" - bins.

*Table 6.* Ablation: comparison between hierarchical classification and a multi-class XGBoost classifier for numerical features. Shows wins and losses of the hierarchical approach versus the multi-class approach over the 11 datasets of the Big Benchmark. Also shows corresponding average improvement/reduction in metric scores.

| metric | Shape | Trend | Detection | $\alpha$-Precision | $\beta$-Recall | MLE | DCR | Training Time | Inference Time |
|---|---|---|---|---|---|---|---|---|---|
| Win | 5 (0.08%) | 9 (0.27%) | 9 (6.07%) | 4 (0.59%) | 10 (11.93%) | 6 (8.30%) | 3 (0.10%) | 3 (29.59%) | 1 (6.67%) |
| Loss | 6 (0.07%) | 2 (0.01%) | 2 (1.00%) | 7 (0.85%) | 1 (1.68%) | 5 (0.60%) | 8 (0.46%) | 8 (89.98%) | 10 (80.72%) |

## E. Quantization and De-quantization strategies

Figure 3 shows the effect of varying quantization and de-quantization strategies in XGenB-AR. Specifically, we investigate a quantile-, uniform-, and KMeans-based quantization, and a uniform- and EQF-sampling (empirical quantile function) dequantization. Here, the quantile-based binning strategy corresponds to the rank-based binning: a quantile-binning with added rank-preserving noise to yield bins with similar amounts of samples.

Results indicate that especially the dequantization strategy has a strong impact on all evaluation metrics, whereby EQF-sampling strongly improves fidelity and utility over uniform-sampling. The quantization strategy is less impactful, although the quantile-based binning tends to yield slightly lower/better Detection scores. An additional advantage is that this binning approach yields balanced classification tasks at each node of the hierarchical classifier, as each bin contains the same number of samples, such that the binary classifiers behave more predictably. It may also prevent resampling rare values from sparse bins, which can be better in terms of privacy risk, although the aggregated DCR scores do not show any meaningful improvement.

## F. Implementation Details

Below we describe implementation details regarding XGenBoost and baseline models, e.g., package versions and hyper-parameters. For each method we round numerical features to the precision of the original dataset after synthesis. We use Python version 3.10 for all methods.

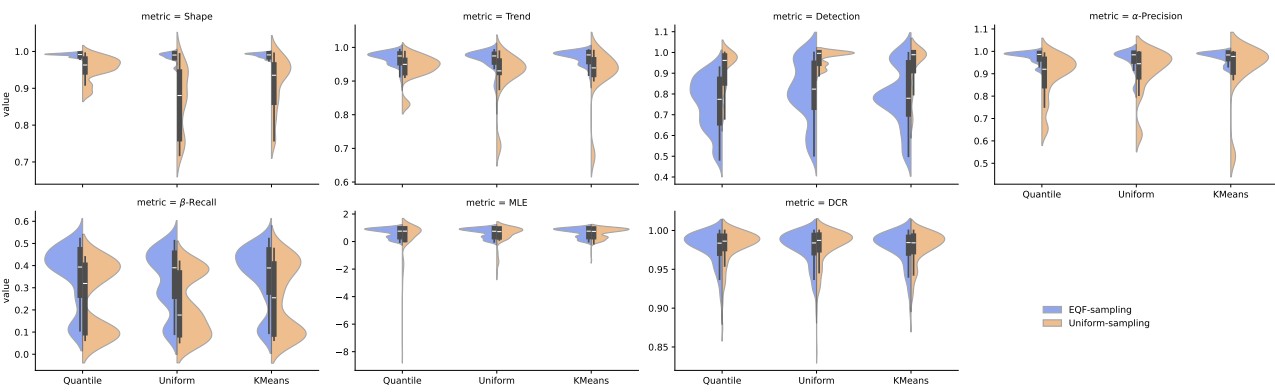

*Figure 3.* Violinplot of metric scores for various quantization strategies (Quantile, Uniform, KMeans) and de-quantization strategies (Uniform-sampling, EQF-sampling) in XGenBoost's autoregressive model, evaluated for 20 sampling seeds per dataset in the Big Benchmark. MLE shows ROCAUC and $R^2$ for classification and regression datasets, respectively.

**XGenBoost** For XGenB-DF we use the same XGBoost hyperparameters as FD and FF (Jolicoeur-Martineau et al., 2024), corresponding to 100 boosting rounds, maximum tree depth of 7, and no L2 regularization term. We also use the same number of timesteps ($T = 50$), noise levels per sample ($K = 100$), and minimum/maximum noise rates ($\beta_{\min} = 0.1$, $\beta_{\max} = 8.0$). These were chosen to allow for a fair comparison, not because they were found to be most effective. Results can potentially be improved by tuning for a better overall or per-dataset set of hyperparameters.

For XGenB-AR we use 30 boosting rounds and max depth of 3. In the hierarchical classifier we use a meta-tree height of 5.

We use XGBoost version 2.1.4.

**SMOTE** We use the SMOTENC, SMOTEN, or regular SMOTE implementation from the imbalanced-learn Python package, dependent on whether datasets are mixed-type, purely categorical, or purely numerical, respectively. We use the default hyperparameters. The implementation is mainly inspired by Mueller et al. (2025), which was in turn inspired by Kotelnikov et al. (2023).

**ARF** We use the default hyperparameters as given in the original paper (Watson et al., 2023), and the implementation from the arfpy Python package.

**Forest Diffusion and Forest Flow** We use the default hyperparameters as given in the original paper (Jolicoeur-Martineau et al., 2024), and the implementation from the ForestDiffusion Python package.

**Unmasking Trees** We use the default hyperparameters as given in the original paper (McCarter, 2024), and the implementation from the unmaskingtrees Python package.

**CTGAN and TVAE** We mostly use the default hyperparameters as given in the original paper (Xu et al., 2019), and the implementation from the ctgan Python package.

Per default we train for 300 epochs, but we cap epochs such that CTGAN and TVAE train for at most 30 000 training steps on each dataset. This ensure reasonable training times on larger datasets. Some works choose to train for a fixed number of training steps on all datasets for this reason (Mueller et al., 2025). However, this leads to an unnecessarily large number of training epochs for smaller datasets. Some works also choose to further enlarge the generator/discriminator (encoder/decoder in TVAE) to match the parameter count of, e.g., diffusion models (Zhang et al., 2024; Mueller et al., 2025). Although this can be useful for comparison sake, we notice that this does not necessarily improve results.

**TabDDPM** We use the implementation of TabDDPM in the Synthcity Python package. We slightly modify the code to allow manual specification of categorical features.

The original paper on TabDDPM does not provide a default set of hyperparameters, but rather tunes parameters per dataset (Kotelnikov et al., 2023). For the Big Benchmark, we utilize the optimal hyperparameters found on the adult dataset in

the original TabDDPM paper, as this is the closest to the datasets in the Big Benchmark in terms of size, and in terms of having reasonable numbers of both numerical and categorical features. This is a similar approach as followed in the Big Benchmark's original paper, Mueller et al. (2025). For the Small Benchmark, we utilize the optimal hyperparameters found on the California dataset, similar as in the Small Benchmark's original paper, Jolicoeur-Martineau et al. (2024).

Lastly, as Synthcity does not allow variable size hidden layers in the denoising network, we modify the denoising architecture while approximately matching the number of trainable parameters. Similarly, Synthcity specifies epochs instead of training steps, such that we calculate the corresponding number of epochs from the given training steps, batch size, and training set size of the California dataset in the TabDDPM paper: epochs $= \dfrac{\text{batch size} \cdot \text{steps}}{N}$. Similar to CTGAN we cap epochs such that TabDDPM trains for at most 30 000 training steps on each dataset.

**TabSyn** We use the default hyperparameters as given in the original paper (Zhang et al., 2024), and the implementation from the the official code `https://github.com/amazon-science/tabsyn/`. Similar to CTGAN we cap epochs such that TabSyn's diffusion model trains for at most 30 000 training steps.

## G. Evaluation metrics

### G.1. Shape and Trend

We evaluate synthetic data fidelity using the Shape and Trend scores from SDMetrics[5]. Shape provides a measure for fidelity of marginal distributions, Trend for bivariate correlations. Definitions are given below.

**Shape.** Take a tabular dataset $\mathcal{X} = \{x_1, \ldots, x_d\}$ containing both numerical and categorical features. For each feature $x_j$, let $P_r(x_j)$ and $P_s(x_j)$ denote their empirical marginal distributions in the real and synthetic datasets, respectively.

For numerical features, we compute the Kolmogorov–Smirnov Complement (KSC):

$$
\begin{aligned}
\text{KSC}(x_j) &= 1 - D_{KS}\big(P_r(x_j), P_s(x_j)\big), \\
D_{KS}(P_r, P_s) &= \sup_x \big|F_r(x) - F_s(x)\big|,
\end{aligned}
\tag{11}
$$

where $F_r, F_s$ are the empirical cumulative distribution functions (CDFs) of $P_r$ and $P_s$.

For categorical features, we compute the Total Variation Complement (TVC):

$$
\begin{aligned}
\text{TVC}(x_j) &= 1 - D_{TV}\big(P_r(x_j), P_s(x_j)\big), \\
D_{TV}(P_r, P_s) &= \frac{1}{2} \sum_{c \in \mathcal{C}_j} \big|P_r(c) - P_s(c)\big|.
\end{aligned}
\tag{12}
$$

The overall Shape score is the average of KSC and TVC scores over all features:

$$
\text{Shape} = \frac{1}{d} \sum_{j=1}^{d} s(x_j), \qquad s(x_j) = \begin{cases} \text{KSC}(x_j), & x_j \text{ numerical}, \\ \text{TVC}(x_j), & x_j \text{ categorical}. \end{cases}
\tag{13}
$$

**Trend.** The overall Trend score is the average of all bivariate feature correlations:

$$
\text{Trend} = \frac{2}{d(d-1)} \sum_{i<j} t(x_i, x_j), \qquad t(x_i, x_j) = \begin{cases} 1 - \dfrac{|\rho_r(x_i, x_j) - \rho_s(x_i, x_j)|}{2}, & x_i, x_j \text{ both numerical}, \\ 1 - D_{TV}\big(P_r(x_i, x_j), P_s(x_i, x_j)\big), & x_i \text{ or } x_j \text{ categorical}. \end{cases}
\tag{14}
$$

Here, $\rho_r$ and $\rho_s$ are Pearson correlations in the real and synthetic data, and $D_{TV}$ is the Total Variation Distance defined in Equation (12), but applied to joint distributions. For numerical–categorical pairs, the numerical feature is first discretized into 10 equal-width bins before applying the same contingency similarity.

---

[5]`https://docs.sdv.dev/sdmetrics/`

### G.2. Detection

Another fidelity metric, which directly compares the joint distribution of synthetic and real data, is computed by using a binary classifier to distinguish between synthetic and real data. The classifier essentially compresses $P_r(X)$ and $P_s(X)$ to a single dimension to estimate a multivariate two-sample test (Friedman, 2004).

We use an XGBoost classifier for which we first tune hyperparameters based on the ROCAUC on an independent validation set. We search the hyperparameter space below for 32 trials using tree-structured Parzen estimators:

- Number of estimators: $n_{\text{estimators}} \in [30, 150]$

- Maximum tree depth: $\text{max\_depth} \in [3, 10]$

- Minimum child weight: $\text{min\_child\_weight} \sim \text{LogUniform}(1.0, 20.0)$

- Gamma: $\gamma \in [0.0, 5.0]$

After tuning hyperparameters, we train the classifier on the training set and evaluate ROCAUC on the test set. Lower scores indicate worse ability in discriminating $P_r(X)$ and $P_s(X)$ and thus better fidelity.

### G.3. Distributional Precision and Recall

We use the $\alpha$-Precision and $\beta$-Recall metric to further disentangle synthetic data quality (Alaa et al., 2022). Here, $\alpha$-Precision indicates coverage of the synthetic by real distribution, and $\beta$-Recall indicates coverage of the real by synthetic distribution. Intuitively, $\alpha$-Precision indicates fidelity or realism, whereas $\beta$-Recall indicates diversity or whether synthetic data captures the full range of the real distribution.

Both metrics rely on estimating distances between synthetic and real points, which is not straightforward in mixed-type tabular datasets. We rely on Gower-type distances (Gower, 1971) for the computation of these metrics, using the trick described in Jolicoeur-Martineau et al. (2024). That is, we achieve Gower distances by min-max scaling numerical features to $[0, 1]$, one-hot encoding categoricals in $\{0, \frac{1}{2}\}$, and relying on L1 distances. This is more suitable than L2 distances between naively one hot encoded features in $\{0, 1\}$, as this would overemphasize distances between categorical features.

### G.4. Machine Learning Efficacy

To assess synthetic data utility, we evaluate its efficacy for training machine learning prediction models. That is, we adopt the Train Synthetic Test Real approach formalized in Esteban et al. (2017). We also report the performance of a model trained on the real training set (Train Real Test Real).

We use XGBoost classifiers/regressors, for which we first tune hyperparameters based on performance when trained on real data and evaluated on an independent validation set. We use the same tuning set-up as described in Appendix G.2.

### G.5. DCR

We use the DCR metric as a proxy for synthetic data privacy. DCR computes distances from the synthetic data to the real training set and the independent test set. Intuitively, when the synthetic data are closer to the training set more often than to the test set, this can be a sign of overfitting. We use the Gower distance here, see Appendix G.3 for a discussion on why. We can compute the DCR score as follows:

$$\text{DCR} = \min\left(1, 2 \cdot (1 - |D_{s,tr}|\right), \tag{15}$$

where $|D_{s,tr}|$ indicates the proportion of synthetic points which are closer to the training set than to the test set. The aggregation of this score is inspired by SDMetrics's DCR score[6]. It ensures that higher scores indicate less privacy risk, and that we do not provide a false sense of privacy improvement. That is, anytime two synthetic sets achieve $|D_{s,tr}| \leq 0.5$, we cannot necessarily say that one is more privacy-preserving than the other, as neither exhibit strong signs of overfitting. Rather, lower scores can be the result of worse fidelity.

---

[6] https://docs.sdv.dev/sdmetrics/data-metrics/privacy/dcroverfittingprotection

We subsample the training set and synthetic set to the same size of the test set to avoid sampling bias in the DCR score. When the test set is smaller than the training set, we estimate the average score across multiple iterations to ensure the score is based on (a bootstrapped version of) the entire training set.

**DCR as a privacy measure**  Note that DCR is merely a proxy for privacy-preservation in synthetic data. Rigorous evaluation of reidentification risks would involve (at least) attribute and membership disclosure attacks. However, these require elaborate assumptions on which quasi-identifiers and sensitive variables are available to attackers, their resources, and so on. For the public datasets considered here, we do not consider this a relevant endeavor. These attacks are also influenced to a large extent by the dataset itself, e.g., its size and variables included, instead of purely by whether the generative model is appropriate and trained well. Therefore, such evaluations on the datasets included in our benchmarks do not provide any guarantee on privacy risk when generating synthetic data in a different context. For these reasons, we choose to use DCR as a *general* measure for overfitting to the training data. Finally, using DCR as a proxy for privacy follows in the footsteps of other similar works (Zhang et al., 2024; Mueller et al., 2025; Shi et al., 2025).

# H. Hardware

GPU-based baselines (CTGAN, TVAE, TabDDPM, TabSyn) were trained using a single NVIDIA TITAN Xp GPU (12GB RAM), and an additional 8 CPU cores (minimum 32GB RAM). CPU-based baselines (ARF, FD, FF, UT, SMOTE) and XGenBoost were trained using 16 CPU cores (minimum 64GB RAM).

# I. Raw Results

Table 7 through Table 30 provide the raw results of the main experiments. Each table shows the results for a single metric for all datasets. Results show the average and standard deviation over 20 sampling seeds ($\mu \pm \sigma$) - except for training time, which is a single value.

## I.1. Small Benchmark

*Table 7.* Small Benchmark: Shape metric scores for all datasets. Average and standard devation ($\mu \pm \sigma$) over 20 sampling seeds. Best results per dataset are in **bold**, second best underlined.

| model | SMOTE | ARF | UT | FD | FF | CTGAN | TVAE | TabDDPM | XGenB-AR | XGenB-DF (x-DDPM) | XGenB-DF (v-DDPM) | XGenB-DF (x-DDIM) | XGenB-DF (v-DDIM) |
|---|---|---|---|---|---|---|---|---|---|---|---|---|---|
| airfoil self noise | $0.927_{\pm0.003}$ | $0.911_{\pm0.003}$ | $0.949_{\pm0.007}$ | $0.908_{\pm0.005}$ | $0.951_{\pm0.004}$ | $0.714_{\pm0.004}$ | $0.798_{\pm0.004}$ | $0.957_{\pm0.006}$ | $\mathbf{0.974}_{\pm0.004}$ | $0.967_{\pm0.005}$ | $0.968_{\pm0.005}$ | $0.967_{\pm0.005}$ | $\underline{0.972}_{\pm0.004}$ |
| bean | $0.980_{\pm0.001}$ | $0.984_{\pm0.002}$ | $0.979_{\pm0.001}$ | $0.975_{\pm0.001}$ | $0.986_{\pm0.002}$ | $0.864_{\pm0.002}$ | $0.923_{\pm0.002}$ | $\mathbf{0.989}_{\pm0.002}$ | $0.987_{\pm0.002}$ | $0.987_{\pm0.002}$ | $\underline{0.988}_{\pm0.002}$ | $0.969_{\pm0.002}$ | $0.969_{\pm0.002}$ |
| blood transfusion | $0.957_{\pm0.006}$ | $0.949_{\pm0.006}$ | $0.954_{\pm0.010}$ | $0.926_{\pm0.010}$ | $0.942_{\pm0.007}$ | $0.791_{\pm0.005}$ | $0.824_{\pm0.008}$ | $0.955_{\pm0.007}$ | $\mathbf{0.968}_{\pm0.007}$ | $0.961_{\pm0.010}$ | $0.963_{\pm0.008}$ | $0.965_{\pm0.010}$ | $\underline{0.967}_{\pm0.010}$ |
| breast cancer diagnostic | $0.936_{\pm0.004}$ | $0.937_{\pm0.005}$ | $0.928_{\pm0.010}$ | $0.923_{\pm0.009}$ | $0.935_{\pm0.007}$ | $0.740_{\pm0.003}$ | $0.853_{\pm0.006}$ | $0.343_{\pm0.007}$ | $\mathbf{0.950}_{\pm0.007}$ | $0.945_{\pm0.007}$ | $\underline{0.947}_{\pm0.007}$ | $0.945_{\pm0.006}$ | $0.943_{\pm0.005}$ |
| california | $0.955_{\pm0.001}$ | $0.962_{\pm0.001}$ | $0.981_{\pm0.001}$ | $0.877_{\pm0.001}$ | $0.950_{\pm0.001}$ | $0.925_{\pm0.001}$ | $0.927_{\pm0.001}$ | $0.977_{\pm0.001}$ | $\mathbf{0.991}_{\pm0.001}$ | $0.981_{\pm0.001}$ | $0.982_{\pm0.001}$ | $0.985_{\pm0.002}$ | $\underline{0.988}_{\pm0.001}$ |
| car | $0.979_{\pm0.002}$ | $\underline{0.982}_{\pm0.003}$ | $0.979_{\pm0.005}$ | $0.981_{\pm0.003}$ | $0.979_{\pm0.005}$ | $0.895_{\pm0.004}$ | $0.906_{\pm0.007}$ | $0.980_{\pm0.004}$ | $\mathbf{0.983}_{\pm0.003}$ | $0.981_{\pm0.004}$ | $0.981_{\pm0.003}$ | $0.981_{\pm0.004}$ | $0.981_{\pm0.003}$ |
| climate model crashes | $0.928_{\pm0.002}$ | $0.928_{\pm0.004}$ | $\underline{0.949}_{\pm0.003}$ | $0.915_{\pm0.002}$ | $0.925_{\pm0.003}$ | $0.784_{\pm0.005}$ | $0.801_{\pm0.005}$ | $0.914_{\pm0.008}$ | $\mathbf{0.952}_{\pm0.004}$ | $0.897_{\pm0.004}$ | $0.919_{\pm0.004}$ | $0.909_{\pm0.003}$ | $0.932_{\pm0.003}$ |
| concrete compression | $0.955_{\pm0.003}$ | $0.892_{\pm0.004}$ | $0.932_{\pm0.005}$ | $0.864_{\pm0.005}$ | $0.883_{\pm0.006}$ | $0.702_{\pm0.004}$ | $0.809_{\pm0.005}$ | $0.951_{\pm0.005}$ | $\mathbf{0.962}_{\pm0.005}$ | $0.956_{\pm0.006}$ | $0.955_{\pm0.005}$ | $0.957_{\pm0.005}$ | $\underline{0.958}_{\pm0.005}$ |
| concrete slump | $0.876_{\pm0.011}$ | $0.845_{\pm0.014}$ | $0.878_{\pm0.021}$ | $0.871_{\pm0.010}$ | $0.881_{\pm0.015}$ | $0.761_{\pm0.016}$ | $0.826_{\pm0.014}$ | $0.446_{\pm0.019}$ | $0.886_{\pm0.017}$ | $0.886_{\pm0.018}$ | $0.889_{\pm0.014}$ | $\underline{0.891}_{\pm0.015}$ | $\mathbf{0.892}_{\pm0.014}$ |
| congress | $0.947_{\pm0.004}$ | $\underline{0.972}_{\pm0.007}$ | $\mathbf{0.973}_{\pm0.006}$ | $0.872_{\pm0.004}$ | $0.871_{\pm0.008}$ | $0.878_{\pm0.008}$ | $0.865_{\pm0.014}$ | $0.937_{\pm0.013}$ | $0.971_{\pm0.007}$ | $0.960_{\pm0.014}$ | $0.959_{\pm0.012}$ | $0.959_{\pm0.013}$ | $0.957_{\pm0.013}$ |
| connectionist bench sonar | $\underline{0.905}_{\pm0.003}$ | $0.883_{\pm0.005}$ | $0.865_{\pm0.011}$ | $0.878_{\pm0.007}$ | $0.878_{\pm0.004}$ | $0.689_{\pm0.003}$ | $0.836_{\pm0.004}$ | $0.412_{\pm0.005}$ | $\mathbf{0.918}_{\pm0.007}$ | $0.893_{\pm0.006}$ | $0.880_{\pm0.005}$ | $0.898_{\pm0.005}$ | $0.891_{\pm0.004}$ |
| connectionist bench vowel | $\underline{0.961}_{\pm0.002}$ | $0.955_{\pm0.003}$ | $0.951_{\pm0.004}$ | $0.952_{\pm0.004}$ | $0.955_{\pm0.003}$ | $0.799_{\pm0.003}$ | $0.889_{\pm0.006}$ | $0.958_{\pm0.003}$ | $\mathbf{0.962}_{\pm0.003}$ | $0.960_{\pm0.004}$ | $0.956_{\pm0.005}$ | $0.961_{\pm0.003}$ | $0.956_{\pm0.004}$ |
| ecoli | - | $0.932_{\pm0.009}$ | $0.949_{\pm0.010}$ | $0.925_{\pm0.008}$ | $0.954_{\pm0.007}$ | $0.770_{\pm0.009}$ | $0.877_{\pm0.010}$ | $0.950_{\pm0.007}$ | $0.953_{\pm0.008}$ | $0.953_{\pm0.008}$ | $\mathbf{0.957}_{\pm0.009}$ | $\underline{0.955}_{\pm0.006}$ | $\underline{0.955}_{\pm0.006}$ |
| glass | $0.919_{\pm0.006}$ | $0.855_{\pm0.014}$ | $0.917_{\pm0.011}$ | $0.867_{\pm0.013}$ | $0.925_{\pm0.009}$ | $0.620_{\pm0.014}$ | $0.799_{\pm0.015}$ | $0.362_{\pm0.008}$ | $\mathbf{0.935}_{\pm0.008}$ | $0.927_{\pm0.012}$ | $\underline{0.930}_{\pm0.009}$ | $0.926_{\pm0.015}$ | $0.929_{\pm0.011}$ |
| ionosphere | $0.880_{\pm0.003}$ | $0.875_{\pm0.004}$ | $0.850_{\pm0.004}$ | $0.872_{\pm0.005}$ | $0.873_{\pm0.003}$ | $0.702_{\pm0.004}$ | $0.865_{\pm0.005}$ | $0.546_{\pm0.004}$ | $\mathbf{0.938}_{\pm0.006}$ | $0.904_{\pm0.012}$ | $0.907_{\pm0.010}$ | $0.908_{\pm0.011}$ | $\underline{0.926}_{\pm0.006}$ |
| iris | $0.933_{\pm0.010}$ | $0.916_{\pm0.016}$ | $0.921_{\pm0.020}$ | $0.923_{\pm0.019}$ | $0.928_{\pm0.021}$ | $0.797_{\pm0.022}$ | $0.848_{\pm0.022}$ | $0.922_{\pm0.019}$ | $0.928_{\pm0.017}$ | $0.926_{\pm0.023}$ | $0.927_{\pm0.021}$ | $\underline{0.934}_{\pm0.014}$ | $\mathbf{0.935}_{\pm0.012}$ |
| libras | $\mathbf{0.935}_{\pm0.003}$ | $0.925_{\pm0.003}$ | $0.918_{\pm0.006}$ | $0.899_{\pm0.007}$ | $0.898_{\pm0.007}$ | $0.780_{\pm0.004}$ | $0.858_{\pm0.005}$ | $0.463_{\pm0.004}$ | $\underline{0.919}_{\pm0.017}$ | $\underline{0.919}_{\pm0.013}$ | $\mathbf{0.920}_{\pm0.011}$ | $\underline{0.919}_{\pm0.015}$ | $0.907_{\pm0.009}$ |
| parkinsons | $0.902_{\pm0.014}$ | $0.899_{\pm0.011}$ | $0.906_{\pm0.010}$ | $0.898_{\pm0.017}$ | $0.910_{\pm0.012}$ | $0.659_{\pm0.007}$ | $0.800_{\pm0.010}$ | $0.380_{\pm0.008}$ | $0.919_{\pm0.017}$ | $0.919_{\pm0.013}$ | $\mathbf{0.920}_{\pm0.011}$ | $\underline{0.919}_{\pm0.015}$ | $0.915_{\pm0.016}$ |
| planning relax | $0.903_{\pm0.006}$ | $0.902_{\pm0.008}$ | $0.908_{\pm0.009}$ | $0.905_{\pm0.008}$ | $0.914_{\pm0.007}$ | $0.622_{\pm0.012}$ | $0.863_{\pm0.006}$ | $0.455_{\pm0.011}$ | $0.916_{\pm0.010}$ | $0.916_{\pm0.011}$ | $0.911_{\pm0.011}$ | $\mathbf{0.919}_{\pm0.008}$ | $\underline{0.918}_{\pm0.009}$ |
| qsar biodegradation | $0.962_{\pm0.002}$ | $0.933_{\pm0.004}$ | $0.950_{\pm0.003}$ | $0.912_{\pm0.004}$ | $0.945_{\pm0.003}$ | $0.817_{\pm0.002}$ | $0.864_{\pm0.003}$ | $0.503_{\pm0.003}$ | $\mathbf{0.974}_{\pm0.002}$ | $\underline{0.965}_{\pm0.003}$ | $0.964_{\pm0.003}$ | $0.962_{\pm0.003}$ | $0.963_{\pm0.003}$ |
| seeds | $\mathbf{0.936}_{\pm0.008}$ | $0.921_{\pm0.010}$ | $0.921_{\pm0.002}$ | $0.925_{\pm0.018}$ | $0.927_{\pm0.015}$ | $0.746_{\pm0.015}$ | $0.853_{\pm0.011}$ | $0.891_{\pm0.017}$ | $\underline{0.931}_{\pm0.013}$ | $0.920_{\pm0.016}$ | $0.923_{\pm0.016}$ | $0.927_{\pm0.015}$ | $0.928_{\pm0.013}$ |
| tictactoe | $0.962_{\pm0.003}$ | $\underline{0.978}_{\pm0.003}$ | $\underline{0.978}_{\pm0.004}$ | $0.975_{\pm0.003}$ | $0.976_{\pm0.005}$ | $0.954_{\pm0.007}$ | $0.873_{\pm0.004}$ | $\mathbf{0.979}_{\pm0.006}$ | $\underline{0.978}_{\pm0.003}$ | $0.976_{\pm0.005}$ | $\underline{0.978}_{\pm0.005}$ | $\underline{0.978}_{\pm0.005}$ | $\underline{0.978}_{\pm0.004}$ |
| wine | $0.913_{\pm0.007}$ | $0.897_{\pm0.016}$ | $0.906_{\pm0.012}$ | $0.913_{\pm0.007}$ | $0.916_{\pm0.013}$ | $0.761_{\pm0.015}$ | $0.857_{\pm0.013}$ | $0.381_{\pm0.016}$ | $0.918_{\pm0.007}$ | $0.914_{\pm0.014}$ | $0.916_{\pm0.013}$ | $\underline{0.920}_{\pm0.011}$ | $\mathbf{0.922}_{\pm0.010}$ |
| wine quality red | $0.942_{\pm0.001}$ | $0.938_{\pm0.006}$ | $0.943_{\pm0.004}$ | $0.938_{\pm0.005}$ | $0.942_{\pm0.003}$ | $0.814_{\pm0.004}$ | $0.852_{\pm0.004}$ | $0.948_{\pm0.003}$ | $\mathbf{0.973}_{\pm0.002}$ | $0.958_{\pm0.003}$ | $0.962_{\pm0.003}$ | $0.960_{\pm0.003}$ | $\underline{0.968}_{\pm0.003}$ |
| wine quality white | $0.956_{\pm0.001}$ | $0.959_{\pm0.002}$ | $0.962_{\pm0.002}$ | $0.957_{\pm0.002}$ | $0.959_{\pm0.001}$ | $0.784_{\pm0.002}$ | $0.871_{\pm0.003}$ | $0.937_{\pm0.004}$ | $\mathbf{0.983}_{\pm0.002}$ | $0.970_{\pm0.002}$ | $0.974_{\pm0.002}$ | $0.973_{\pm0.002}$ | $\underline{0.979}_{\pm0.001}$ |
| yacht hydrodynamics | $0.889_{\pm0.007}$ | $0.868_{\pm0.007}$ | $0.908_{\pm0.012}$ | $0.866_{\pm0.011}$ | $0.943_{\pm0.010}$ | $0.626_{\pm0.014}$ | $0.792_{\pm0.006}$ | $0.889_{\pm0.014}$ | $\mathbf{0.947}_{\pm0.009}$ | $\underline{0.945}_{\pm0.012}$ | $\underline{0.945}_{\pm0.011}$ | $\mathbf{0.947}_{\pm0.010}$ | $\underline{0.945}_{\pm0.009}$ |
| yeast | $0.968_{\pm0.002}$ | $0.942_{\pm0.004}$ | $0.971_{\pm0.003}$ | $0.923_{\pm0.004}$ | $0.966_{\pm0.003}$ | $0.786_{\pm0.003}$ | $0.834_{\pm0.005}$ | $\underline{0.974}_{\pm0.003}$ | $\mathbf{0.978}_{\pm0.003}$ | $0.970_{\pm0.004}$ | $0.973_{\pm0.004}$ | $0.971_{\pm0.004}$ | $\underline{0.974}_{\pm0.003}$ |

## I.2. Big Benchmark

*Table 8.* Small Benchmark: Trend metric scores for all datasets. Average and standard devation ($\mu \pm \sigma$) over 20 sampling seeds. Best results per dataset are in **bold**, second best underlined.

| model | SMOTE | ARF | UT | FD | FF | CTGAN | TVAE | TabDDPM | XGenB-AR | XGenB-DF (x-DDPM) | XGenB-DF (v-DDPM) | XGenB-DF (x-DDIM) | XGenB-DF (v-DDIM) |
|---|---|---|---|---|---|---|---|---|---|---|---|---|---|
| airfoil self noise | 0.985±0.003 | 0.983±0.003 | 0.976±0.003 | 0.987±0.004 | 0.988±0.002 | 0.881±0.004 | 0.962±0.005 | 0.988±0.002 | 0.986±0.003 | 0.988±0.003 | 0.989±0.004 | 0.988±0.004 | 0.987±0.004 |
| bean | 0.982±0.002 | 0.924±0.005 | 0.983±0.002 | 0.986±0.002 | 0.990±0.002 | 0.917±0.001 | 0.917±0.002 | 0.983±0.002 | 0.984±0.002 | 0.989±0.002 | 0.989±0.002 | 0.981±0.002 | 0.981±0.002 |
| blood transfusion | 0.938±0.005 | 0.870±0.039 | 0.943±0.018 | 0.965±0.005 | 0.963±0.016 | 0.668±0.020 | 0.742±0.014 | 0.962±0.010 | 0.955±0.008 | 0.961±0.012 | 0.960±0.014 | 0.962±0.013 | 0.961±0.014 |
| breast cancer diagnostic | 0.964±0.004 | 0.910±0.004 | 0.956±0.004 | 0.956±0.006 | 0.960±0.006 | 0.787±0.002 | 0.891±0.005 | 0.759±0.002 | 0.964±0.003 | 0.954±0.006 | 0.955±0.006 | 0.969±0.004 | 0.971±0.004 |
| california | 0.993±0.001 | 0.988±0.001 | 0.988±0.004 | 0.987±0.001 | 0.994±0.001 | 0.941±0.001 | 0.962±0.001 | 0.984±0.003 | 0.982±0.003 | 0.988±0.006 | 0.989±0.005 | 0.988±0.004 | 0.993±0.001 |
| car | 0.950±0.002 | 0.960±0.003 | 0.959±0.005 | 0.959±0.004 | 0.959±0.004 | 0.842±0.006 | 0.823±0.006 | 0.938±0.004 | 0.962±0.003 | 0.960±0.004 | 0.960±0.004 | 0.961±0.004 | 0.961±0.003 |
| climate model crashes | 0.965±0.001 | 0.939±0.002 | 0.969±0.001 | 0.964±0.002 | 0.966±0.001 | 0.936±0.002 | 0.914±0.002 | 0.920±0.009 | 0.970±0.002 | 0.959±0.002 | 0.966±0.002 | 0.963±0.001 | 0.968±0.001 |
| concrete compression | 0.983±0.002 | 0.970±0.002 | 0.968±0.004 | 0.986±0.004 | 0.983±0.005 | 0.895±0.003 | 0.952±0.004 | 0.951±0.013 | 0.981±0.004 | 0.983±0.004 | 0.984±0.003 | 0.983±0.003 | 0.983±0.003 |
| concrete slump | 0.955±0.015 | 0.921±0.007 | 0.944±0.013 | 0.954±0.011 | 0.951±0.008 | 0.870±0.013 | 0.919±0.006 | 0.830±0.006 | 0.944±0.009 | 0.952±0.010 | 0.956±0.009 | 0.949±0.011 | 0.950±0.009 |
| congress | 0.911±0.004 | 0.936±0.009 | 0.950±0.007 | 0.772±0.005 | 0.771±0.009 | 0.730±0.005 | 0.785±0.015 | 0.905±0.013 | 0.947±0.007 | 0.932±0.014 | 0.929±0.011 | 0.930±0.012 | 0.927±0.013 |
| connectionist bench sonar | 0.954±0.004 | 0.904±0.003 | 0.949±0.003 | 0.950±0.005 | 0.954±0.002 | 0.872±0.001 | 0.919±0.003 | 0.857±0.001 | 0.950±0.003 | 0.950±0.003 | 0.952±0.003 | 0.955±0.002 | 0.953±0.002 |
| connectionist bench vowel | 0.950±0.002 | 0.919±0.004 | 0.948±0.005 | 0.953±0.004 | 0.954±0.002 | 0.847±0.002 | 0.781±0.005 | 0.825±0.008 | 0.949±0.003 | 0.956±0.004 | 0.955±0.004 | 0.957±0.003 | 0.957±0.004 |
| ecoli | - | 0.855±0.009 | 0.927±0.010 | 0.930±0.007 | 0.929±0.008 | 0.777±0.010 | 0.780±0.012 | 0.912±0.011 | 0.920±0.008 | 0.932±0.009 | 0.931±0.012 | 0.933±0.006 | 0.933±0.007 |
| glass | 0.838±0.015 | 0.826±0.010 | 0.880±0.019 | 0.902±0.017 | 0.906±0.018 | 0.770±0.009 | 0.810±0.010 | 0.658±0.010 | 0.885±0.010 | 0.910±0.020 | 0.906±0.020 | 0.904±0.016 | 0.899±0.022 |
| ionosphere | 0.927±0.004 | 0.893±0.005 | 0.915±0.005 | 0.939±0.004 | 0.941±0.004 | 0.843±0.002 | 0.915±0.003 | 0.798±0.002 | 0.951±0.003 | 0.946±0.003 | 0.946±0.004 | 0.948±0.004 | 0.953±0.003 |
| iris | 0.918±0.011 | 0.837±0.029 | 0.904±0.011 | 0.910±0.014 | 0.910±0.018 | 0.648±0.018 | 0.781±0.019 | 0.893±0.017 | 0.891±0.015 | 0.911±0.017 | 0.907±0.014 | 0.908±0.025 | 0.907±0.019 |
| libras | 0.965±0.004 | 0.919±0.002 | 0.956±0.004 | 0.932±0.005 | 0.938±0.005 | 0.856±0.001 | 0.930±0.003 | 0.848±0.000 | 0.942±0.006 | 0.957±0.005 | 0.947±0.005 | 0.956±0.004 | 0.945±0.003 |
| parkinsons | 0.962±0.004 | 0.877±0.010 | 0.946±0.007 | 0.947±0.012 | 0.952±0.009 | 0.734±0.004 | 0.930±0.003 | 0.739±0.004 | 0.933±0.009 | 0.950±0.009 | 0.954±0.009 | 0.957±0.008 | 0.957±0.007 |
| planning relax | 0.937±0.008 | 0.900±0.007 | 0.927±0.006 | 0.935±0.006 | 0.937±0.007 | 0.845±0.005 | 0.876±0.006 | 0.788±0.004 | 0.932±0.006 | 0.940±0.007 | 0.938±0.008 | 0.938±0.005 | 0.936±0.006 |
| qsar biodegradation | 0.956±0.003 | 0.850±0.005 | 0.948±0.005 | 0.949±0.004 | 0.952±0.005 | 0.847±0.007 | 0.836±0.008 | 0.788±0.003 | 0.953±0.004 | 0.953±0.004 | 0.948±0.006 | 0.953±0.004 | 0.951±0.004 |
| seeds | 0.945±0.006 | 0.889±0.014 | 0.938±0.009 | 0.945±0.008 | 0.945±0.007 | 0.670±0.011 | 0.811±0.012 | 0.880±0.016 | 0.935±0.007 | 0.943±0.010 | 0.943±0.008 | 0.946±0.009 | 0.944±0.009 |
| tictactoe | 0.921±0.003 | 0.949±0.003 | 0.956±0.005 | 0.950±0.004 | 0.945±0.005 | 0.901±0.006 | 0.782±0.007 | 0.957±0.006 | 0.955±0.003 | 0.955±0.005 | 0.956±0.005 | 0.956±0.004 | 0.956±0.004 |
| wine | 0.937±0.004 | 0.892±0.009 | 0.933±0.007 | 0.934±0.009 | 0.932±0.007 | 0.805±0.006 | 0.894±0.006 | 0.681±0.005 | 0.931±0.006 | 0.933±0.008 | 0.935±0.010 | 0.937±0.008 | 0.939±0.007 |
| wine quality red | 0.984±0.001 | 0.970±0.002 | 0.983±0.002 | 0.985±0.002 | 0.985±0.002 | 0.905±0.002 | 0.958±0.002 | 0.890±0.005 | 0.981±0.002 | 0.982±0.002 | 0.984±0.002 | 0.984±0.002 | 0.986±0.001 |
| wine quality white | 0.990±0.001 | 0.981±0.001 | 0.990±0.001 | 0.992±0.001 | 0.991±0.001 | 0.915±0.001 | 0.942±0.001 | 0.855±0.005 | 0.990±0.001 | 0.984±0.001 | 0.987±0.001 | 0.989±0.001 | 0.991±0.002 |
| yacht hydrodynamics | 0.974±0.004 | 0.959±0.006 | 0.963±0.007 | 0.976±0.005 | 0.973±0.008 | 0.914±0.006 | 0.941±0.006 | 0.937±0.013 | 0.971±0.005 | 0.972±0.005 | 0.975±0.005 | 0.973±0.007 | 0.976±0.007 |
| yeast | 0.960±0.005 | 0.878±0.011 | 0.960±0.007 | 0.957±0.013 | 0.956±0.017 | 0.837±0.013 | 0.757±0.012 | 0.937±0.009 | 0.954±0.010 | 0.957±0.017 | 0.957±0.018 | 0.960±0.009 | 0.960±0.011 |

*Table 9.* Small Benchmark: Detection metric scores for all datasets. Average and standard deviation ($\mu \pm \sigma$) over 20 sampling seeds. Best results per dataset are in **bold**, second best underlined.

| model | SMOTE | ARF | UT | FD | FF | CTGAN | TVAE | TabDDPM | XGenB-AR | XGenB-DF (x-DDPM) | XGenB-DF (v-DDPM) | XGenB-DF (x-DDIM) | XGenB-DF (v-DDIM) |
|---|---|---|---|---|---|---|---|---|---|---|---|---|---|
| airfoil self noise | 0.985±0.004 | 0.996±0.004 | 0.951±0.012 | 0.991±0.004 | 0.982±0.006 | 1.000±0.000 | 0.929±0.010 | 0.712±0.022 | 0.652±0.027 | 0.582±0.024 | 0.543±0.009 | 0.638±0.022 | 0.578±0.034 |
| bean | 0.682±0.007 | 0.970±0.006 | 0.933±0.008 | 0.990±0.002 | 0.625±0.014 | 1.000±0.000 | 1.000±0.000 | 0.511±0.007 | 0.898±0.014 | 0.555±0.008 | 0.543±0.009 | 0.595±0.005 | 0.604±0.008 |
| blood transfusion | 0.651±0.030 | 0.877±0.014 | 0.768±0.022 | 0.964±0.009 | 0.953±0.013 | 0.996±0.002 | 0.996±0.002 | 0.901±0.013 | 0.525±0.043 | 0.530±0.033 | 0.546±0.034 | 0.519±0.041 | 0.521±0.018 |
| breast cancer diagnostic | 0.598±0.025 | 0.929±0.025 | 0.614±0.025 | 0.704±0.042 | 0.613±0.032 | 1.000±0.000 | 0.952±0.016 | 1.000±0.000 | 0.565±0.037 | 0.658±0.030 | 0.590±0.038 | 0.615±0.036 | 0.560±0.029 |
| california | 0.805±0.005 | 0.875±0.004 | 0.721±0.005 | 0.976±0.001 | 0.816±0.006 | 0.969±0.001 | 0.935±0.002 | 1.000±0.000 | 0.706±0.008 | 0.677±0.006 | 0.641±0.007 | 0.665±0.009 | 0.630±0.008 |
| car | 0.800±0.012 | 0.501±0.016 | 0.506±0.021 | 0.490±0.033 | 0.488±0.022 | 0.785±0.016 | 0.872±0.008 | 0.696±0.022 | 0.501±0.003 | 0.501±0.030 | 0.505±0.021 | 0.516±0.026 | 0.499±0.008 |
| climate model crashes | 0.752±0.023 | 0.833±0.033 | 0.493±0.018 | 0.845±0.027 | 0.866±0.011 | 0.979±0.008 | 0.945±0.008 | 0.715±0.047 | 0.500±0.023 | 0.832±0.030 | 0.780±0.024 | 0.780±0.021 | 0.709±0.032 |
| concrete compression | 0.859±0.025 | 0.980±0.006 | 0.936±0.009 | 0.966±0.011 | 0.937±0.015 | 1.000±0.000 | 0.999±0.001 | 0.907±0.012 | 0.799±0.031 | 0.786±0.022 | 0.742±0.024 | 0.798±0.020 | 0.752±0.025 |
| concrete slump | 0.727±0.057 | 0.821±0.067 | 0.651±0.069 | 0.635±0.112 | 0.592±0.098 | 0.893±0.049 | 0.825±0.076 | 0.935±0.029 | 0.521±0.056 | 0.507±0.041 | 0.537±0.083 | 0.538±0.084 | 0.532±0.091 |
| congress | 0.768±0.026 | 0.537±0.039 | 0.498±0.043 | 0.995±0.006 | 0.992±0.004 | 0.941±0.004 | 0.935±0.012 | 0.553±0.037 | 0.494±0.051 | 0.556±0.046 | 0.560±0.042 | 0.556±0.043 | 0.552±0.032 |
| connectionist bench sonar | 0.698±0.045 | 0.848±0.056 | 0.823±0.050 | 0.820±0.041 | 0.873±0.034 | 0.997±0.003 | 0.956±0.020 | 1.000±0.000 | 0.564±0.055 | 0.843±0.038 | 0.870±0.047 | 0.801±0.042 | 0.840±0.035 |
| connectionist bench vowel | 0.583±0.016 | 0.823±0.019 | 0.685±0.026 | 0.577±0.028 | 0.551±0.030 | 0.987±0.005 | 0.918±0.012 | 0.541±0.030 | 0.684±0.021 | 0.574±0.027 | 0.579±0.025 | 0.571±0.031 | 0.577±0.034 |
| ecoli | - | 0.696±0.043 | 0.508±0.034 | 0.643±0.056 | 0.485±0.054 | 0.976±0.017 | 0.766±0.034 | 0.509±0.042 | 0.533±0.053 | 0.529±0.056 | 0.503±0.069 | 0.532±0.060 | 0.536±0.059 |
| glass | 0.687±0.039 | 0.928±0.017 | 0.592±0.059 | 0.754±0.043 | 0.507±0.036 | 0.989±0.006 | 0.965±0.014 | 0.995±0.007 | 0.521±0.069 | 0.565±0.067 | 0.506±0.053 | 0.553±0.056 | 0.500±0.042 |
| ionosphere | 0.696±0.029 | 0.949±0.018 | 0.914±0.027 | 0.755±0.028 | 0.802±0.027 | 0.998±0.002 | 0.975±0.011 | 0.559±0.044 | 0.498±0.045 | 0.676±0.045 | 0.632±0.040 | 0.607±0.043 | 0.611±0.055 |
| iris | 0.480±0.060 | 0.591±0.076 | 0.539±0.067 | 0.518±0.096 | 0.538±0.080 | 0.954±0.028 | 0.655±0.052 | 0.503±0.017 | 0.498±0.082 | 0.566±0.100 | 0.553±0.075 | 0.497±0.015 | 0.500±0.010 |
| libras | 0.568±0.030 | 0.978±0.012 | 0.769±0.036 | 0.787±0.029 | 0.786±0.050 | 1.000±0.001 | 0.986±0.008 | 1.000±0.000 | 0.820±0.022 | 0.708±0.050 | 0.722±0.034 | 0.667±0.049 | 0.803±0.039 |
| parkinsons | 0.577±0.065 | 0.820±0.047 | 0.544±0.054 | 0.646±0.054 | 0.558±0.059 | 0.997±0.007 | 0.951±0.022 | 1.000±0.000 | 0.508±0.065 | 0.618±0.055 | 0.554±0.066 | 0.551±0.067 | 0.561±0.074 |
| planning relax | 0.583±0.047 | 0.742±0.048 | 0.550±0.048 | 0.583±0.060 | 0.531±0.053 | 0.976±0.014 | 0.829±0.053 | 1.000±0.000 | 0.520±0.058 | 0.521±0.057 | 0.529±0.071 | 0.509±0.038 | 0.531±0.059 |
| qsar biodegradation | 0.797±0.021 | 0.981±0.006 | 0.890±0.015 | 0.988±0.005 | 0.952±0.010 | 1.000±0.000 | 0.999±0.001 | 1.000±0.000 | 0.720±0.021 | 0.767±0.021 | 0.749±0.021 | 0.791±0.021 | 0.766±0.020 |
| seeds | 0.592±0.050 | 0.621±0.057 | 0.622±0.060 | 0.461±0.060 | 0.546±0.074 | 0.962±0.017 | 0.822±0.049 | 0.585±0.060 | 0.492±0.058 | 0.534±0.054 | 0.552±0.060 | 0.518±0.058 | 0.540±0.053 |
| tictactoe | 0.852±0.019 | 0.583±0.031 | 0.502±0.026 | 0.521±0.036 | 0.504±0.025 | 0.829±0.020 | 0.921±0.010 | 0.510±0.024 | 0.494±0.058 | 0.506±0.034 | 0.484±0.028 | 0.500±0.005 | 0.502±0.006 |
| wine | 0.573±0.057 | 0.578±0.052 | 0.677±0.069 | 0.554±0.079 | 0.548±0.062 | 0.975±0.022 | 0.773±0.049 | 0.998±0.002 | 0.508±0.077 | 0.583±0.058 | 0.566±0.074 | 0.630±0.065 | 0.580±0.075 |
| wine quality red | 0.912±0.011 | 0.977±0.005 | 0.993±0.004 | 0.981±0.007 | 0.982±0.005 | 0.982±0.005 | 0.987±0.003 | 0.944±0.009 | 0.639±0.023 | 0.688±0.019 | 0.625±0.017 | 0.640±0.022 | 0.579±0.021 |
| wine quality white | 0.953±0.005 | 0.992±0.002 | 0.998±0.001 | 0.993±0.001 | 0.993±0.002 | 0.993±0.002 | 0.993±0.001 | 0.852±0.011 | 0.743±0.016 | 0.730±0.010 | 0.678±0.009 | 0.688±0.009 | 0.644±0.012 |
| yacht hydrodynamics | 0.933±0.024 | 0.972±0.014 | 0.748±0.048 | 0.950±0.021 | 0.515±0.056 | 0.999±0.005 | 0.997±0.002 | 0.660±0.062 | 0.507±0.062 | 0.513±0.046 | 0.522±0.049 | 0.541±0.043 |  |
| yeast | 0.659±0.020 | 0.782±0.013 | 0.560±0.023 | 0.789±0.018 | 0.586±0.022 | 0.972±0.004 | 0.915±0.006 | 0.571±0.027 | 0.515±0.024 | 0.595±0.024 | 0.559±0.024 | 0.575±0.029 | 0.533±0.021 |

*Table 10.* Small Benchmark: $\alpha$-Precision metric scores for all datasets. Average and standard deviation ($\mu \pm \sigma$) over 20 sampling seeds. Best results per dataset are in **bold**, second best underlined.

| model | SMOTE | ARF | UT | FD | FF | CTGAN | TVAE | TabDDPM | XGenB-AR | XGenB-DF (x-DDPM) | XGenB-DF (v-DDPM) | XGenB-DF (x-DDIM) | XGenB-DF (v-DDIM) |
|---|---|---|---|---|---|---|---|---|---|---|---|---|---|
| airfoil self noise | 0.891±0.009 | 0.966±0.016 | 0.972±0.007 | 0.958±0.019 | 0.953±0.020 | 0.692±0.017 | 0.933±0.008 | 0.971±0.013 | 0.980±0.007 | 0.979±0.009 | 0.955±0.015 | 0.973±0.013 | 0.975±0.010 |
| bean | 0.965±0.003 | 0.983±0.004 | 0.948±0.004 | 0.988±0.004 | 0.987±0.006 | 0.746±0.006 | 0.928±0.005 | 0.992±0.003 | 0.982±0.005 | 0.983±0.005 | 0.981±0.005 | 0.900±0.004 | 0.899±0.004 |
| blood transfusion | 0.944±0.014 | 0.884±0.022 | 0.955±0.020 | 0.955±0.012 | 0.965±0.016 | 0.611±0.028 | 0.671±0.015 | 0.000±0.000 | 0.949±0.025 | 0.965±0.011 | 0.956±0.021 | 0.963±0.016 | 0.960±0.016 |
| breast cancer diagnostic | 0.891±0.017 | 0.884±0.022 | 0.818±0.031 | 0.898±0.040 | 0.886±0.034 | 0.369±0.010 | 0.734±0.017 | 0.000±0.000 | 0.949±0.025 | 0.965±0.011 | 0.939±0.018 | 0.876±0.029 | 0.850±0.031 |
| california | 0.868±0.003 | 0.978±0.005 | 0.973±0.004 | 0.915±0.005 | 0.945±0.005 | 0.933±0.005 | 0.944±0.005 | 0.959±0.005 | 0.984±0.004 | 0.943±0.004 | 0.928±0.004 | 0.970±0.007 | 0.960±0.007 |
| car | 0.966±0.009 | 0.982±0.007 | 0.976±0.010 | 0.982±0.005 | 0.977±0.006 | 0.951±0.008 | 0.865±0.006 | 0.949±0.015 | 0.981±0.007 | 0.980±0.006 | 0.982±0.007 | 0.981±0.009 | 0.982±0.007 |
| climate model crashes | 0.548±0.017 | 0.855±0.036 | 0.968±0.010 | 0.436±0.029 | 0.571±0.026 | 0.328±0.020 | 0.818±0.022 | 0.692±0.046 | 0.962±0.009 | 0.955±0.029 | 0.450±0.030 | 0.373±0.025 | 0.564±0.022 |
| concrete compression | 0.942±0.010 | 0.977±0.008 | 0.886±0.021 | 0.866±0.024 | 0.874±0.024 | 0.529±0.022 | 0.913±0.011 | 0.956±0.019 | 0.961±0.017 | 0.915±0.024 | 0.875±0.020 | 0.926±0.020 | 0.905±0.021 |
| concrete slump | 0.880±0.034 | 0.873±0.037 | 0.909±0.042 | 0.807±0.036 | 0.870±0.036 | 0.684±0.043 | 0.692±0.046 | 0.063±0.036 | 0.894±0.039 | 0.898±0.048 | 0.817±0.060 | 0.915±0.035 | 0.866±0.044 |
| congress | 0.779±0.016 | 0.936±0.022 | 0.959±0.016 | 0.524±0.024 | 0.539±0.026 | 0.601±0.027 | 0.461±0.037 | 0.889±0.033 | 0.958±0.017 | 0.906±0.031 | 0.912±0.034 | 0.908±0.034 | 0.900±0.040 |
| connectionist bench sonar | 0.696±0.027 | 0.869±0.035 | 0.728±0.040 | 0.393±0.041 | 0.454±0.047 | 0.228±0.010 | 0.555±0.029 | 0.600±0.020 | 0.937±0.025 | 0.607±0.052 | 0.387±0.042 | 0.604±0.041 | 0.450±0.032 |
| connectionist bench vowel | 0.937±0.008 | 0.963±0.011 | 0.847±0.019 | 0.865±0.027 | 0.892±0.023 | 0.375±0.012 | 0.921±0.012 | 0.956±0.022 | 0.962±0.008 | 0.961±0.016 | 0.875±0.020 | 0.937±0.025 | 0.857±0.026 |
| ecoli | - | 0.945±0.020 | 0.945±0.030 | 0.940±0.030 | 0.952±0.020 | 0.635±0.046 | 0.698±0.031 | 0.940±0.027 | 0.960±0.025 | 0.960±0.019 | 0.954±0.012 | 0.939±0.028 | 0.912±0.039 |
| glass | 0.896±0.022 | 0.865±0.044 | 0.914±0.028 | 0.927±0.025 | 0.940±0.020 | 0.462±0.024 | 0.706±0.043 | 0.009±0.005 | 0.943±0.029 | 0.944±0.021 | 0.944±0.026 | 0.946±0.028 | 0.942±0.019 |
| ionosphere | 0.835±0.017 | 0.826±0.021 | 0.817±0.036 | 0.729±0.022 | 0.790±0.019 | 0.539±0.014 | 0.768±0.024 | 0.043±0.002 | 0.943±0.013 | 0.946±0.019 | 0.787±0.044 | 0.948±0.015 | 0.875±0.033 |
| iris | 0.952±0.018 | 0.929±0.026 | 0.925±0.023 | 0.938±0.028 | 0.934±0.024 | 0.687±0.037 | 0.771±0.043 | 0.920±0.027 | 0.929±0.026 | 0.938±0.033 | 0.944±0.023 | 0.948±0.021 | 0.939±0.026 |
| libras | 0.832±0.021 | 0.864±0.014 | 0.717±0.043 | 0.656±0.039 | 0.639±0.045 | 0.350±0.012 | 0.759±0.019 | 0.000±0.000 | 0.799±0.045 | 0.945±0.020 | 0.729±0.032 | 0.886±0.046 | 0.621±0.041 |
| parkinsons | 0.821±0.035 | 0.842±0.048 | 0.878±0.054 | 0.869±0.047 | 0.886±0.047 | 0.069±0.007 | 0.728±0.037 | 0.055±0.002 | 0.936±0.024 | 0.931±0.022 | 0.888±0.034 | 0.912±0.039 | 0.865±0.046 |
| planning relax | 0.759±0.026 | 0.877±0.030 | 0.850±0.045 | 0.767±0.037 | 0.833±0.057 | 0.171±0.020 | 0.610±0.026 | 0.000±0.000 | 0.921±0.027 | 0.855±0.047 | 0.773±0.050 | 0.895±0.036 | 0.856±0.049 |
| qsar biodegradation | 0.851±0.011 | 0.884±0.019 | 0.763±0.019 | 0.824±0.019 | 0.831±0.017 | 0.428±0.010 | 0.652±0.014 | 0.000±0.000 | 0.942±0.018 | 0.858±0.019 | 0.811±0.019 | 0.824±0.019 | 0.793±0.021 |
| seeds | 0.956±0.016 | 0.901±0.025 | 0.935±0.028 | 0.956±0.013 | 0.944±0.024 | 0.594±0.042 | 0.774±0.024 | 0.863±0.045 | 0.938±0.019 | 0.937±0.025 | 0.942±0.023 | 0.943±0.021 | 0.931±0.024 |
| tictactoe | 0.921±0.008 | 0.953±0.012 | 0.965±0.009 | 0.956±0.008 | 0.952±0.011 | 0.934±0.011 | 0.532±0.026 | 0.959±0.010 | 0.956±0.005 | 0.974±0.013 | 0.972±0.011 | 0.970±0.013 | 0.975±0.009 |
| wine | 0.794±0.032 | 0.940±0.024 | 0.876±0.054 | 0.848±0.057 | 0.844±0.059 | 0.316±0.037 | 0.753±0.052 | 0.000±0.000 | 0.926±0.027 | 0.934±0.037 | 0.833±0.051 | 0.944±0.025 | 0.850±0.047 |
| wine quality red | 0.806±0.008 | 0.897±0.013 | 0.899±0.016 | 0.920±0.021 | 0.877±0.020 | 0.584±0.015 | 0.869±0.010 | 0.934±0.018 | 0.976±0.009 | 0.865±0.022 | 0.842±0.020 | 0.907±0.018 | 0.900±0.018 |
| wine quality white | 0.853±0.005 | 0.951±0.010 | 0.922±0.009 | 0.932±0.010 | 0.872±0.011 | 0.603±0.006 | 0.962±0.010 | 0.976±0.010 | 0.976±0.010 | 0.859±0.008 | 0.857±0.010 | 0.904±0.008 | 0.907±0.008 |
| yacht hydrodynamics | 0.891±0.020 | 0.906±0.028 | 0.832±0.042 | 0.951±0.023 | 0.938±0.031 | 0.943±0.023 | 0.362±0.028 | 0.846±0.056 | 0.937±0.023 | 0.951±0.019 | 0.911±0.032 | 0.942±0.022 | 0.929±0.039 |
| yeast | 0.919±0.010 | 0.894±0.024 | 0.933±0.015 | 0.974±0.013 | 0.939±0.018 | 0.489±0.011 | 0.658±0.017 | 0.960±0.014 | 0.976±0.008 | 0.919±0.020 | 0.910±0.022 | 0.917±0.021 | 0.917±0.020 |

*Table 11.* Small Benchmark: $\beta$-Recall metric scores for all datasets. Average and standard deviation ($\mu \pm \sigma$) over 20 sampling seeds. Best results per dataset are in **bold**, second best underlined.

| model | SMOTE | ARF | UT | FD | FF | CTGAN | TVAE | TabDDPM | XGenB-AR | XGenB-DF (x-DDPM) | XGenB-DF (v-DDPM) | XGenB-DF (x-DDIM) | XGenB-DF (v-DDIM) |
|---|---|---|---|---|---|---|---|---|---|---|---|---|---|
| airfoil self noise | 0.299±0.012 | 0.067±0.009 | 0.416±0.018 | 0.143±0.012 | **0.606**±0.022 | 0.002±0.002 | 0.073±0.008 | 0.586±0.017 | 0.468±0.017 | 0.511±0.017 | 0.549±0.020 | 0.490±0.015 | 0.519±0.016 |
| bean | **0.668**±0.003 | 0.134±0.003 | 0.200±0.004 | 0.062±0.002 | 0.565±0.008 | 0.000±0.000 | 0.003±0.002 | 0.498±0.008 | 0.271±0.006 | 0.573±0.005 | 0.562±0.005 | 0.564±0.006 | 0.547±0.007 |
| blood transfusion | 0.771±0.013 | 0.436±0.028 | 0.733±0.035 | 0.336±0.025 | 0.610±0.036 | 0.084±0.022 | 0.140±0.019 | 0.522±0.028 | 0.692±0.025 | 0.764±0.029 | **0.779**±0.017 | 0.756±0.027 | 0.768±0.029 |
| breast cancer diagnostic | 0.821±0.013 | 0.051±0.011 | 0.577±0.029 | 0.542±0.024 | 0.828±0.022 | 0.000±0.000 | 0.023±0.008 | 0.000±0.000 | 0.583±0.021 | **0.835**±0.022 | 0.809±0.017 | 0.826±0.018 | 0.771±0.027 |
| california | **0.548**±0.004 | 0.252±0.005 | 0.420±0.004 | 0.103±0.003 | 0.509±0.001 | 0.164±0.003 | 0.273±0.003 | 0.395±0.005 | 0.380±0.004 | 0.434±0.005 | 0.445±0.006 | 0.415±0.005 | 0.425±0.005 |
| car | 0.987±0.003 | **0.995**±0.002 | **0.995**±0.001 | **0.995**±0.001 | **0.995**±0.001 | 0.959±0.005 | 0.927±0.008 | 0.980±0.004 | **0.995**±0.001 | **0.995**±0.001 | **0.995**±0.001 | **0.995**±0.001 | **0.995**±0.001 |
| climate model crashes | 0.881±0.014 | 0.507±0.024 | 0.748±0.023 | 0.887±0.024 | 0.902±0.022 | 0.124±0.019 | 0.388±0.023 | 0.876±0.024 | 0.713±0.021 | **0.903**±0.016 | 0.873±0.023 | 0.890±0.019 | 0.850±0.016 |
| concrete compression | **0.682**±0.016 | 0.050±0.008 | 0.268±0.015 | 0.184±0.012 | 0.424±0.020 | 0.002±0.002 | 0.033±0.007 | 0.488±0.011 | 0.260±0.016 | 0.383±0.020 | 0.392±0.020 | 0.339±0.014 | 0.332±0.016 |
| concrete slump | 0.745±0.075 | 0.132±0.042 | 0.655±0.060 | 0.675±0.050 | **0.758**±0.057 | 0.002±0.002 | 0.178±0.044 | 0.528±0.087 | 0.690±0.080 | 0.727±0.050 | 0.687±0.083 | 0.687±0.065 |  |
| congress | 0.922±0.011 | 0.774±0.028 | 0.931±0.019 | 0.266±0.025 | 0.269±0.015 | 0.163±0.027 | 0.600±0.021 | 0.894±0.026 | 0.874±0.028 | 0.943±0.022 | 0.945±0.015 | **0.949**±0.027 | 0.935±0.023 |
| connectionist bench sonar | **0.912**±0.026 | 0.055±0.018 | 0.533±0.040 | 0.663±0.042 | 0.700±0.047 | 0.000±0.000 | 0.245±0.034 | 0.588±0.040 | 0.411±0.038 | 0.769±0.038 | 0.689±0.046 | 0.679±0.044 | 0.599±0.027 |
| connectionist bench vowel | 0.751±0.016 | 0.032±0.007 | 0.136±0.018 | 0.352±0.019 | 0.630±0.024 | 0.002±0.002 | 0.018±0.006 | **0.789**±0.019 | 0.096±0.010 | 0.688±0.026 | 0.621±0.032 | 0.627±0.020 | 0.553±0.021 |
| ecoli | - | 0.393±0.035 | 0.760±0.038 | 0.690±0.028 | **0.829**±0.028 | 0.087±0.009 | 0.338±0.031 | 0.752±0.027 | 0.690±0.026 | 0.799±0.030 | 0.823±0.019 | 0.803±0.021 | 0.804±0.024 |
| glass | 0.792±0.026 | 0.187±0.043 | 0.661±0.047 | 0.402±0.042 | **0.796**±0.042 | 0.030±0.012 | 0.113±0.019 | 0.003±0.007 | 0.528±0.051 | 0.790±0.038 | 0.775±0.038 | 0.795±0.038 | 0.764±0.033 |
| ionosphere | **0.805**±0.023 | 0.041±0.012 | 0.457±0.034 | 0.493±0.034 | 0.675±0.036 | 0.002±0.002 | 0.063±0.009 | 0.045±0.018 | 0.368±0.026 | 0.687±0.034 | 0.675±0.034 | 0.617±0.040 | 0.556±0.032 |
| iris | 0.806±0.040 | 0.477±0.062 | 0.771±0.057 | 0.703±0.047 | **0.867**±0.037 | 0.051±0.029 | 0.323±0.062 | 0.788±0.039 | 0.649±0.067 | 0.857±0.049 | 0.852±0.048 | 0.846±0.061 | 0.841±0.058 |
| libras | **0.714**±0.018 | 0.004±0.006 | 0.134±0.029 | 0.283±0.032 | 0.390±0.034 | 0.000±0.000 | 0.005±0.004 | 0.000±0.000 | 0.129±0.023 | 0.634±0.035 | 0.466±0.036 | 0.606±0.039 | 0.288±0.021 |
| parkinsons | **0.823**±0.032 | 0.064±0.020 | 0.582±0.048 | 0.594±0.050 | 0.724±0.039 | 0.000±0.000 | 0.049±0.018 | 0.000±0.000 | 0.508±0.043 | 0.771±0.059 | 0.711±0.047 | 0.719±0.043 | 0.654±0.030 |
| planning relax | **0.871**±0.034 | 0.260±0.036 | 0.682±0.036 | 0.780±0.042 | 0.803±0.043 | 0.020±0.015 | 0.255±0.035 | 0.002±0.004 | 0.636±0.045 | 0.786±0.041 | 0.783±0.041 | 0.746±0.040 | 0.741±0.044 |
| qsar biodegradation | **0.711**±0.011 | 0.034±0.007 | 0.275±0.016 | 0.168±0.013 | 0.420±0.016 | 0.000±0.000 | 0.045±0.008 | 0.000±0.000 | 0.256±0.014 | 0.491±0.015 | 0.442±0.016 | 0.421±0.017 | 0.367±0.019 |
| seeds | 0.790±0.019 | 0.192±0.036 | 0.659±0.061 | 0.616±0.034 | **0.796**±0.028 | 0.003±0.008 | 0.090±0.018 | 0.670±0.024 | 0.564±0.057 | 0.790±0.050 | 0.781±0.054 | 0.781±0.049 | 0.768±0.040 |
| tictactoe | 0.992±0.002 | 0.994±0.002 | **0.996**±0.001 | 0.995±0.001 | 0.995±0.001 | 0.961±0.009 | 0.920±0.015 | **0.996**±0.001 | 0.995±0.001 | **0.996**±0.001 | **0.996**±0.001 | **0.996**±0.001 | **0.996**±0.001 |
| wine | 0.829±0.028 | 0.256±0.042 | 0.784±0.051 | 0.856±0.031 | **0.897**±0.038 | 0.013±0.013 | 0.313±0.035 | 0.002±0.008 | 0.651±0.050 | 0.882±0.039 | 0.879±0.038 | 0.872±0.048 | 0.862±0.041 |
| wine quality red | **0.686**±0.020 | 0.230±0.014 | 0.454±0.011 | 0.395±0.013 | 0.516±0.011 | 0.037±0.007 | 0.221±0.007 | 0.562±0.009 | 0.394±0.017 | 0.494±0.015 | 0.488±0.013 | 0.472±0.012 | 0.454±0.013 |
| wine quality white | **0.672**±0.008 | 0.211±0.009 | 0.335±0.007 | 0.279±0.007 | 0.383±0.006 | 0.044±0.003 | 0.168±0.006 | 0.405±0.007 | 0.286±0.008 | 0.379±0.007 | 0.362±0.005 | 0.353±0.009 | 0.339±0.010 |
| yacht hydrodynamics | 0.396±0.023 | 0.139±0.027 | 0.474±0.031 | 0.264±0.036 | **0.703**±0.029 | 0.001±0.003 | 0.061±0.009 | 0.510±0.044 | 0.479±0.037 | 0.657±0.043 | 0.698±0.038 | 0.624±0.043 | 0.638±0.037 |
| yeast | **0.861**±0.009 | 0.398±0.020 | 0.625±0.018 | 0.492±0.015 | 0.729±0.017 | 0.112±0.010 | 0.342±0.013 | 0.695±0.013 | 0.531±0.018 | 0.708±0.016 | 0.718±0.015 | 0.690±0.018 | 0.696±0.017 |

*Table 12.* Small Benchmark: MLE ($R^2$) metric scores for all datasets. Average and standard deviation ($\mu \pm \sigma$) over 20 sampling seeds. Best results per dataset are in **bold**, second best underlined. Real indicates MLE results when training on real data.

| model | Real | SMOTE | ARF | UT | FD | FF | CTGAN | TVAE | TabDDPM | XGenB-AR | XGenB-DF (x-DDPM) | XGenB-DF (v-DDPM) | XGenB-DF (x-DDIM) | XGenB-DF (v-DDIM) |
|---|---|---|---|---|---|---|---|---|---|---|---|---|---|---|
| airfoil self noise | 0.922±0.000 | 0.837±0.018 | 0.634±0.041 | 0.736±0.032 | 0.741±0.023 | 0.845±0.023 | -1.163±0.318 | 0.115±0.135 | **0.878**±0.013 | 0.737±0.029 | 0.830±0.017 | 0.817±0.016 | 0.827±0.017 | 0.812±0.026 |
| california | 0.826±0.003 | 0.761±0.004 | 0.680±0.009 | 0.762±0.004 | 0.688±0.009 | 0.711±0.006 | 0.389±0.017 | 0.662±0.008 | 0.768±0.005 | 0.727±0.006 | **0.772**±0.004 | 0.770±0.004 | 0.768±0.004 | 0.767±0.004 |
| concrete compression | 0.913±0.000 | **0.851**±0.014 | 0.452±0.073 | 0.704±0.034 | 0.744±0.021 | 0.813±0.018 | -0.776±0.255 | 0.435±0.109 | 0.835±0.016 | 0.749±0.025 | 0.818±0.018 | 0.810±0.019 | 0.806±0.025 | 0.790±0.028 |
| concrete slump | 0.837±0.000 | **0.774**±0.041 | 0.034±0.300 | 0.515±0.114 | 0.742±0.075 | 0.723±0.092 | -1.049±0.615 | 0.044±0.266 | -0.768±0.935 | 0.595±0.112 | 0.667±0.120 | 0.627±0.095 | 0.616±0.144 | 0.627±0.122 |
| wine quality red | 0.446±0.000 | **0.387**±0.025 | 0.260±0.043 | 0.301±0.043 | 0.324±0.035 | 0.301±0.030 | -0.066±0.062 | 0.301±0.015 | 0.329±0.040 | 0.292±0.033 | 0.347±0.030 | 0.348±0.037 | 0.328±0.037 | 0.332±0.031 |
| wine quality white | 0.486±0.000 | **0.336**±0.023 | 0.171±0.020 | 0.213±0.026 | 0.223±0.029 | 0.235±0.021 | -0.316±0.060 | 0.223±0.022 | 0.254±0.017 | 0.189±0.020 | 0.253±0.025 | 0.250±0.021 | 0.243±0.029 | 0.234±0.032 |
| yacht hydrodynamics | 0.998±0.000 | 0.997±0.001 | 0.333±0.246 | 0.992±0.004 | 0.965±0.020 | 0.995±0.005 | -0.592±0.604 | 0.188±0.213 | 0.919±0.027 | 0.932±0.038 | 0.971±0.025 | 0.990±0.007 | 0.963±0.035 | 0.982±0.024 |

*Table 13.* Small Benchmark: MLE (ROCAUC) metric scores for all datasets. Average and standard deviation ($\mu \pm \sigma$) over 20 sampling seeds. Best results per dataset are in **bold**, second best underlined. Real indicates MLE results when training on real data.

| model | Real | SMOTE | ARF | UT | FD | FF | CTGAN | TVAE | TabDDPM | XGenB-AR | XGenB-DF (x-DDPM) | XGenB-DF (v-DDPM) | XGenB-DF (x-DDIM) | XGenB-DF (v-DDIM) |
|---|---|---|---|---|---|---|---|---|---|---|---|---|---|---|
| bean | 0.997±0.000 | **0.996**±0.000 | 0.995±0.000 | **0.996**±0.000 | 0.996±0.000 | **0.996**±0.000 | 0.983±0.001 | 0.989±0.001 | 0.989±0.001 | **0.996**±0.000 | **0.996**±0.000 | **0.996**±0.000 | **0.996**±0.000 | **0.996**±0.000 |
| blood transfusion | 0.783±0.010 | **0.780**±0.010 | 0.766±0.033 | 0.746±0.020 | 0.775±0.019 | 0.761±0.023 | 0.561±0.001 | 0.500±0.000 | 0.728±0.025 | 0.727±0.030 | 0.743±0.025 | 0.741±0.030 | 0.731±0.029 | 0.728±0.026 |
| breast cancer diagnostic | 0.991±0.000 | 0.990±0.002 | 0.982±0.009 | 0.984±0.006 | 0.990±0.002 | 0.989±0.003 | 0.848±0.078 | 0.988±0.004 | 0.499±0.050 | 0.990±0.003 | **0.991**±0.003 | 0.991±0.002 | **0.991**±0.003 | 0.991±0.002 |
| car | 1.000±0.000 | 0.993±0.001 | 0.985±0.003 | 0.996±0.001 | 0.987±0.002 | 0.988±0.002 | 0.855±0.014 | 0.950±0.001 | 0.815±0.013 | 0.986±0.004 | **0.998**±0.001 | 0.998±0.001 | 0.997±0.001 | 0.998±0.001 |
| climate model crashes | 0.919±0.005 | 0.865±0.021 | 0.720±0.093 | 0.863±0.049 | **0.894**±0.029 | 0.885±0.026 | 0.570±0.069 | 0.500±0.000 | 0.881±0.032 | 0.889±0.028 | 0.839±0.026 | 0.868±0.031 | 0.862±0.031 | 0.868±0.053 |
| congress | 0.988±0.000 | **0.986**±0.004 | 0.968±0.013 | 0.981±0.012 | 0.977±0.006 | 0.977±0.009 | 0.558±0.192 | 0.962±0.014 | 0.989±0.005 | 0.982±0.010 | 0.970±0.011 | 0.973±0.011 | 0.973±0.013 | 0.969±0.017 |
| connectionist bench sonar | 0.914±0.000 | **0.886**±0.026 | 0.681±0.086 | 0.777±0.038 | 0.811±0.050 | 0.799±0.037 | 0.470±0.089 | 0.767±0.048 | 0.502±0.061 | 0.786±0.043 | 0.833±0.034 | 0.830±0.033 | 0.822±0.053 | 0.825±0.036 |
| connectionist bench vowel | 0.992±0.000 | 0.968±0.005 | 0.898±0.014 | 0.948±0.012 | 0.970±0.006 | 0.968±0.006 | 0.494±0.031 | 0.849±0.011 | **0.974**±0.002 | 0.949±0.008 | 0.969±0.005 | 0.969±0.005 | 0.971±0.004 | 0.971±0.005 |
| ecoli | 0.987±0.001 | - | 0.977±0.006 | 0.981±0.005 | **0.986**±0.004 | 0.984±0.007 | 0.850±0.040 | 0.899±0.014 | 0.982±0.004 | 0.982±0.005 | 0.982±0.004 | 0.982±0.004 | 0.982±0.004 | 0.983±0.004 |
| glass | 0.941±0.000 | 0.899±0.013 | 0.866±0.024 | 0.884±0.014 | 0.903±0.012 | **0.913**±0.014 | 0.615±0.075 | 0.814±0.023 | 0.679±0.065 | 0.893±0.021 | 0.900±0.018 | 0.902±0.023 | 0.897±0.023 | 0.903±0.027 |
| ionosphere | 0.963±0.000 | 0.881±0.037 | 0.898±0.045 | 0.932±0.031 | 0.949±0.025 | 0.938±0.014 | 0.456±0.135 | 0.925±0.023 | 0.484±0.098 | **0.961**±0.016 | **0.964**±0.013 | 0.964±0.013 | **0.961**±0.016 | 0.950±0.017 |
| iris | 0.955±0.000 | 0.958±0.032 | **0.979**±0.014 | 0.962±0.019 | 0.970±0.022 | 0.968±0.022 | 0.887±0.077 | 0.943±0.030 | 0.969±0.023 | 0.971±0.022 | 0.960±0.029 | 0.959±0.032 | 0.960±0.016 | 0.960±0.025 |
| libras | 0.955±0.004 | **0.925**±0.006 | 0.851±0.022 | 0.866±0.018 | 0.861±0.020 | 0.789±0.029 | 0.514±0.045 | 0.789±0.029 | 0.494±0.012 | 0.832±0.023 | 0.865±0.018 | 0.865±0.022 | 0.863±0.020 | 0.879±0.013 |
| parkinsons | 0.920±0.002 | **0.904**±0.023 | 0.755±0.060 | 0.812±0.080 | 0.862±0.043 | 0.846±0.044 | 0.647±0.125 | 0.528±0.077 | 0.495±0.039 | 0.847±0.038 | 0.839±0.053 | 0.844±0.043 | 0.854±0.034 | 0.844±0.036 |
| planning relax | 0.465±0.100 | **0.544**±0.048 | 0.474±0.068 | 0.497±0.087 | 0.491±0.068 | 0.494±0.067 | 0.442±0.058 | 0.500±0.040 | 0.497±0.029 | 0.478±0.074 | 0.506±0.073 | 0.531±0.094 | 0.465±0.083 | 0.466±0.087 |
| qsar biodegradation | 0.910±0.001 | 0.886±0.007 | 0.855±0.014 | 0.879±0.015 | 0.880±0.012 | 0.881±0.008 | 0.383±0.094 | 0.798±0.023 | 0.488±0.078 | 0.878±0.016 | 0.884±0.014 | 0.887±0.013 | 0.885±0.011 | **0.890**±0.014 |
| seeds | 0.995±0.000 | 0.969±0.011 | 0.956±0.016 | 0.976±0.010 | **0.985**±0.008 | **0.985**±0.009 | 0.720±0.078 | 0.982±0.009 | 0.983±0.011 | 0.980±0.012 | 0.981±0.008 | 0.981±0.010 | 0.981±0.010 | 0.979±0.008 |
| tictactoe | 0.984±0.000 | 0.919±0.020 | 0.517±0.038 | 0.947±0.021 | 0.933±0.025 | 0.904±0.019 | 0.511±0.048 | 0.711±0.031 | 0.972±0.009 | 0.917±0.030 | **0.981**±0.009 | 0.980±0.009 | 0.980±0.010 | 0.981±0.010 |
| wine | 0.999±0.001 | 0.985±0.004 | 0.987±0.013 | 0.994±0.005 | 0.993±0.006 | **0.999**±0.004 | 0.596±0.126 | 0.971±0.026 | 0.542±0.023 | 0.990±0.006 | 0.988±0.012 | 0.989±0.013 | 0.993±0.007 | 0.993±0.005 |
| yeast | 0.935±0.000 | 0.919±0.002 | 0.916±0.004 | 0.921±0.004 | **0.926**±0.004 | 0.921±0.004 | 0.593±0.045 | 0.852±0.016 | 0.923±0.003 | 0.922±0.004 | 0.923±0.003 | 0.922±0.004 | 0.926±0.004 | 0.923±0.004 |

*Table 14.* Small Benchmark: MLE (F1) metric scores for all datasets. Average and standard deviation ($\mu \pm \sigma$) over 20 sampling seeds. Best results per dataset are in **bold**, second best underlined. Real indicates MLE results when training on real data.

| model | Real | SMOTE | ARF | UT | FD | FF | CTGAN | TVAE | TabDDPM | XGenB-AR | XGenB-DF (x-DDPM) | XGenB-DF (v-DDPM) | XGenB-DF (x-DDIM) | XGenB-DF (v-DDIM) |
|---|---|---|---|---|---|---|---|---|---|---|---|---|---|---|
| bean | 0.931±0.001 | 0.927±0.000 | 0.915±0.003 | 0.924±0.002 | 0.923±0.001 | **0.928**±0.003 | 0.858±0.006 | 0.890±0.001 | 0.929±0.003 | 0.924±0.000 | 0.926±0.002 | 0.925±0.002 | 0.925±0.002 | 0.925±0.003 |
| blood transfusion | 0.432±0.044 | **0.475**±0.046 | 0.344±0.138 | 0.421±0.059 | 0.371±0.111 | 0.413±0.137 | 0.285±0.115 | 0.000±0.000 | 0.438±0.081 | 0.372±0.116 | 0.432±0.080 | 0.412±0.083 | 0.366±0.124 | 0.370±0.119 |
| breast cancer diagnostic | 0.947±0.004 | 0.943±0.009 | 0.950±0.016 | 0.948±0.009 | 0.949±0.008 | 0.951±0.009 | 0.879±0.091 | 0.932±0.015 | 0.582±0.342 | 0.956±0.011 | **0.957**±0.010 | 0.954±0.009 | 0.950±0.010 | 0.951±0.012 |
| car | 0.977±0.000 | 0.915±0.013 | 0.874±0.013 | 0.944±0.011 | 0.884±0.016 | 0.891±0.017 | 0.648±0.026 | 0.760±0.016 | 0.585±0.026 | 0.883±0.020 | **0.961**±0.012 | 0.955±0.012 | 0.953±0.010 | 0.960±0.011 |
| climate model crashes | 0.975±0.005 | **0.975**±0.004 | 0.963±0.003 | 0.957±0.001 | 0.972±0.007 | 0.972±0.006 | 0.892±0.046 | 0.957±0.006 | 0.973±0.006 | 0.963±0.007 | 0.966±0.009 | 0.969±0.007 | 0.967±0.008 | 0.970±0.007 |
| congress | 0.947±0.005 | 0.950±0.004 | 0.925±0.014 | 0.943±0.008 | 0.943±0.014 | 0.938±0.007 | 0.605±0.127 | 0.915±0.015 | **0.952**±0.006 | 0.945±0.006 | 0.946±0.006 | 0.947±0.009 | 0.943±0.011 | 0.946±0.007 |
| connectionist bench sonar | 0.800±0.008 | **0.815**±0.029 | 0.653±0.072 | 0.653±0.049 | 0.739±0.049 | 0.721±0.051 | 0.483±0.175 | 0.722±0.060 | 0.328±0.303 | 0.733±0.045 | 0.733±0.033 | 0.749±0.047 | 0.737±0.045 | 0.742±0.041 |
| connectionist bench vowel | 0.833±0.000 | 0.740±0.024 | 0.489±0.032 | 0.638±0.042 | 0.748±0.023 | 0.739±0.025 | 0.080±0.021 | 0.392±0.033 | **0.767**±0.027 | 0.653±0.028 | 0.749±0.039 | 0.742±0.035 | 0.753±0.023 | 0.755±0.024 |
| ecoli | 0.803±0.007 | - | 0.780±0.038 | 0.809±0.043 | **0.818**±0.025 | 0.809±0.030 | 0.421±0.175 | 0.684±0.043 | 0.797±0.032 | 0.796±0.046 | 0.788±0.026 | 0.788±0.028 | 0.795±0.031 | 0.788±0.034 |
| glass | 0.789±0.023 | 0.623±0.049 | 0.584±0.052 | 0.645±0.054 | 0.659±0.060 | **0.680**±0.046 | 0.288±0.089 | 0.507±0.057 | 0.321±0.077 | 0.637±0.062 | 0.636±0.045 | 0.647±0.044 | 0.641±0.060 | 0.644±0.057 |
| ionosphere | 0.939±0.000 | 0.746±0.041 | 0.655±0.129 | 0.688±0.114 | 0.869±0.056 | 0.834±0.056 | 0.309±0.126 | 0.830±0.042 | 0.014±0.044 | 0.857±0.046 | **0.893**±0.025 | 0.871±0.039 | 0.886±0.029 | 0.854±0.046 |
| iris | 0.933±0.000 | 0.890±0.042 | 0.908±0.054 | **0.927**±0.034 | 0.908±0.049 | 0.920±0.041 | 0.750±0.083 | 0.827±0.061 | 0.927±0.040 | 0.912±0.045 | 0.907±0.048 | 0.905±0.047 | 0.918±0.031 | 0.918±0.044 |
| libras | 0.693±0.014 | **0.581**±0.021 | 0.386±0.040 | 0.426±0.041 | 0.478±0.050 | 0.483±0.056 | 0.072±0.033 | 0.327±0.061 | 0.067±0.010 | 0.406±0.047 | 0.492±0.038 | 0.481±0.046 | 0.477±0.053 | 0.507±0.042 |
| parkinsons | 0.630±0.039 | **0.684**±0.033 | 0.472±0.112 | 0.505±0.129 | 0.572±0.069 | 0.572±0.094 | 0.469±0.077 | 0.000±0.000 | 0.000±0.000 | 0.568±0.079 | 0.603±0.069 | 0.589±0.064 | 0.625±0.093 | 0.582±0.085 |
| planning relax | 0.062±0.067 | **0.239**±0.124 | 0.109±0.111 | 0.029±0.061 | 0.232±0.150 | 0.239±0.131 | 0.178±0.116 | 0.000±0.000 | 0.000±0.000 | 0.154±0.083 | 0.229±0.112 | 0.230±0.124 | 0.221±0.120 | 0.197±0.144 |
| qsar biodegradation | 0.891±0.007 | 0.876±0.008 | 0.848±0.014 | 0.868±0.013 | 0.871±0.016 | 0.877±0.010 | 0.750±0.030 | 0.821±0.014 | 0.798±0.020 | 0.866±0.016 | 0.878±0.015 | 0.879±0.012 | 0.881±0.013 | **0.882**±0.013 |
| seeds | 0.929±0.000 | 0.874±0.021 | 0.833±0.044 | 0.870±0.037 | 0.887±0.044 | **0.900**±0.041 | 0.500±0.087 | 0.886±0.032 | 0.898±0.037 | 0.893±0.041 | 0.879±0.041 | 0.898±0.044 | 0.877±0.038 | 0.873±0.038 |
| tictactoe | 0.921±0.000 | 0.767±0.040 | 0.625±0.044 | 0.813±0.049 | 0.784±0.041 | 0.733±0.035 | 0.229±0.070 | 0.477±0.056 | 0.876±0.024 | 0.765±0.060 | 0.906±0.032 | 0.901±0.030 | 0.903±0.035 | **0.913**±0.026 |
| wine | 0.972±0.000 | 0.893±0.035 | 0.924±0.041 | 0.936±0.037 | 0.936±0.038 | **0.940**±0.034 | 0.394±0.142 | 0.897±0.042 | 0.362±0.031 | 0.928±0.032 | 0.906±0.064 | 0.925±0.051 | 0.935±0.054 | 0.940±0.043 |
| yeast | 0.614±0.008 | 0.560±0.015 | 0.517±0.029 | 0.550±0.020 | 0.572±0.014 | 0.564±0.019 | 0.180±0.040 | 0.418±0.018 | **0.575**±0.016 | 0.570±0.019 | 0.564±0.027 | 0.570±0.026 | 0.574±0.019 | 0.566±0.020 |

*Table 15.* Small Benchmark: MLE (RMSE) metric scores for all datasets. Average and standard deviation ($\mu \pm \sigma$) over 20 sampling seeds. Best results per dataset are in **bold**, second best underlined. Real indicates MLE results when training on real data.

| model | Real | SMOTE | ARF | UT | FD | FF | CTGAN | TVAE | TabDDPM | XGenB-AR | XGenB-DF (x-DDPM) | XGenB-DF (v-DDPM) | XGenB-DF (x-DDIM) | XGenB-DF (v-DDIM) |
|---|---|---|---|---|---|---|---|---|---|---|---|---|---|---|
| airfoil self noise | 0.291±0.000 | 0.421±0.023 | 0.629±0.035 | 0.535±0.033 | 0.530±0.025 | **0.410**±0.030 | 1.528±0.110 | 0.977±0.073 | 0.363±0.019 | 0.533±0.029 | 0.429±0.021 | 0.446±0.019 | 0.433±0.021 | 0.451±0.031 |
| california | 0.415±0.004 | 0.486±0.004 | 0.562±0.008 | 0.485±0.004 | 0.555±0.008 | 0.534±0.007 | 0.778±0.011 | 0.578±0.007 | 0.480±0.005 | 0.519±0.006 | **0.475**±0.006 | 0.477±0.004 | 0.479±0.004 | 0.481±0.004 |
| concrete compression | 0.285±0.000 | **0.371**±0.017 | 0.712±0.047 | 0.524±0.030 | 0.488±0.020 | 0.417±0.020 | 1.282±0.093 | 0.722±0.067 | 0.391±0.018 | 0.482±0.024 | 0.411±0.021 | 0.420±0.021 | 0.424±0.028 | 0.441±0.030 |
| concrete slump | 0.376±0.000 | **0.442**±0.044 | 0.907±0.142 | 0.646±0.077 | 0.469±0.071 | 0.485±0.077 | 1.321±0.204 | 0.904±0.129 | 1.206±0.302 | 0.589±0.079 | 0.530±0.098 | 0.566±0.073 | 0.569±0.109 | 0.563±0.091 |
| wine quality red | 0.744±0.000 | **0.783**±0.016 | 0.860±0.025 | 0.836±0.016 | 0.822±0.020 | 0.821±0.021 | 1.032±0.030 | 0.836±0.009 | 0.819±0.024 | 0.841±0.040 | 0.808±0.015 | 0.807±0.022 | 0.820±0.022 | 0.817±0.019 |
| wine quality white | 0.706±0.000 | **0.802**±0.014 | 0.897±0.011 | 0.874±0.014 | 0.868±0.016 | 0.861±0.012 | 1.129±0.026 | 0.868±0.013 | 0.851±0.010 | 0.887±0.015 | 0.851±0.014 | 0.853±0.011 | 0.857±0.016 | 0.862±0.018 |
| yacht hydrodynamics | 0.031±0.000 | **0.041**±0.004 | 0.629±0.110 | 0.067±0.015 | 0.140±0.038 | 0.051±0.018 | 0.974±0.160 | 0.698±0.093 | 0.219±0.035 | 0.196±0.060 | 0.122±0.053 | 0.076±0.025 | 0.137±0.060 | 0.093±0.049 |

*Table 16.* Small Benchmark: DCR metric scores for all datasets. Average and standard devation ($\mu \pm \sigma$) over 20 sampling seeds. Best results per dataset are in **bold**, second best underlined.

| model | SMOTE | ARF | UT | FD | FF | CTGAN | TVAE | TabDDPM | XGenB-AR | XGenB-DF (x-DDPM) | XGenB-DF (v-DDPM) | XGenB-DF (x-DDIM) | XGenB-DF (v-DDIM) |
|---|---|---|---|---|---|---|---|---|---|---|---|---|---|
| airfoil self noise | $0.722_{\pm0.039}$ | $\underline{0.916}_{\pm0.030}$ | $0.864_{\pm0.047}$ | $0.857_{\pm0.039}$ | $0.836_{\pm0.026}$ | $\mathbf{0.939}_{\pm0.042}$ | $0.901_{\pm0.038}$ | $0.818_{\pm0.048}$ | $0.878_{\pm0.043}$ | $0.852_{\pm0.036}$ | $0.828_{\pm0.040}$ | $0.859_{\pm0.050}$ | $0.847_{\pm0.048}$ |
| bean | $0.817_{\pm0.010}$ | $0.915_{\pm0.014}$ | $0.934_{\pm0.015}$ | $0.962_{\pm0.014}$ | $0.905_{\pm0.015}$ | $\mathbf{0.987}_{\pm0.011}$ | $0.967_{\pm0.010}$ | $\underline{0.976}_{\pm0.011}$ | $0.943_{\pm0.013}$ | $0.901_{\pm0.015}$ | $0.907_{\pm0.013}$ | $0.917_{\pm0.015}$ | $0.922_{\pm0.015}$ |
| blood transfusion | $0.988_{\pm0.019}$ | $\mathbf{1.000}_{\pm0.000}$ | $0.965_{\pm0.032}$ | $0.983_{\pm0.023}$ | $0.986_{\pm0.024}$ | $0.984_{\pm0.017}$ | $0.983_{\pm0.030}$ | $\underline{0.996}_{\pm0.009}$ | $0.996_{\pm0.010}$ | $0.990_{\pm0.018}$ | $0.991_{\pm0.012}$ | $0.996_{\pm0.011}$ | $0.988_{\pm0.014}$ |
| breast cancer diagnostic | $0.667_{\pm0.055}$ | $\underline{0.892}_{\pm0.054}$ | $0.718_{\pm0.069}$ | $0.741_{\pm0.068}$ | $0.690_{\pm0.068}$ | $\mathbf{0.947}_{\pm0.024}$ | $0.847_{\pm0.029}$ | $0.723_{\pm0.083}$ | $0.718_{\pm0.071}$ | $0.644_{\pm0.066}$ | $0.661_{\pm0.068}$ | $0.692_{\pm0.057}$ | $0.704_{\pm0.058}$ |
| california | $0.842_{\pm0.012}$ | $0.968_{\pm0.011}$ | $0.939_{\pm0.010}$ | $0.964_{\pm0.007}$ | $0.941_{\pm0.011}$ | $\underline{0.974}_{\pm0.011}$ | $\mathbf{0.986}_{\pm0.010}$ | $0.723_{\pm0.012}$ | $0.962_{\pm0.010}$ | $0.964_{\pm0.010}$ | $0.961_{\pm0.013}$ | $0.966_{\pm0.008}$ | $0.962_{\pm0.010}$ |
| car | $\mathbf{1.000}_{\pm0.000}$ | $1.000_{\pm0.000}$ | $1.000_{\pm0.000}$ | $1.000_{\pm0.000}$ | $1.000_{\pm0.000}$ | $1.000_{\pm0.000}$ | $1.000_{\pm0.000}$ | $1.000_{\pm0.000}$ | $1.000_{\pm0.000}$ | $1.000_{\pm0.000}$ | $1.000_{\pm0.000}$ | $1.000_{\pm0.000}$ | $1.000_{\pm0.000}$ |
| climate model crashes | $0.535_{\pm0.047}$ | $0.887_{\pm0.056}$ | $0.636_{\pm0.066}$ | $0.526_{\pm0.042}$ | $0.492_{\pm0.036}$ | $\mathbf{0.957}_{\pm0.053}$ | $\underline{0.931}_{\pm0.059}$ | $0.503_{\pm0.059}$ | $0.677_{\pm0.051}$ | $0.504_{\pm0.051}$ | $0.572_{\pm0.056}$ | $0.533_{\pm0.066}$ | $0.565_{\pm0.051}$ |
| concrete compression | $0.697_{\pm0.043}$ | $\underline{0.918}_{\pm0.045}$ | $0.880_{\pm0.048}$ | $0.858_{\pm0.053}$ | $0.821_{\pm0.051}$ | $0.897_{\pm0.058}$ | $\mathbf{0.925}_{\pm0.023}$ | $0.760_{\pm0.044}$ | $0.859_{\pm0.049}$ | $0.811_{\pm0.049}$ | $0.822_{\pm0.050}$ | $0.829_{\pm0.047}$ | $0.845_{\pm0.044}$ |
| concrete slump | $0.717_{\pm0.120}$ | $\underline{0.916}_{\pm0.071}$ | $0.697_{\pm0.139}$ | $0.779_{\pm0.117}$ | $0.791_{\pm0.128}$ | $0.907_{\pm0.081}$ | $\mathbf{0.938}_{\pm0.053}$ | $0.732_{\pm0.120}$ | $0.790_{\pm0.090}$ | $0.751_{\pm0.116}$ | $0.820_{\pm0.135}$ | $0.748_{\pm0.162}$ | $0.767_{\pm0.135}$ |
| congress | $0.992_{\pm0.016}$ | $\mathbf{1.000}_{\pm0.000}$ | $0.996_{\pm0.011}$ | $\underline{0.999}_{\pm0.003}$ | $1.000_{\pm0.000}$ | $1.000_{\pm0.000}$ | $1.000_{\pm0.000}$ | $0.999_{\pm0.006}$ | $1.000_{\pm0.000}$ | $0.996_{\pm0.007}$ | $0.984_{\pm0.022}$ | $0.984_{\pm0.025}$ | $0.988_{\pm0.018}$ |
| connectionist bench sonar | $0.541_{\pm0.076}$ | $0.713_{\pm0.098}$ | $0.556_{\pm0.074}$ | $0.587_{\pm0.127}$ | $0.553_{\pm0.099}$ | $0.693_{\pm0.114}$ | $\underline{0.769}_{\pm0.128}$ | $\mathbf{0.883}_{\pm0.095}$ | $0.694_{\pm0.066}$ | $0.601_{\pm0.086}$ | $0.613_{\pm0.087}$ | $0.610_{\pm0.106}$ | $0.610_{\pm0.102}$ |
| connectionist bench vowel | $0.646_{\pm0.033}$ | $0.912_{\pm0.036}$ | $0.806_{\pm0.043}$ | $0.748_{\pm0.051}$ | $0.697_{\pm0.042}$ | $\mathbf{0.983}_{\pm0.021}$ | $\underline{0.974}_{\pm0.024}$ | $0.708_{\pm0.101}$ | $0.770_{\pm0.072}$ | $0.699_{\pm0.080}$ | $0.687_{\pm0.067}$ | $0.731_{\pm0.071}$ | $0.697_{\pm0.073}$ |
| ecoli | - | $\underline{0.878}_{\pm0.075}$ | $0.695_{\pm0.067}$ | $0.773_{\pm0.055}$ | $0.745_{\pm0.087}$ | $0.813_{\pm0.087}$ | $\mathbf{0.961}_{\pm0.044}$ | $0.708_{\pm0.101}$ | $0.770_{\pm0.072}$ | $0.699_{\pm0.080}$ | $0.687_{\pm0.067}$ | $0.731_{\pm0.071}$ | $0.697_{\pm0.073}$ |
| glass | $0.755_{\pm0.064}$ | $0.938_{\pm0.059}$ | $0.871_{\pm0.060}$ | $0.859_{\pm0.095}$ | $0.767_{\pm0.098}$ | $\mathbf{0.993}_{\pm0.018}$ | $\underline{0.952}_{\pm0.053}$ | $0.633_{\pm0.144}$ | $0.888_{\pm0.060}$ | $0.762_{\pm0.084}$ | $0.762_{\pm0.105}$ | $0.787_{\pm0.084}$ | $0.788_{\pm0.083}$ |
| ionosphere | $0.674_{\pm0.061}$ | $\underline{0.880}_{\pm0.061}$ | $0.725_{\pm0.086}$ | $0.744_{\pm0.086}$ | $0.742_{\pm0.071}$ | $0.827_{\pm0.090}$ | $\mathbf{0.937}_{\pm0.041}$ | $0.842_{\pm0.091}$ | $0.740_{\pm0.092}$ | $0.641_{\pm0.070}$ | $0.755_{\pm0.087}$ | $0.652_{\pm0.092}$ | $0.735_{\pm0.088}$ |
| iris | $0.616_{\pm0.139}$ | $0.783_{\pm0.094}$ | $0.638_{\pm0.090}$ | $0.697_{\pm0.106}$ | $0.613_{\pm0.144}$ | $\underline{0.882}_{\pm0.083}$ | $\mathbf{0.912}_{\pm0.080}$ | $0.599_{\pm0.124}$ | $0.748_{\pm0.118}$ | $0.637_{\pm0.071}$ | $0.574_{\pm0.105}$ | $0.622_{\pm0.105}$ | $0.619_{\pm0.108}$ |
| libras | $0.750_{\pm0.059}$ | $0.884_{\pm0.051}$ | $0.823_{\pm0.106}$ | $0.751_{\pm0.075}$ | $0.712_{\pm0.063}$ | $0.816_{\pm0.100}$ | $\underline{0.928}_{\pm0.054}$ | $\mathbf{0.942}_{\pm0.047}$ | $0.786_{\pm0.080}$ | $0.637_{\pm0.071}$ | $0.696_{\pm0.070}$ | $0.631_{\pm0.085}$ | $0.694_{\pm0.086}$ |
| parkinsons | $0.778_{\pm0.107}$ | $0.918_{\pm0.067}$ | $0.783_{\pm0.094}$ | $0.775_{\pm0.115}$ | $0.800_{\pm0.090}$ | $0.738_{\pm0.099}$ | $\underline{0.921}_{\pm0.070}$ | $\mathbf{0.982}_{\pm0.029}$ | $0.599_{\pm0.124}$ | $0.795_{\pm0.089}$ | $0.762_{\pm0.105}$ | $0.755_{\pm0.104}$ | $0.764_{\pm0.091}$ |
| planning relax | $0.646_{\pm0.109}$ | $\underline{0.848}_{\pm0.097}$ | $0.740_{\pm0.070}$ | $0.663_{\pm0.100}$ | $0.695_{\pm0.073}$ | $0.835_{\pm0.084}$ | $\mathbf{0.901}_{\pm0.054}$ | $0.758_{\pm0.069}$ | $0.762_{\pm0.110}$ | $0.671_{\pm0.088}$ | $0.706_{\pm0.097}$ | $0.705_{\pm0.086}$ | $0.723_{\pm0.081}$ |
| qsar biodegradation | $0.624_{\pm0.046}$ | $0.873_{\pm0.062}$ | $0.873_{\pm0.045}$ | $0.798_{\pm0.050}$ | $0.806_{\pm0.047}$ | $\mathbf{0.973}_{\pm0.022}$ | $\underline{0.945}_{\pm0.046}$ | $0.893_{\pm0.064}$ | $0.846_{\pm0.044}$ | $0.768_{\pm0.053}$ | $0.773_{\pm0.056}$ | $0.835_{\pm0.054}$ | $0.837_{\pm0.048}$ |
| seeds | $0.627_{\pm0.093}$ | $0.841_{\pm0.077}$ | $0.567_{\pm0.096}$ | $0.614_{\pm0.094}$ | $0.582_{\pm0.108}$ | $\mathbf{0.955}_{\pm0.059}$ | $\underline{0.850}_{\pm0.076}$ | $0.740_{\pm0.100}$ | $0.710_{\pm0.103}$ | $0.584_{\pm0.079}$ | $0.588_{\pm0.115}$ | $0.573_{\pm0.107}$ | $0.588_{\pm0.114}$ |
| tictactoe | $\underline{0.993}_{\pm0.011}$ | $\mathbf{1.000}_{\pm0.000}$ | $1.000_{\pm0.000}$ | $1.000_{\pm0.000}$ | $1.000_{\pm0.000}$ | $1.000_{\pm0.000}$ | $1.000_{\pm0.000}$ | $1.000_{\pm0.000}$ | $1.000_{\pm0.000}$ | $1.000_{\pm0.000}$ | $1.000_{\pm0.000}$ | $1.000_{\pm0.000}$ | $1.000_{\pm0.000}$ |
| wine | $0.525_{\pm0.072}$ | $0.787_{\pm0.092}$ | $0.495_{\pm0.068}$ | $0.500_{\pm0.098}$ | $0.434_{\pm0.118}$ | $\mathbf{0.898}_{\pm0.057}$ | $\underline{0.811}_{\pm0.073}$ | $0.682_{\pm0.137}$ | $0.596_{\pm0.100}$ | $0.474_{\pm0.075}$ | $0.482_{\pm0.086}$ | $0.503_{\pm0.087}$ | $0.498_{\pm0.100}$ |
| wine quality red | $0.712_{\pm0.033}$ | $0.928_{\pm0.035}$ | $0.821_{\pm0.034}$ | $0.873_{\pm0.040}$ | $0.790_{\pm0.038}$ | $\mathbf{0.995}_{\pm0.012}$ | $\underline{0.992}_{\pm0.015}$ | $0.713_{\pm0.041}$ | $0.864_{\pm0.047}$ | $0.827_{\pm0.041}$ | $0.812_{\pm0.030}$ | $0.845_{\pm0.036}$ | $0.835_{\pm0.037}$ |
| wine quality white | $0.770_{\pm0.020}$ | $0.969_{\pm0.018}$ | $0.921_{\pm0.024}$ | $0.962_{\pm0.017}$ | $0.928_{\pm0.013}$ | $\mathbf{1.000}_{\pm0.001}$ | $\underline{0.977}_{\pm0.016}$ | $0.846_{\pm0.033}$ | $0.959_{\pm0.020}$ | $0.924_{\pm0.025}$ | $0.932_{\pm0.022}$ | $0.944_{\pm0.024}$ | $0.951_{\pm0.018}$ |
| yacht hydrodynamics | $0.703_{\pm0.065}$ | $0.852_{\pm0.075}$ | $0.803_{\pm0.087}$ | $0.839_{\pm0.089}$ | $0.825_{\pm0.072}$ | $\underline{0.920}_{\pm0.069}$ | $\mathbf{0.994}_{\pm0.012}$ | $0.856_{\pm0.070}$ | $0.817_{\pm0.078}$ | $0.822_{\pm0.077}$ | $0.842_{\pm0.072}$ | $0.838_{\pm0.099}$ | $0.860_{\pm0.076}$ |
| yeast | $0.618_{\pm0.039}$ | $0.907_{\pm0.041}$ | $0.783_{\pm0.036}$ | $0.794_{\pm0.041}$ | $0.697_{\pm0.036}$ | $\mathbf{0.982}_{\pm0.019}$ | $\underline{0.956}_{\pm0.024}$ | $0.701_{\pm0.041}$ | $0.843_{\pm0.034}$ | $0.717_{\pm0.032}$ | $0.691_{\pm0.039}$ | $0.732_{\pm0.035}$ | $0.712_{\pm0.044}$ |

*Table 17.* Small Benchmark: training time in minutes for all datasets. Best results per dataset are in **bold**, second best underlined.

| model | SMOTE | ARF | UT | FD | FF | CTGAN | TVAE | TabDDPM | XGenB-AR | XGenB-DF (x-DDPM) | XGenB-DF (v-DDPM) | XGenB-DF (x-DDIM) | XGenB-DF (v-DDIM) |
|---|---|---|---|---|---|---|---|---|---|---|---|---|---|
| airfoil self noise | - | $\underline{0.050}$ | 0.205 | 0.346 | 0.371 | 0.067 | 0.081 | 1.158 | **0.018** | 0.339 | 0.326 | 0.419 | 0.298 |
| bean | - | 4.083 | 9.696 | 16.384 | 16.676 | 1.940 | $\underline{1.208}$ | 9.132 | **0.069** | 18.009 | 15.328 | 13.069 | 14.507 |
| blood transfusion | - | $\underline{0.031}$ | 0.120 | 0.143 | 0.137 | 0.080 | 0.038 | 0.729 | **0.011** | 0.134 | 0.140 | 0.130 | 0.134 |
| breast cancer diagnostic | - | 0.178 | 2.439 | 6.791 | 6.375 | 0.211 | $\underline{0.151}$ | 0.674 | **0.102** | 6.213 | 6.016 | 5.511 | 5.755 |
| california | - | 3.100 | 3.789 | 9.872 | 10.087 | 1.695 | $\underline{1.264}$ | 14.155 | **0.037** | 14.022 | 10.392 | 8.443 | 10.041 |
| car | - | 0.069 | 0.123 | 3.478 | 3.448 | 0.154 | $\underline{0.065}$ | 1.814 | **0.001** | 2.858 | 2.964 | 2.773 | 2.916 |
| climate model crashes | - | **0.013** | 1.029 | 2.248 | 2.217 | 0.154 | 0.118 | 0.656 | $\underline{0.060}$ | 2.038 | 1.950 | 2.162 | 1.804 |
| concrete compression | - | $\underline{0.046}$ | 0.434 | 0.527 | 0.540 | 0.084 | 0.136 | 0.871 | **0.028** | 0.556 | 0.492 | 0.458 | 0.448 |
| concrete slump | - | **0.008** | 0.173 | 0.204 | 0.186 | 0.069 | 0.042 | 0.465 | $\underline{0.024}$ | 0.167 | 0.171 | 0.159 | 0.165 |
| congress | - | 0.033 | 0.153 | 6.894 | 6.437 | 0.107 | $\underline{0.029}$ | 1.792 | **0.003** | 1.982 | 2.089 | 2.067 | 2.051 |
| connectionist bench sonar | - | 0.059 | 4.553 | 21.489 | 20.400 | 0.328 | 0.246 | 0.617 | $\underline{0.220}$ | 19.929 | 18.293 | 18.031 | 17.882 |
| connectionist bench vowel | - | $\underline{0.055}$ | 0.600 | 3.058 | 2.905 | 0.147 | 0.142 | 0.826 | **0.033** | 3.006 | 2.832 | 3.013 | 2.797 |
| ecoli | - | $\underline{0.024}$ | 0.185 | 0.646 | 0.627 | 0.091 | 0.043 | 0.579 | **0.020** | 0.616 | 0.770 | 0.622 | 0.743 |
| glass | - | **0.021** | 0.241 | 0.919 | 0.923 | 0.097 | 0.047 | 0.549 | $\underline{0.028}$ | 1.015 | 0.969 | 0.976 | 0.985 |
| ionosphere | - | **0.089** | 2.073 | 7.347 | 6.894 | 0.211 | 0.147 | 1.058 | $\underline{0.111}$ | 6.168 | 6.010 | 5.906 | 5.931 |
| iris | - | $\underline{0.016}$ | 0.104 | 0.161 | 0.155 | 0.096 | 0.046 | 0.587 | **0.011** | 0.129 | 0.135 | 0.132 | 0.129 |
| libras | - | **0.298** | 17.731 | 203.331 | 237.620 | 0.498 | 0.396 | 0.645 | $\underline{0.355}$ | 261.324 | 204.372 | 193.673 | 255.079 |
| parkinsons | - | $\underline{0.027}$ | 0.792 | 2.836 | 2.639 | 0.155 | 0.097 | 0.541 | **0.073** | 2.355 | 2.434 | 2.354 | 2.229 |
| planning relax | - | **0.021** | 0.381 | 0.868 | 0.816 | 0.116 | 0.064 | 0.513 | $\underline{0.038}$ | 0.918 | 0.724 | 0.693 | 0.692 |
| qsar biodegradation | - | 0.730 | 4.112 | 14.501 | 13.275 | $\underline{0.266}$ | 0.310 | 1.566 | **0.137** | 11.631 | 11.541 | 10.780 | 11.150 |
| seeds | - | **0.018** | 0.215 | 0.411 | 0.373 | 0.092 | 0.043 | 0.558 | $\underline{0.021}$ | 0.369 | 0.380 | 0.396 | 0.364 |
| tictactoe | - | 0.052 | 0.117 | 2.753 | 2.654 | 0.088 | $\underline{0.049}$ | 1.653 | **0.002** | 2.260 | 2.358 | 2.234 | 2.309 |
| wine | - | **0.022** | 0.410 | 1.332 | 1.248 | 0.121 | 0.067 | 0.612 | $\underline{0.042}$ | 1.263 | 1.202 | 1.184 | 1.120 |
| wine quality red | - | $\underline{0.075}$ | 0.683 | 1.050 | 1.071 | 0.119 | 0.150 | 1.098 | **0.035** | 1.073 | 0.956 | 0.831 | 0.879 |
| wine quality white | - | 0.453 | 1.677 | 3.549 | 3.674 | 0.443 | $\underline{0.383}$ | 3.160 | **0.041** | 4.243 | 3.194 | 2.592 | 3.263 |
| yacht hydrodynamics | - | $\underline{0.029}$ | 0.118 | 0.194 | 0.191 | 0.064 | 0.040 | 0.583 | **0.021** | 0.170 | 0.177 | 0.173 | 0.165 |
| yeast | - | $\underline{0.074}$ | 0.382 | 1.643 | 1.604 | 0.103 | 0.096 | 1.129 | **0.025** | 1.429 | 1.697 | 1.422 | 1.669 |

*Table 18.* Small Benchmark: inference time in seconds for all datasets. Best results per dataset are in **bold**, second best underlined.

| model | SMOTE | ARF | UT | FD | FF | CTGAN | TVAE | TabDDPM | XGenB-AR | XGenB-DF (x-DDPM) | XGenB-DF (v-DDPM) | XGenB-DF (x-DDIM) | XGenB-DF (v-DDIM) |
|---|---|---|---|---|---|---|---|---|---|---|---|---|---|
| airfoil self noise | $0.039_{\pm0.000}$ | $\underline{0.035}_{\pm0.000}$ | $17.895_{\pm1.361}$ | $0.322_{\pm0.005}$ | $0.153_{\pm0.001}$ | $0.023_{\pm0.000}$ | $\mathbf{0.023}_{\pm0.000}$ | $1.356_{\pm0.010}$ | $0.153_{\pm0.000}$ | $0.723_{\pm0.003}$ | $0.709_{\pm0.003}$ | $0.692_{\pm0.005}$ | $0.709_{\pm0.010}$ |
| bean | $0.318_{\pm0.011}$ | $1.713_{\pm0.022}$ | $574.660_{\pm4.101}$ | $5.602_{\pm0.096}$ | $2.596_{\pm0.047}$ | $\underline{0.142}_{\pm0.001}$ | $\mathbf{0.095}_{\pm0.000}$ | $15.054_{\pm0.058}$ | $0.907_{\pm0.003}$ | $8.309_{\pm0.106}$ | $8.613_{\pm0.053}$ | $8.137_{\pm0.221}$ | $8.070_{\pm0.108}$ |
| blood transfusion | $0.040_{\pm0.000}$ | $0.086_{\pm0.001}$ | $6.601_{\pm0.070}$ | $0.300_{\pm0.005}$ | $0.154_{\pm0.002}$ | $\mathbf{0.019}_{\pm0.000}$ | $\underline{0.027}_{\pm0.000}$ | $2.556_{\pm0.014}$ | $0.092_{\pm0.000}$ | $1.376_{\pm0.008}$ | $1.363_{\pm0.006}$ | $1.372_{\pm0.006}$ | $1.358_{\pm0.005}$ |
| breast cancer diagnostic | $\mathbf{0.022}_{\pm0.000}$ | $0.219_{\pm0.002}$ | $44.544_{\pm0.877}$ | $1.493_{\pm0.080}$ | $0.733_{\pm0.020}$ | $\underline{0.081}_{\pm0.014}$ | $0.137_{\pm0.001}$ | $4.102_{\pm0.011}$ | $0.806_{\pm0.004}$ | $1.792_{\pm0.037}$ | $2.592_{\pm0.024}$ | $2.093_{\pm0.135}$ | $2.577_{\pm0.017}$ |
| california | $\underline{0.081}_{\pm0.000}$ | $0.342_{\pm0.008}$ | $483.389_{\pm11.395}$ | $3.181_{\pm0.020}$ | $1.389_{\pm0.015}$ | $0.119_{\pm0.001}$ | $\mathbf{0.076}_{\pm0.001}$ | $22.548_{\pm0.400}$ | $0.583_{\pm0.005}$ | $3.627_{\pm0.100}$ | $4.174_{\pm0.136}$ | $3.044_{\pm0.052}$ | |
| car | $0.579_{\pm0.004}$ | $1.020_{\pm0.002}$ | $7.883_{\pm0.062}$ | $1.510_{\pm0.034}$ | $0.728_{\pm0.016}$ | $0.026_{\pm0.000}$ | $\mathbf{0.022}_{\pm0.000}$ | $6.951_{\pm0.021}$ | $0.049_{\pm0.000}$ | $5.131_{\pm0.322}$ | $5.463_{\pm0.290}$ | $5.292_{\pm0.377}$ | $5.026_{\pm0.329}$ |
| climate model crashes | $0.045_{\pm0.005}$ | $0.136_{\pm0.002}$ | $25.862_{\pm0.343}$ | $0.836_{\pm0.037}$ | $0.414_{\pm0.010}$ | $\underline{0.051}_{\pm0.000}$ | $0.085_{\pm0.001}$ | $1.414_{\pm0.024}$ | $0.164_{\pm0.001}$ | $1.536_{\pm0.171}$ | $1.498_{\pm0.147}$ | $1.507_{\pm0.335}$ | $1.496_{\pm0.182}$ |
| concrete compression | $\underline{0.040}_{\pm0.000}$ | $0.048_{\pm0.001}$ | $21.772_{\pm0.188}$ | $0.306_{\pm0.007}$ | $0.145_{\pm0.004}$ | $\mathbf{0.028}_{\pm0.000}$ | $0.059_{\pm0.004}$ | $1.284_{\pm0.016}$ | $0.234_{\pm0.001}$ | $0.742_{\pm0.004}$ | $0.752_{\pm0.015}$ | $0.707_{\pm0.002}$ | $0.722_{\pm0.005}$ |
| concrete slump | $0.039_{\pm0.000}$ | $0.030_{\pm0.000}$ | $1.959_{\pm0.017}$ | $0.220_{\pm0.002}$ | $0.122_{\pm0.001}$ | $\mathbf{0.023}_{\pm0.000}$ | $\underline{0.027}_{\pm0.000}$ | $7.164_{\pm0.656}$ | $0.180_{\pm0.001}$ | $0.706_{\pm0.042}$ | $0.697_{\pm0.009}$ | $0.702_{\pm0.002}$ | $0.694_{\pm0.001}$ |
| congress | $0.287_{\pm0.001}$ | $0.660_{\pm0.010}$ | $4.736_{\pm0.026}$ | $1.535_{\pm0.079}$ | $0.731_{\pm0.022}$ | $\underline{0.040}_{\pm0.000}$ | $\mathbf{0.038}_{\pm0.000}$ | $11.621_{\pm0.027}$ | $0.062_{\pm0.000}$ | $2.663_{\pm0.026}$ | $2.625_{\pm0.039}$ | $2.723_{\pm0.056}$ | $2.607_{\pm0.042}$ |
| connectionist bench sonar | $\mathbf{0.033}_{\pm0.000}$ | $0.300_{\pm0.002}$ | $33.308_{\pm0.940}$ | $2.586_{\pm0.098}$ | $1.256_{\pm0.042}$ | $\underline{0.148}_{\pm0.002}$ | $0.201_{\pm0.001}$ | $1.655_{\pm0.064}$ | $1.584_{\pm0.008}$ | $2.901_{\pm0.025}$ | $3.732_{\pm0.096}$ | $2.787_{\pm0.059}$ | $3.652_{\pm0.042}$ |
| connectionist bench vowel | $0.178_{\pm0.001}$ | $0.136_{\pm0.002}$ | $26.553_{\pm0.719}$ | $2.873_{\pm0.078}$ | $1.476_{\pm0.024}$ | $\mathbf{0.034}_{\pm0.000}$ | $\underline{0.065}_{\pm0.003}$ | $1.307_{\pm0.008}$ | $0.269_{\pm0.001}$ | $7.757_{\pm0.016}$ | $7.701_{\pm0.031}$ | $7.689_{\pm0.045}$ | $7.655_{\pm0.024}$ |
| ecoli | - | $0.063_{\pm0.001}$ | $4.659_{\pm0.255}$ | $1.001_{\pm0.043}$ | $0.525_{\pm0.024}$ | $\mathbf{0.024}_{\pm0.000}$ | $\underline{0.032}_{\pm0.000}$ | $2.648_{\pm0.016}$ | $0.153_{\pm0.002}$ | $5.348_{\pm0.343}$ | $5.260_{\pm0.330}$ | $5.089_{\pm0.460}$ | $5.105_{\pm0.459}$ |
| glass | $0.076_{\pm0.004}$ | $0.057_{\pm0.001}$ | $4.525_{\pm0.101}$ | $0.973_{\pm0.015}$ | $0.523_{\pm0.013}$ | $\mathbf{0.029}_{\pm0.000}$ | $\underline{0.036}_{\pm0.000}$ | $12.733_{\pm0.806}$ | $0.217_{\pm0.001}$ | $4.268_{\pm0.005}$ | $4.204_{\pm0.008}$ | $4.128_{\pm0.004}$ | $4.145_{\pm0.007}$ |
| ionosphere | $\underline{0.089}_{\pm0.001}$ | $0.213_{\pm0.002}$ | $28.830_{\pm0.673}$ | $1.548_{\pm0.075}$ | $0.754_{\pm0.001}$ | $\mathbf{0.086}_{\pm0.001}$ | $0.127_{\pm0.001}$ | $2.833_{\pm0.010}$ | $0.842_{\pm0.002}$ | $2.574_{\pm0.023}$ | $2.650_{\pm0.029}$ | $1.954_{\pm0.082}$ | $2.613_{\pm0.040}$ |
| iris | $0.054_{\pm0.001}$ | $0.036_{\pm0.001}$ | $1.428_{\pm0.020}$ | $0.386_{\pm0.021}$ | $0.194_{\pm0.007}$ | $\mathbf{0.017}_{\pm0.000}$ | $\underline{0.020}_{\pm0.001}$ | $2.494_{\pm0.401}$ | $0.088_{\pm0.001}$ | $2.089_{\pm0.008}$ | $2.058_{\pm0.016}$ | $2.022_{\pm0.007}$ | $2.047_{\pm0.006}$ |
| libras | $0.059_{\pm0.001}$ | $0.699_{\pm0.012}$ | $84.474_{\pm2.128}$ | $15.016_{\pm1.254}$ | $9.292_{\pm0.635}$ | $\underline{0.218}_{\pm0.001}$ | $0.334_{\pm0.002}$ | $1.717_{\pm0.041}$ | $2.487_{\pm0.006}$ | $33.046_{\pm0.635}$ | $29.166_{\pm0.408}$ | $38.363_{\pm0.200}$ | |
| parkinsons | $\mathbf{0.019}_{\pm0.000}$ | $0.103_{\pm0.001}$ | $11.297_{\pm1.005}$ | $0.938_{\pm0.054}$ | $0.467_{\pm0.022}$ | $\underline{0.060}_{\pm0.000}$ | $0.078_{\pm0.001}$ | $2.733_{\pm0.435}$ | $0.552_{\pm0.003}$ | $1.494_{\pm0.003}$ | $1.460_{\pm0.004}$ | $1.441_{\pm0.007}$ | $1.453_{\pm0.009}$ |
| planning relax | $\underline{0.043}_{\pm0.000}$ | $0.065_{\pm0.001}$ | $5.599_{\pm0.069}$ | $0.539_{\pm0.014}$ | $0.283_{\pm0.007}$ | $\mathbf{0.035}_{\pm0.000}$ | $0.044_{\pm0.000}$ | $2.642_{\pm0.014}$ | $0.296_{\pm0.001}$ | $1.430_{\pm0.002}$ | $1.422_{\pm0.003}$ | $1.414_{\pm0.005}$ | $1.414_{\pm0.003}$ |
| qsar biodegradation | $\underline{0.118}_{\pm0.001}$ | $0.695_{\pm0.006}$ | $67.593_{\pm1.279}$ | $2.118_{\pm0.048}$ | $1.036_{\pm0.038}$ | $\mathbf{0.114}_{\pm0.002}$ | $0.238_{\pm0.001}$ | $3.409_{\pm0.168}$ | $1.116_{\pm0.004}$ | $2.831_{\pm0.023}$ | $2.820_{\pm0.064}$ | $2.758_{\pm0.039}$ | $2.781_{\pm0.011}$ |
| seeds | $0.056_{\pm0.001}$ | $0.050_{\pm0.001}$ | $3.796_{\pm0.034}$ | $0.555_{\pm0.004}$ | $0.293_{\pm0.008}$ | $\mathbf{0.024}_{\pm0.000}$ | $\underline{0.029}_{\pm0.000}$ | $2.796_{\pm0.011}$ | $0.164_{\pm0.001}$ | $2.101_{\pm0.003}$ | $2.059_{\pm0.004}$ | $2.036_{\pm0.002}$ | $2.043_{\pm0.010}$ |
| tictactoe | $0.374_{\pm0.002}$ | $0.817_{\pm0.014}$ | $6.186_{\pm0.038}$ | $0.993_{\pm0.025}$ | $0.483_{\pm0.011}$ | $\underline{0.029}_{\pm0.001}$ | $\mathbf{0.026}_{\pm0.000}$ | $8.328_{\pm0.031}$ | $0.050_{\pm0.000}$ | $2.674_{\pm0.092}$ | $3.027_{\pm0.098}$ | $3.175_{\pm0.064}$ | $2.579_{\pm0.047}$ |
| wine | $0.058_{\pm0.000}$ | $0.067_{\pm0.001}$ | $5.976_{\pm0.034}$ | $0.841_{\pm0.013}$ | $0.428_{\pm0.025}$ | $\mathbf{0.039}_{\pm0.000}$ | $\underline{0.049}_{\pm0.001}$ | $10.684_{\pm0.622}$ | $0.318_{\pm0.002}$ | $2.155_{\pm0.005}$ | $2.119_{\pm0.004}$ | $2.092_{\pm0.015}$ | $2.100_{\pm0.005}$ |
| wine quality red | $\underline{0.041}_{\pm0.000}$ | $0.071_{\pm0.001}$ | $43.026_{\pm1.178}$ | $0.513_{\pm0.004}$ | $0.251_{\pm0.003}$ | $\mathbf{0.036}_{\pm0.000}$ | $0.036_{\pm0.000}$ | $2.457_{\pm0.015}$ | $0.304_{\pm0.001}$ | $0.757_{\pm0.006}$ | $0.778_{\pm0.067}$ | $1.086_{\pm0.049}$ | $0.728_{\pm0.015}$ |
| wine quality white | $\underline{0.052}_{\pm0.006}$ | $0.160_{\pm0.001}$ | $145.033_{\pm0.946}$ | $1.066_{\pm0.011}$ | $0.488_{\pm0.010}$ | $0.056_{\pm0.000}$ | $\mathbf{0.048}_{\pm0.000}$ | $4.855_{\pm0.032}$ | $0.418_{\pm0.002}$ | $1.402_{\pm0.014}$ | $1.409_{\pm0.023}$ | $1.612_{\pm0.051}$ | $1.292_{\pm0.021}$ |
| yacht hydrodynamics | $0.039_{\pm0.000}$ | $0.030_{\pm0.000}$ | $3.850_{\pm0.188}$ | $0.227_{\pm0.006}$ | $0.108_{\pm0.005}$ | $\mathbf{0.021}_{\pm0.000}$ | $\underline{0.029}_{\pm0.000}$ | $2.622_{\pm0.081}$ | $0.161_{\pm0.001}$ | $0.718_{\pm0.002}$ | $0.699_{\pm0.002}$ | $0.691_{\pm0.001}$ | $0.703_{\pm0.002}$ |
| yeast | $0.151_{\pm0.004}$ | $0.172_{\pm0.002}$ | $24.122_{\pm0.558}$ | $1.856_{\pm0.078}$ | $0.937_{\pm0.022}$ | $\underline{0.033}_{\pm0.000}$ | $\mathbf{0.031}_{\pm0.000}$ | $2.704_{\pm0.080}$ | $0.220_{\pm0.001}$ | $6.996_{\pm0.153}$ | $6.879_{\pm0.148}$ | $6.818_{\pm0.011}$ | $6.870_{\pm0.025}$ |

*Table 19.* Big Benchmark: Shape metric scores for all datasets. Average and standard devation ($\mu \pm \sigma$) over 20 sampling seeds. Best results per dataset are in **bold**, second best underlined.

| model | SMOTE | ARF | CTGAN | TVAE | TabDDPM | TabSyn | XGenB-AR |
|---|---|---|---|---|---|---|---|
| acsincome | - | $0.977_{\pm 0.001}$ | $0.856_{\pm 0.001}$ | $0.855_{\pm 0.001}$ | - | $\underline{0.978}_{\pm 0.001}$ | $\mathbf{0.987}_{\pm 0.001}$ |
| adult | $0.925_{\pm 0.000}$ | $0.956_{\pm 0.001}$ | $0.842_{\pm 0.001}$ | $0.913_{\pm 0.001}$ | $0.987_{\pm 0.001}$ | $\underline{0.990}_{\pm 0.000}$ | $\mathbf{0.994}_{\pm 0.001}$ |
| bank | $0.961_{\pm 0.000}$ | $0.972_{\pm 0.001}$ | $0.895_{\pm 0.000}$ | $0.887_{\pm 0.001}$ | $0.980_{\pm 0.001}$ | $\underline{0.990}_{\pm 0.000}$ | $\mathbf{0.992}_{\pm 0.001}$ |
| beijing | $0.978_{\pm 0.000}$ | $0.974_{\pm 0.001}$ | $0.917_{\pm 0.001}$ | $0.889_{\pm 0.001}$ | $0.985_{\pm 0.001}$ | $\underline{0.992}_{\pm 0.001}$ | $\mathbf{0.993}_{\pm 0.001}$ |
| churn | $0.972_{\pm 0.001}$ | $0.966_{\pm 0.002}$ | $0.836_{\pm 0.003}$ | $0.938_{\pm 0.002}$ | $0.922_{\pm 0.003}$ | $\underline{0.975}_{\pm 0.003}$ | $\mathbf{0.982}_{\pm 0.002}$ |
| covertype | - | $0.995_{\pm 0.000}$ | $0.955_{\pm 0.000}$ | $0.980_{\pm 0.000}$ | $\underline{0.996}_{\pm 0.000}$ | $0.990_{\pm 0.000}$ | $\mathbf{0.997}_{\pm 0.000}$ |
| default | $0.955_{\pm 0.001}$ | $0.953_{\pm 0.001}$ | $0.854_{\pm 0.001}$ | $0.901_{\pm 0.001}$ | $0.983_{\pm 0.001}$ | $\underline{0.986}_{\pm 0.001}$ | $\mathbf{0.991}_{\pm 0.001}$ |
| diabetes | $0.955_{\pm 0.000}$ | $\underline{0.980}_{\pm 0.000}$ | $0.923_{\pm 0.000}$ | $0.898_{\pm 0.000}$ | - | $0.942_{\pm 0.000}$ | $\mathbf{0.995}_{\pm 0.000}$ |
| lending | $0.939_{\pm 0.001}$ | $0.944_{\pm 0.001}$ | $0.876_{\pm 0.001}$ | $0.854_{\pm 0.001}$ | $0.469_{\pm 0.001}$ | $\underline{0.959}_{\pm 0.001}$ | $\mathbf{0.984}_{\pm 0.001}$ |
| news | $0.915_{\pm 0.000}$ | $0.899_{\pm 0.000}$ | $0.837_{\pm 0.000}$ | $0.866_{\pm 0.000}$ | $0.952_{\pm 0.001}$ | $\underline{0.977}_{\pm 0.000}$ | $\mathbf{0.991}_{\pm 0.000}$ |
| nmes | $0.967_{\pm 0.001}$ | $0.956_{\pm 0.001}$ | $0.890_{\pm 0.002}$ | $0.854_{\pm 0.002}$ | $0.939_{\pm 0.002}$ | $\underline{0.980}_{\pm 0.001}$ | $\mathbf{0.990}_{\pm 0.001}$ |

*Table 20.* Big Benchmark: Trend metric scores for all datasets. Average and standard devation ($\mu \pm \sigma$) over 20 sampling seeds. Best results per dataset are in **bold**, second best underlined.

| model | SMOTE | ARF | CTGAN | TVAE | TabDDPM | TabSyn | XGenB-AR |
|---|---|---|---|---|---|---|---|
| acsincome | - | $0.712_{\pm 0.018}$ | $0.710_{\pm 0.003}$ | $0.661_{\pm 0.010}$ | - | $\underline{0.941}_{\pm 0.002}$ | $\mathbf{0.946}_{\pm 0.002}$ |
| adult | $0.859_{\pm 0.004}$ | $0.636_{\pm 0.011}$ | $0.782_{\pm 0.005}$ | $0.838_{\pm 0.025}$ | $0.963_{\pm 0.001}$ | $\underline{0.977}_{\pm 0.001}$ | $\mathbf{0.980}_{\pm 0.003}$ |
| bank | $0.931_{\pm 0.003}$ | $0.572_{\pm 0.010}$ | $0.759_{\pm 0.006}$ | $0.810_{\pm 0.003}$ | $0.956_{\pm 0.001}$ | $\mathbf{0.980}_{\pm 0.001}$ | $\underline{0.978}_{\pm 0.002}$ |
| beijing | $0.989_{\pm 0.001}$ | $0.892_{\pm 0.003}$ | $0.901_{\pm 0.001}$ | $0.912_{\pm 0.003}$ | $0.985_{\pm 0.001}$ | $\mathbf{0.994}_{\pm 0.001}$ | $\underline{0.990}_{\pm 0.001}$ |
| churn | $\underline{0.961}_{\pm 0.002}$ | $0.735_{\pm 0.018}$ | $0.775_{\pm 0.007}$ | $0.901_{\pm 0.004}$ | $0.885_{\pm 0.004}$ | $\underline{0.961}_{\pm 0.002}$ | $\mathbf{0.965}_{\pm 0.004}$ |
| covertype | - | $0.819_{\pm 0.013}$ | $0.900_{\pm 0.004}$ | $0.920_{\pm 0.008}$ | $0.973_{\pm 0.003}$ | $\underline{0.979}_{\pm 0.003}$ | $\mathbf{0.988}_{\pm 0.004}$ |
| default | $0.883_{\pm 0.009}$ | $0.599_{\pm 0.015}$ | $0.740_{\pm 0.003}$ | $0.832_{\pm 0.012}$ | $0.909_{\pm 0.002}$ | $\mathbf{0.960}_{\pm 0.009}$ | $\underline{0.920}_{\pm 0.019}$ |
| diabetes | $\underline{0.917}_{\pm 0.001}$ | $0.725_{\pm 0.006}$ | $0.813_{\pm 0.004}$ | $0.802_{\pm 0.003}$ | - | $0.893_{\pm 0.001}$ | $\mathbf{0.978}_{\pm 0.002}$ |
| lending | $0.915_{\pm 0.005}$ | $0.699_{\pm 0.005}$ | $0.840_{\pm 0.002}$ | $0.762_{\pm 0.004}$ | $0.653_{\pm 0.000}$ | $\underline{0.936}_{\pm 0.002}$ | $\mathbf{0.952}_{\pm 0.002}$ |
| news | $0.936_{\pm 0.006}$ | $0.704_{\pm 0.004}$ | $0.845_{\pm 0.002}$ | $0.837_{\pm 0.004}$ | $0.910_{\pm 0.001}$ | $\mathbf{0.978}_{\pm 0.001}$ | $\underline{0.973}_{\pm 0.007}$ |
| nmes | $0.945_{\pm 0.004}$ | $0.586_{\pm 0.018}$ | $0.733_{\pm 0.014}$ | $0.807_{\pm 0.009}$ | $0.901_{\pm 0.002}$ | $\underline{0.968}_{\pm 0.005}$ | $\mathbf{0.971}_{\pm 0.005}$ |

*Table 21.* Big Benchmark: Detection metric scores for all datasets. Average and standard devation ($\mu \pm \sigma$) over 20 sampling seeds. Best results per dataset are in **bold**, second best underlined.

| model | SMOTE | ARF | CTGAN | TVAE | TabDDPM | TabSyn | XGenB-AR |
|---|---|---|---|---|---|---|---|
| acsincome | - | $0.842_{\pm 0.003}$ | $0.964_{\pm 0.001}$ | $0.974_{\pm 0.000}$ | - | $\mathbf{0.611}_{\pm 0.003}$ | $\underline{0.689}_{\pm 0.003}$ |
| adult | $0.956_{\pm 0.001}$ | $0.956_{\pm 0.002}$ | $0.998_{\pm 0.000}$ | $0.978_{\pm 0.001}$ | $\underline{0.619}_{\pm 0.004}$ | $0.668_{\pm 0.004}$ | $\mathbf{0.614}_{\pm 0.004}$ |
| bank | $0.917_{\pm 0.003}$ | $0.927_{\pm 0.002}$ | $1.000_{\pm 0.000}$ | $0.998_{\pm 0.000}$ | $0.881_{\pm 0.004}$ | $\underline{0.866}_{\pm 0.003}$ | $\mathbf{0.817}_{\pm 0.003}$ |
| beijing | $0.938_{\pm 0.004}$ | $0.985_{\pm 0.002}$ | $0.997_{\pm 0.000}$ | $0.998_{\pm 0.000}$ | $0.876_{\pm 0.003}$ | $\mathbf{0.768}_{\pm 0.007}$ | $\underline{0.813}_{\pm 0.005}$ |
| churn | $\underline{0.778}_{\pm 0.011}$ | $0.900_{\pm 0.011}$ | $0.999_{\pm 0.001}$ | $0.977_{\pm 0.003}$ | $0.932_{\pm 0.007}$ | $0.866_{\pm 0.011}$ | $\mathbf{0.739}_{\pm 0.016}$ |
| covertype | - | $0.975_{\pm 0.001}$ | $1.000_{\pm 0.000}$ | $0.998_{\pm 0.000}$ | $\mathbf{0.800}_{\pm 0.005}$ | $\underline{0.844}_{\pm 0.003}$ | $0.873_{\pm 0.003}$ |
| default | $0.927_{\pm 0.002}$ | $0.996_{\pm 0.000}$ | $1.000_{\pm 0.000}$ | $0.999_{\pm 0.000}$ | $0.915_{\pm 0.004}$ | $\mathbf{0.865}_{\pm 0.004}$ | $\underline{0.881}_{\pm 0.005}$ |
| diabetes | $\underline{0.896}_{\pm 0.001}$ | $0.981_{\pm 0.001}$ | $0.985_{\pm 0.000}$ | $0.998_{\pm 0.000}$ | - | $0.955_{\pm 0.001}$ | $\mathbf{0.774}_{\pm 0.002}$ |
| lending | $0.981_{\pm 0.003}$ | $0.999_{\pm 0.000}$ | $1.000_{\pm 0.000}$ | $1.000_{\pm 0.000}$ | $1.000_{\pm 0.000}$ | $\underline{0.933}_{\pm 0.006}$ | $\mathbf{0.652}_{\pm 0.015}$ |
| news | $1.000_{\pm 0.000}$ | $1.000_{\pm 0.000}$ | $1.000_{\pm 0.000}$ | $1.000_{\pm 0.000}$ | $0.995_{\pm 0.000}$ | $\underline{0.987}_{\pm 0.001}$ | $\mathbf{0.923}_{\pm 0.004}$ |
| nmes | $0.652_{\pm 0.008}$ | $0.845_{\pm 0.009}$ | $0.927_{\pm 0.004}$ | $0.987_{\pm 0.001}$ | $0.848_{\pm 0.005}$ | $\underline{0.633}_{\pm 0.008}$ | $\mathbf{0.511}_{\pm 0.012}$ |

*Table 22.* Big Benchmark: $\alpha$-Precision metric scores for all datasets. Average and standard deviation ($\mu \pm \sigma$) over 20 sampling seeds. Best results per dataset are in **bold**, second best underlined.

| model | SMOTE | ARF | CTGAN | TVAE | TabDDPM | TabSyn | XGenB-AR |
|---|---|---|---|---|---|---|---|
| acsincome | - | $\underline{0.991}_{\pm 0.003}$ | $0.846_{\pm 0.004}$ | $0.859_{\pm 0.004}$ | - | $0.984_{\pm 0.003}$ | $\mathbf{0.992}_{\pm 0.003}$ |
| adult | $0.617_{\pm 0.001}$ | $0.974_{\pm 0.003}$ | $0.696_{\pm 0.003}$ | $0.847_{\pm 0.003}$ | $0.945_{\pm 0.003}$ | $\underline{0.991}_{\pm 0.002}$ | $\mathbf{0.997}_{\pm 0.001}$ |
| bank | $0.810_{\pm 0.002}$ | $\mathbf{0.995}_{\pm 0.001}$ | $0.708_{\pm 0.002}$ | $0.892_{\pm 0.004}$ | $0.959_{\pm 0.004}$ | $\underline{0.993}_{\pm 0.003}$ | $0.984_{\pm 0.004}$ |
| beijing | $0.916_{\pm 0.002}$ | $0.957_{\pm 0.003}$ | $0.928_{\pm 0.003}$ | $\underline{0.980}_{\pm 0.001}$ | $0.953_{\pm 0.004}$ | $\mathbf{0.988}_{\pm 0.002}$ | $\mathbf{0.988}_{\pm 0.002}$ |
| churn | $0.957_{\pm 0.007}$ | $\mathbf{0.975}_{\pm 0.005}$ | $0.715_{\pm 0.007}$ | $0.895_{\pm 0.012}$ | $0.938_{\pm 0.005}$ | $0.951_{\pm 0.010}$ | $\underline{0.958}_{\pm 0.017}$ |
| covertype | - | $0.940_{\pm 0.002}$ | $0.753_{\pm 0.002}$ | $0.787_{\pm 0.004}$ | $\underline{0.957}_{\pm 0.004}$ | $\mathbf{0.971}_{\pm 0.004}$ | $0.918_{\pm 0.002}$ |
| default | $0.944_{\pm 0.002}$ | $0.965_{\pm 0.002}$ | $0.729_{\pm 0.002}$ | $0.859_{\pm 0.004}$ | $0.983_{\pm 0.003}$ | $\underline{0.985}_{\pm 0.004}$ | $\mathbf{0.989}_{\pm 0.002}$ |
| diabetes | $0.489_{\pm 0.001}$ | $\underline{0.982}_{\pm 0.002}$ | $0.790_{\pm 0.003}$ | $0.448_{\pm 0.002}$ | - | $0.949_{\pm 0.002}$ | $\mathbf{0.996}_{\pm 0.001}$ |
| lending | $0.651_{\pm 0.004}$ | $0.930_{\pm 0.006}$ | $0.757_{\pm 0.005}$ | $0.663_{\pm 0.007}$ | $0.000_{\pm 0.000}$ | $\mathbf{0.973}_{\pm 0.006}$ | $\underline{0.972}_{\pm 0.007}$ |
| news | $0.399_{\pm 0.002}$ | $0.934_{\pm 0.002}$ | $0.892_{\pm 0.002}$ | $0.571_{\pm 0.004}$ | $\mathbf{0.962}_{\pm 0.003}$ | $0.884_{\pm 0.004}$ | $\underline{0.955}_{\pm 0.004}$ |
| nmes | $0.811_{\pm 0.005}$ | $0.934_{\pm 0.010}$ | $0.778_{\pm 0.008}$ | $0.606_{\pm 0.012}$ | $0.868_{\pm 0.014}$ | $\underline{0.975}_{\pm 0.007}$ | $\mathbf{0.983}_{\pm 0.006}$ |

*Table 23.* Big Benchmark: $\beta$-Recall metric scores for all datasets. Average and standard deviation ($\mu \pm \sigma$) over 20 sampling seeds. Best results per dataset are in **bold**, second best underlined.

| model | SMOTE | ARF | CTGAN | TVAE | TabDDPM | TabSyn | XGenB-AR |
|---|---|---|---|---|---|---|---|
| acsincome | - | $\mathbf{0.460}_{\pm 0.003}$ | $0.233_{\pm 0.002}$ | $0.323_{\pm 0.003}$ | - | $\underline{0.449}_{\pm 0.002}$ | $0.436_{\pm 0.002}$ |
| adult | $0.300_{\pm 0.001}$ | $0.297_{\pm 0.003}$ | $0.246_{\pm 0.002}$ | $0.386_{\pm 0.003}$ | $\mathbf{0.509}_{\pm 0.003}$ | $0.467_{\pm 0.003}$ | $\underline{0.478}_{\pm 0.002}$ |
| bank | $\mathbf{0.461}_{\pm 0.002}$ | $0.354_{\pm 0.003}$ | $0.217_{\pm 0.002}$ | $0.169_{\pm 0.002}$ | $\underline{0.440}_{\pm 0.004}$ | $0.417_{\pm 0.003}$ | $0.393_{\pm 0.003}$ |
| beijing | $\mathbf{0.613}_{\pm 0.002}$ | $0.101_{\pm 0.002}$ | $0.053_{\pm 0.001}$ | $0.051_{\pm 0.001}$ | $0.098_{\pm 0.001}$ | $\underline{0.134}_{\pm 0.002}$ | $0.113_{\pm 0.002}$ |
| churn | $\mathbf{0.570}_{\pm 0.008}$ | $0.146_{\pm 0.010}$ | $0.003_{\pm 0.001}$ | $0.062_{\pm 0.004}$ | $0.094_{\pm 0.008}$ | $0.179_{\pm 0.010}$ | $\underline{0.257}_{\pm 0.012}$ |
| covertype | - | $0.105_{\pm 0.001}$ | $0.002_{\pm 0.000}$ | $0.042_{\pm 0.001}$ | $\mathbf{0.156}_{\pm 0.004}$ | $\underline{0.146}_{\pm 0.003}$ | $0.109_{\pm 0.002}$ |
| default | $\mathbf{0.683}_{\pm 0.003}$ | $0.299_{\pm 0.004}$ | $0.160_{\pm 0.002}$ | $0.228_{\pm 0.002}$ | $0.449_{\pm 0.004}$ | $\underline{0.455}_{\pm 0.004}$ | $0.426_{\pm 0.003}$ |
| diabetes | $\mathbf{0.552}_{\pm 0.002}$ | $0.299_{\pm 0.002}$ | $0.143_{\pm 0.002}$ | $0.340_{\pm 0.002}$ | - | $0.208_{\pm 0.002}$ | $\underline{0.389}_{\pm 0.003}$ |
| lending | $\mathbf{0.630}_{\pm 0.003}$ | $0.212_{\pm 0.006}$ | $0.049_{\pm 0.003}$ | $0.231_{\pm 0.004}$ | $0.000_{\pm 0.000}$ | $0.399_{\pm 0.009}$ | $\underline{0.479}_{\pm 0.010}$ |
| news | $0.288_{\pm 0.002}$ | $0.006_{\pm 0.001}$ | $0.023_{\pm 0.001}$ | $0.009_{\pm 0.001}$ | $\underline{0.331}_{\pm 0.004}$ | $0.296_{\pm 0.003}$ | $\mathbf{0.371}_{\pm 0.003}$ |
| nmes | $\mathbf{0.691}_{\pm 0.011}$ | $0.389_{\pm 0.007}$ | $0.284_{\pm 0.008}$ | $0.240_{\pm 0.007}$ | $0.461_{\pm 0.009}$ | $\underline{0.516}_{\pm 0.012}$ | $0.501_{\pm 0.012}$ |

*Table 24.* Big Benchmark: MLE ($R^2$) metric scores for all datasets. Average and standard deviation ($\mu \pm \sigma$) over 20 sampling seeds. Best results per dataset are in **bold**, second best underlined. Real indicates MLE results when training on real data.

| model | Real | SMOTE | ARF | CTGAN | TVAE | TabDDPM | TabSyn | XGenB-AR |
|---|---|---|---|---|---|---|---|---|
| acsincome | $0.390_{\pm 0.007}$ | - | $\mathbf{0.376}_{\pm 0.008}$ | $0.212_{\pm 0.005}$ | $0.285_{\pm 0.009}$ | - | $\underline{0.362}_{\pm 0.007}$ | $0.334_{\pm 0.007}$ |
| beijing | $0.803_{\pm 0.000}$ | $\mathbf{0.671}_{\pm 0.005}$ | $0.470_{\pm 0.010}$ | $0.041_{\pm 0.019}$ | $0.084_{\pm 0.034}$ | $0.447_{\pm 0.015}$ | $\underline{0.572}_{\pm 0.009}$ | $0.407_{\pm 0.011}$ |
| lending | $0.999_{\pm 0.000}$ | $\mathbf{0.997}_{\pm 0.000}$ | $0.885_{\pm 0.010}$ | $-0.094_{\pm 0.061}$ | $0.940_{\pm 0.006}$ | $-1.623_{\pm 1.042}$ | $0.984_{\pm 0.002}$ | $\underline{0.992}_{\pm 0.001}$ |
| news | $0.022_{\pm 0.004}$ | $0.009_{\pm 0.025}$ | $\mathbf{0.012}_{\pm 0.009}$ | $0.006_{\pm 0.003}$ | $-0.035_{\pm 0.001}$ | $-0.573_{\pm 0.077}$ | $-0.000_{\pm 0.020}$ | $\underline{0.010}_{\pm 0.016}$ |
| nmes | $0.177_{\pm 0.000}$ | $0.103_{\pm 0.035}$ | $0.143_{\pm 0.016}$ | $-0.071_{\pm 0.023}$ | $0.069_{\pm 0.010}$ | $0.109_{\pm 0.033}$ | $\mathbf{0.149}_{\pm 0.015}$ | $\underline{0.148}_{\pm 0.019}$ |

*Table 25.* Big Benchmark: MLE (ROCAUC) metric scores for all datasets. Average and standard deviation ($\mu \pm \sigma$) over 20 sampling seeds. Best results per dataset are in **bold**, second best underlined. Real indicates MLE results when training on real data.

| model | Real | SMOTE | ARF | CTGAN | TVAE | TabDDPM | TabSyn | XGenB-AR |
|---|---|---|---|---|---|---|---|---|
| adult | $0.929_{\pm 0.000}$ | $0.912_{\pm 0.001}$ | $0.908_{\pm 0.001}$ | $0.891_{\pm 0.002}$ | $0.880_{\pm 0.002}$ | $0.911_{\pm 0.001}$ | $\underline{0.913}_{\pm 0.001}$ | $\mathbf{0.914}_{\pm 0.002}$ |
| bank | $0.951_{\pm 0.002}$ | $\underline{0.945}_{\pm 0.001}$ | $0.939_{\pm 0.001}$ | $0.878_{\pm 0.007}$ | $0.931_{\pm 0.003}$ | $0.941_{\pm 0.002}$ | $0.938_{\pm 0.002}$ | $\mathbf{0.947}_{\pm 0.001}$ |
| churn | $0.988_{\pm 0.001}$ | $\mathbf{0.980}_{\pm 0.002}$ | $0.949_{\pm 0.012}$ | $0.843_{\pm 0.019}$ | $0.868_{\pm 0.014}$ | $0.877_{\pm 0.025}$ | $0.948_{\pm 0.015}$ | $\underline{0.950}_{\pm 0.011}$ |
| covertype | $0.964_{\pm 0.001}$ | - | $\mathbf{0.898}_{\pm 0.002}$ | $0.597_{\pm 0.010}$ | $0.824_{\pm 0.002}$ | $\underline{0.874}_{\pm 0.001}$ | $0.852_{\pm 0.001}$ | $0.834_{\pm 0.003}$ |
| default | $0.774_{\pm 0.002}$ | $\underline{0.760}_{\pm 0.003}$ | $0.758_{\pm 0.003}$ | $0.701_{\pm 0.005}$ | $0.723_{\pm 0.004}$ | $0.759_{\pm 0.003}$ | $\mathbf{0.761}_{\pm 0.002}$ | $0.757_{\pm 0.003}$ |
| diabetes | $0.699_{\pm 0.002}$ | $\mathbf{0.688}_{\pm 0.002}$ | $\underline{0.678}_{\pm 0.003}$ | $0.573_{\pm 0.003}$ | $0.659_{\pm 0.001}$ | - | $0.646_{\pm 0.004}$ | $0.675_{\pm 0.002}$ |

*Table 26.* Big Benchmark: MLE (F1) metric scores for all datasets. Average and standard deviation ($\mu \pm \sigma$) over 20 sampling seeds. Best results per dataset are in **bold**, second best underlined. Real indicates MLE results when training on real data.

| model | Real | SMOTE | ARF | CTGAN | TVAE | TabDDPM | TabSyn | XGenB-AR |
|---|---|---|---|---|---|---|---|---|
| adult | $0.716_{\pm 0.002}$ | $\mathbf{0.687}_{\pm 0.003}$ | $0.649_{\pm 0.008}$ | $0.646_{\pm 0.008}$ | $0.620_{\pm 0.006}$ | $0.678_{\pm 0.005}$ | $0.675_{\pm 0.004}$ | $\underline{0.682}_{\pm 0.004}$ |
| bank | $0.605_{\pm 0.001}$ | $\mathbf{0.574}_{\pm 0.010}$ | $0.456_{\pm 0.016}$ | $0.515_{\pm 0.012}$ | $0.540_{\pm 0.014}$ | $0.534_{\pm 0.010}$ | $0.542_{\pm 0.012}$ | $\underline{0.565}_{\pm 0.008}$ |
| churn | $0.877_{\pm 0.001}$ | $\mathbf{0.840}_{\pm 0.021}$ | $0.698_{\pm 0.045}$ | $0.535_{\pm 0.037}$ | $0.582_{\pm 0.042}$ | $0.340_{\pm 0.094}$ | $0.705_{\pm 0.031}$ | $\underline{0.713}_{\pm 0.042}$ |
| covertype | $0.898_{\pm 0.003}$ | - | $\mathbf{0.814}_{\pm 0.002}$ | $0.347_{\pm 0.024}$ | $0.737_{\pm 0.003}$ | $\underline{0.789}_{\pm 0.002}$ | $0.766_{\pm 0.003}$ | $0.764_{\pm 0.003}$ |
| default | $0.448_{\pm 0.004}$ | $\mathbf{0.451}_{\pm 0.007}$ | $0.368_{\pm 0.014}$ | $0.247_{\pm 0.019}$ | $0.338_{\pm 0.009}$ | $\underline{0.439}_{\pm 0.008}$ | $0.416_{\pm 0.011}$ | $0.413_{\pm 0.013}$ |
| diabetes | $0.588_{\pm 0.003}$ | $\underline{0.574}_{\pm 0.004}$ | $0.508_{\pm 0.007}$ | $0.085_{\pm 0.011}$ | $\mathbf{0.586}_{\pm 0.004}$ | - | $0.435_{\pm 0.009}$ | $0.560_{\pm 0.004}$ |

*Table 27.* Big Benchmark: MLE (RMSE) metric scores for all datasets. Average and standard deviation ($\mu \pm \sigma$) over 20 sampling seeds. Best results per dataset are in **bold**, second best underlined. Real indicates MLE results when training on real data.

| model | Real | SMOTE | ARF | CTGAN | TVAE | TabDDPM | TabSyn | XGenB-AR |
|---|---|---|---|---|---|---|---|---|
| acsincome | $0.774_{\pm 0.014}$ | - | $\mathbf{0.783}_{\pm 0.016}$ | $0.880_{\pm 0.017}$ | $0.838_{\pm 0.017}$ | - | $\underline{0.792}_{\pm 0.015}$ | $0.809_{\pm 0.015}$ |
| beijing | $0.441_{\pm 0.000}$ | $\mathbf{0.571}_{\pm 0.005}$ | $0.724_{\pm 0.007}$ | $0.974_{\pm 0.010}$ | $0.952_{\pm 0.017}$ | $0.740_{\pm 0.010}$ | $\underline{0.651}_{\pm 0.007}$ | $0.766_{\pm 0.007}$ |
| lending | $0.030_{\pm 0.000}$ | $\mathbf{0.056}_{\pm 0.003}$ | $0.338_{\pm 0.015}$ | $1.043_{\pm 0.029}$ | $0.245_{\pm 0.013}$ | $1.588_{\pm 0.311}$ | $0.126_{\pm 0.006}$ | $\underline{0.090}_{\pm 0.006}$ |
| news | $0.998_{\pm 0.002}$ | $\underline{1.004}_{\pm 0.013}$ | $\mathbf{1.003}_{\pm 0.005}$ | $1.006_{\pm 0.002}$ | $1.026_{\pm 0.000}$ | $1.265_{\pm 0.031}$ | $1.009_{\pm 0.010}$ | $\underline{1.004}_{\pm 0.008}$ |
| nmes | $0.985_{\pm 0.000}$ | $1.028_{\pm 0.020}$ | $1.005_{\pm 0.010}$ | $1.123_{\pm 0.012}$ | $1.048_{\pm 0.006}$ | $1.024_{\pm 0.019}$ | $\mathbf{1.001}_{\pm 0.009}$ | $\underline{1.002}_{\pm 0.011}$ |

*Table 28.* Big Benchmark: DCR metric scores for all datasets. Average and standard devation ($\mu \pm \sigma$) over 20 sampling seeds. Best results per dataset are in **bold**, second best underlined.

| model | SMOTE | ARF | CTGAN | TVAE | TabDDPM | TabSyn | XGenB-AR |
|---|---|---|---|---|---|---|---|
| acsincome | - | $0.980_{\pm 0.006}$ | $\underline{0.998}_{\pm 0.002}$ | $0.994_{\pm 0.006}$ | - | $\mathbf{0.999}_{\pm 0.002}$ | $\underline{0.998}_{\pm 0.002}$ |
| adult | $0.948_{\pm 0.006}$ | $0.914_{\pm 0.007}$ | $\underline{0.994}_{\pm 0.005}$ | $\mathbf{0.996}_{\pm 0.004}$ | $0.970_{\pm 0.007}$ | $0.980_{\pm 0.008}$ | $0.984_{\pm 0.007}$ |
| bank | $0.882_{\pm 0.006}$ | $0.915_{\pm 0.007}$ | $0.982_{\pm 0.007}$ | $0.984_{\pm 0.010}$ | $0.978_{\pm 0.008}$ | $\mathbf{0.993}_{\pm 0.006}$ | $\underline{0.989}_{\pm 0.006}$ |
| beijing | $0.734_{\pm 0.005}$ | $0.940_{\pm 0.008}$ | $\underline{0.981}_{\pm 0.007}$ | $0.978_{\pm 0.011}$ | $\mathbf{0.987}_{\pm 0.007}$ | $0.974_{\pm 0.009}$ | $0.975_{\pm 0.006}$ |
| churn | $0.834_{\pm 0.032}$ | $0.933_{\pm 0.031}$ | $\underline{0.982}_{\pm 0.015}$ | $0.965_{\pm 0.025}$ | $\underline{0.982}_{\pm 0.018}$ | $\mathbf{0.983}_{\pm 0.016}$ | $0.943_{\pm 0.024}$ |
| covertype | - | $0.969_{\pm 0.004}$ | $0.993_{\pm 0.010}$ | $0.993_{\pm 0.005}$ | $\mathbf{0.997}_{\pm 0.003}$ | $\underline{0.995}_{\pm 0.004}$ | $0.994_{\pm 0.005}$ |
| default | $0.751_{\pm 0.009}$ | $0.970_{\pm 0.010}$ | $0.979_{\pm 0.008}$ | $0.974_{\pm 0.009}$ | $0.979_{\pm 0.009}$ | $\mathbf{0.985}_{\pm 0.008}$ | $\underline{0.984}_{\pm 0.007}$ |
| diabetes | $0.807_{\pm 0.004}$ | $0.912_{\pm 0.005}$ | $\mathbf{0.998}_{\pm 0.002}$ | $0.990_{\pm 0.008}$ | - | $0.973_{\pm 0.006}$ | $\underline{0.991}_{\pm 0.005}$ |
| lending | $0.852_{\pm 0.017}$ | $0.898_{\pm 0.016}$ | $0.964_{\pm 0.012}$ | $\mathbf{1.000}_{\pm 0.001}$ | $0.965_{\pm 0.020}$ | $\underline{0.974}_{\pm 0.014}$ | $0.950_{\pm 0.015}$ |
| news | $0.824_{\pm 0.007}$ | $0.966_{\pm 0.007}$ | $\mathbf{0.987}_{\pm 0.007}$ | $0.983_{\pm 0.005}$ | $\underline{0.986}_{\pm 0.007}$ | $0.981_{\pm 0.009}$ | $\underline{0.986}_{\pm 0.006}$ |
| nmes | $0.816_{\pm 0.026}$ | $0.955_{\pm 0.022}$ | $\underline{0.986}_{\pm 0.013}$ | $\mathbf{0.991}_{\pm 0.013}$ | $0.950_{\pm 0.022}$ | $0.953_{\pm 0.024}$ | $0.961_{\pm 0.020}$ |

*Table 29.* Big Benchmark: training time in minutes for all datasets. Best results per dataset are in **bold**, second best underlined.

| model | SMOTE | ARF | CTGAN | TVAE | TabDDPM | TabSyn | XGenB-AR |
|---|---|---|---|---|---|---|---|
| acsincome | - | 89.414 | 16.844 | $\underline{8.263}$ | - | 684.166 | **2.877** |
| adult | - | 4.784 | 6.400 | $\underline{2.722}$ | 25.883 | 70.789 | **0.077** |
| bank | - | 10.545 | 6.309 | $\underline{2.969}$ | 26.113 | 89.043 | **0.057** |
| beijing | - | 5.931 | 5.177 | $\underline{2.917}$ | 22.569 | 73.557 | **0.044** |
| churn | - | 0.341 | 0.416 | $\underline{0.295}$ | 1.874 | 45.802 | **0.022** |
| covertype | - | 143.582 | 26.471 | $\underline{12.510}$ | 40.873 | 950.345 | **0.948** |
| default | - | 8.761 | 5.138 | $\underline{2.721}$ | 17.875 | 68.006 | **0.066** |
| diabetes | - | 57.591 | 30.213 | $\underline{6.804}$ | - | 173.673 | **0.294** |
| lending | - | 7.834 | 5.856 | $\underline{1.795}$ | 16.474 | 55.721 | **0.144** |
| news | - | 30.179 | 13.072 | $\underline{8.499}$ | 24.721 | 126.172 | **0.234** |
| nmes | - | $\underline{0.353}$ | 0.710 | 0.443 | 2.636 | 46.095 | **0.029** |

*Table 30.* Big Benchmark: inference time in seconds for all datasets. Best results per dataset are in **bold**, second best underlined.

| model | SMOTE | ARF | CTGAN | TVAE | TabDDPM | TabSyn | XGenB-AR |
|---|---|---|---|---|---|---|---|
| acsincome | - | $101.964_{\pm 3.579}$ | $\underline{2.003}_{\pm 0.038}$ | $\mathbf{1.454}_{\pm 0.053}$ | - | $37.829_{\pm 0.245}$ | $8.898_{\pm 0.276}$ |
| adult | $12.889_{\pm 0.179}$ | $38.817_{\pm 0.605}$ | $\underline{0.449}_{\pm 0.006}$ | $\mathbf{0.264}_{\pm 0.007}$ | $141.720_{\pm 1.729}$ | $14.286_{\pm 0.091}$ | $2.481_{\pm 0.073}$ |
| bank | $12.391_{\pm 0.339}$ | $39.714_{\pm 0.669}$ | $\underline{0.453}_{\pm 0.005}$ | $\mathbf{0.261}_{\pm 0.005}$ | $136.956_{\pm 0.468}$ | $11.998_{\pm 0.085}$ | $1.645_{\pm 0.048}$ |
| beijing | $6.272_{\pm 0.050}$ | $4.916_{\pm 0.154}$ | $\underline{0.294}_{\pm 0.007}$ | $\mathbf{0.166}_{\pm 0.004}$ | $70.876_{\pm 0.665}$ | $9.149_{\pm 0.012}$ | $1.020_{\pm 0.008}$ |
| churn | $0.203_{\pm 0.003}$ | $1.520_{\pm 0.047}$ | $\underline{0.059}_{\pm 0.001}$ | $\mathbf{0.051}_{\pm 0.001}$ | $10.963_{\pm 0.074}$ | $0.611_{\pm 0.003}$ | $0.218_{\pm 0.004}$ |
| covertype | - | $519.985_{\pm 9.916}$ | $\underline{3.092}_{\pm 0.049}$ | $\mathbf{1.689}_{\pm 0.035}$ | $1295.988_{\pm 182.294}$ | $39.751_{\pm 0.205}$ | $9.042_{\pm 0.350}$ |
| default | $6.011_{\pm 0.178}$ | $27.181_{\pm 0.223}$ | $\underline{0.385}_{\pm 0.008}$ | $\mathbf{0.230}_{\pm 0.005}$ | $91.342_{\pm 1.070}$ | $8.278_{\pm 0.006}$ | $1.751_{\pm 0.042}$ |
| diabetes | $71.852_{\pm 0.931}$ | $231.081_{\pm 4.525}$ | $\underline{3.795}_{\pm 0.066}$ | $\mathbf{3.508}_{\pm 0.534}$ | - | $25.129_{\pm 0.108}$ | $11.303_{\pm 0.233}$ |
| lending | $3.082_{\pm 0.045}$ | $11.232_{\pm 1.543}$ | $\underline{0.647}_{\pm 0.025}$ | $\mathbf{0.555}_{\pm 0.075}$ | $98.118_{\pm 3.302}$ | $2.462_{\pm 0.007}$ | $1.702_{\pm 0.076}$ |
| news | $11.031_{\pm 0.354}$ | $60.550_{\pm 1.296}$ | $\underline{1.044}_{\pm 0.014}$ | $\mathbf{0.591}_{\pm 0.007}$ | $155.496_{\pm 0.811}$ | $8.380_{\pm 0.024}$ | $5.550_{\pm 0.176}$ |
| nmes | $0.380_{\pm 0.013}$ | $3.368_{\pm 0.041}$ | $\mathbf{0.086}_{\pm 0.001}$ | $\mathbf{0.086}_{\pm 0.069}$ | $14.274_{\pm 0.103}$ | $0.856_{\pm 0.004}$ | $\underline{0.334}_{\pm 0.009}$ |

