# OpenReview forum: "XGenBoost: Synthesizing Small and Large Tabular Datasets with XGBoost"
_ICML.cc/2026/Conference — Submitted to ICML 2026_

### Official Review · Reviewer_CKoH · 2026-02-28

**Soundness:** 3
**Presentation:** 3
**Significance:** 2
**Originality:** 2
**Overall Recommendation:** 3
**Confidence:** 2

**Summary:**

The authors present a pair of generative architectures for mixed-type tabular synthesis based on XGBoost. The first one is a De-
noising Diffusion Implicit Model (DDIM) with XGBoost as score-estimator suited for smaller datasets. The second one is a hierarchical autoregressive model whose conditionals are learned via XG-Boost classifiers, suited for large-scale tabular synthesis. They show that, when designed according to the constraints and strengths of tree learners, these architectures are capable of outperforming deep generative models in terms of both quality and computational costs.

**Compliance With Llm Reviewing Policy:**

Affirmed.

**Final Justification:**

Based on authors' rebuttal, I remain confident in my evaluation of the paper.

**Key Questions For Authors:**

- Unless I have missed it, the authors do not appear to report the statistical significance of their results. I would like to see them.

- The impact statement is poorly developed and requires substantial revision. This is more a personal consideration and a refuse from the authors would not compromise the acceptance of the paper.

**Limitations:**

Missing of statistical significance of the results.

**Strengths And Weaknesses:**

Overall, this submission is technically sound with clear, theoretically grounded claims. The mathematical formulation is adequate, and the authors effectively contextualize their work within the existing literature. The writing is clear and the presentation is satisfactory, though the overall significance of the contribution is moderate. There are two primary issues that need to be addressed. Most importantly, unless I have missed it, the authors do not appear to report the statistical significance of their results, which represents a major limitation of the current work. Additionally, unlike the rest of the paper, I feel the impact statement is poorly developed and requires substantial revision.

---

> ### Author Rebuttal · Authors · 2026-03-27
>
> Firstly, many thanks for pointing out that our contribution is technically sound, effectively contextualized, with good presentation.
> We address the mentioned issues point-by-point:
> - Although we do not explicitly mention the statistical significance of the results, the absolute metric scores are given in-depth in Appendix I. From these, the statistical significance can easily be computed.
>
> Reporting mean and standard deviation of absolute metric scores is standard reporting practice in this field. See for example the following impactful works published at similar venues: Xu et al. (2019), Watson et al. (2023), Jolicoeur-Martineau et al. (2024), Zhang et al. (2024), Mueller et al. (2025), Shi et al. (2025). We hope that you agree that we must follow the standard reporting practice, and that the information we provide can already be used to compute significance.
>
> For sake of completeness, we provide a small analysis on the statistical significance from the already provided raw metric scores below.
>
> We perform one-sided Wilcoxon signed-rank test on the null hypothesis that i) XGenB scores “better” than the best baseline if XGenB performs best, or that ii) the best baseline scores “better” than XGenB if the baseline performs best. “Better” is lower for Detection Score, higher otherwise. We ignore datasets for which the best baseline could not be trained (e.g. SMOTE on covertype, acsincome, and ecoli – see the paper for further details). The tables at the end of this rebuttal provide the results. We provide adjusted p-values for Holm-Bonferroni correction to correct for multiple testing within each benchmark.
>
> In the Small Benchmark, XGenB significantly outperforms (on a 5% level) the best-performing baselines on the Shape and Detection metric, indicating **superior fidelity**. There is no significant difference in Trend and MLE scores. TVAE significantly outperforms XGenB on DCR.
>
> In the Large Benchmark,  XGenB significantly outperforms (on a 5% level) the best-performing baseline on the Shape metric. There is no significant difference in Trend, Detection, and MLE. This means that our method achieves **similar or better fidelity** than best-performing diffusion-based baselines, at much **lower training cost** (Table 29 in the paper). There is no significant difference in DCR, indicating that this result is **not** accompanied by an increase in privacy risk.
>
> Hopefully this analysis addresses any remaining concerns on significance of our method.
>
> - We take your comment regarding the Impact Statement very seriously, and we are happy to amend it. We would be interested to hear exactly what you found to be out of order, and will amend it accordingly.
>
> You seem positive about the proposed method and experiments. We are interested to learn whether our response has addressed your remaining concerns.
>
>
> **Small Benchmark (XGenB-DF (v-DDIM)**
> | |Shape|Trend |Detection|MLE|DCR|
> |---|---|---|---|---|---|
> |Best baseline|SMOTE|FF|FF|SMOTE|TVAE|
> |Is XGenB better/worse than baseline? **Bold** indicates significant on 5% level|**Better**|Better|**Better**|Worse|**Worse**|
> |Test results|W=284, p=0.003, p_adj=0.009|W=229, p=0.036, p_adj=0.072|W=56, p=0.001, p_adj=0.005|W=109, p=0.075, p_adj=0.075|W=1, p<0.001, p_adj<0.005|
>
> **Big Benchmark (XGenB-AR)**
> | |Shape|Trend |Detection|MLE|DCR|
> |---|---|---|---|---|---|
> |Best baseline|TabSyn|TabSyn|TabSyn|SMOTE|CTGAN|
> |Is XGenB better/worse than baseline? **Bold** indicates significant on 5% level|**Better**|Better|Better|Worse|Worse|
> |Test results|W=66, p<0.001, p_adj<0.005|W=44, p=0.174, p_adj=0.344|W=13, p=0.042, p_adj=0.164|W=14, p=0.172, p_adj=0.344|W=10, p=0.041, p_adj=0.164|

---

> > ### Author Rebuttal · Reviewer_CKoH · 2026-04-02
> >
> > I would like to thank the authors for the rebuttal.
> >
> > I acknowledge that reporting the mean and standard deviation of absolute metric scores is standard practice in this field, but I insist that this is not a good practice. Statistical tests are needed to scientifically ground claims when models are compared, and, in this case, the results speak for themselves. Speaking about the Impact Statement, I recognize that my initial impression is probably too strong. I would suggest reviewing and reducing the first two paragraphs, which are more of a repetition of the introduction, and aligning the last two paragraphs more with the conclusions. After carefully reading the rebuttal, I remain confident in my initial evaluation of the paper.

---

> > > ### Author Response · Authors · 2026-04-02
> > >
> > > Thanks for addressing our rebuttal and for providing further details to improve the Impact Statement. We provide an updated version at the bottom of this message; we combined the first two paragraphs and aligned the last two with the conclusions from the paper, as requested.
> > >
> > > **Seeing as you acknowledge that we've addressed all your concerns adequately, we hope you consider updating your score. When choosing the option "Fully Resolved", this is explicitly encouraged.**
> > >
> > > As all concerns have been addressed, we have to make some assumptions as to why your evaluation of this work has not improved. We believe it may be due to your mention that *"the overall significance of the contribution is moderate."*, in your initial review. We would therefore like to further address the significance of this work. Notably, our work **addresses more issues than similar work published at similar venues**, and can therefore be considered similarly or more significant to this field:
> > > - Other works on tree-based tabular synthesis only consider small datasets (Jolicoeur-Martineau et al. (2024); McCarter (2024)). We account for both small- and large-scale synthesis and design a new architecture to address this.
> > > - XGenB-DF provides a tunable fidelity-privacy tradeoff through a new dropout procedure for tree-based models (see Appendix B for the ablation). Previous methods do not provide such a tunable parameter.
> > > - XGenB-DF **significantly** improves on the Detection score compared to the best baseline. The Detection score is, by far, the most important fidelity metric, as it is the only one which compresses the similarity of the full multivariate distribution into a single score (see Appendix G.2). This shows that XGenB-DF **significantly** improves on the fidelity frontier; an important feat in this field.
> > > - XGenB-AR hugely improves on the training cost in large-scale tabular synthesis (Table 29) while performing similar (not significantly different), fidelity-wise, to the best baseline. This shows that we improve on the quality-cost frontier. This is of great significance to this field, as it allows researchers with fewer resources access to strong models.
> > > - XGenB-AR explicitly accounts for complex numerical distribution shapes (zero-inflated or highly skewed features, etc.) through EQF-based resampling, and for high-cardinality categorical features through a clustering approach. Note that for this reason, XGenB-AR **significantly** outperforms the best baseline on the Shape metric, indicating just how effective our new approaches are. Most other works do not explicitly account for such complex feature shapes - which is likely why we outperform them on this metric.
> > >
> > > We recognize that not everyone is interested in the field of tabular synthesis. However, we believe that significance should be evaluated with respect to the field it considers. Considering the above, we have to reassure you that this work addresses some of the most important problems in this field, e.g., fidelity frontier, fidelity-privacy tradeoff, training cost frontier, complex numerical and categorical feature shapes.
> > >
> > > We hope you reconsider your assessment of the originality and significance of this work. If not, we would love to learn what *would* warrant further increasing the score. Currently, it seems we've addressed all remaining concerns and you're generally positive about the work, but the overall recommendation is still below acceptance threshold; this is difficult to reconcile.
> > >
> > > **Updated Impact Statement**
> > >
> > > *This paper presents methods which generate synthetic observations similar to those found in real-world datasets. Results from synthetic data should always be compared to results from real-world data, as there are no guarantees that inferences drawn from synthetic data generalize to the real-world. Furthermore, synthetic data can exhibit privacy risk as information from the real data may leak through to the synthetic data. When using synthetic data for privacy-preserving purposes, always include a rigorous evaluation of privacy risks, containing, e.g., attribute and membership disclosure analyses.*
> > >
> > > *The results from XGenB-DF show that designing architectures according to appropriate inductive biases for the underlying dataset can greatly improve fidelity. Most notably, using categorical instead of numerical diffusion processes for categorical data. Furthermore, it shows how applying improvements from other domains (deep learning) to new learning settings (tree-based learners) can provide considerable wins, e.g., velocity-based diffusion, and DDIM vs. DDPM sampling.*
> > >
> > > *Lastly, we show that XGenB-AR can generate high-fidelity synthetic data at much lower training cost than previous methods. This democratizes access to high-fidelity tabular synthesis to researchers with modest compute resources. Additionally, reducing computational requirements reduces energy demand, which can be beneficial from both a financial and sustainability perspective.*

---

### Official Review · Reviewer_CuKG · 2026-03-06

**Soundness:** 3
**Presentation:** 2
**Significance:** 2
**Originality:** 3
**Overall Recommendation:** 4
**Confidence:** 2

**Summary:**

The paper proposes XGenBoost, a pair of tabular data generative models that replace neural networks with gradient-boosted decision trees (XGBoost) as function approximators. For small datasets, XGenB-DF is a mixed-type diffusion model that combines Gaussian diffusion for numericals and multinomial diffusion for categoricals, using per-feature XGBoost estimators and a DDIM sampler for faster generation. For large datasets, XGenB-AR is a fixed-order autoregressive model that learns conditionals via XGBoost classifiers, uses hierarchical classification for quantized numericals, and de-quantization via per-bin empirical quantile functions; it also introduces a clustering-based strategy to cap categorical cardinality. Across two benchmarks (27 small and 11 large datasets), the authors report improved fidelity and utility at lower training cost than neural and prior tree-based baselines.

**Compliance With Llm Reviewing Policy:**

Affirmed.

**Final Justification:**

The response addresses my earlier concern about scalability, and the discussion of autoregressive sensitivity is helpful, although I still think the paper would benefit from more systematic analysis of feature ordering and AR robustness; accordingly, I am raising my score slightly while keeping some reservations.

**Key Questions For Authors:**

For XGenB-DF, what are the typical values of T and K, and how many models are trained in total per dataset (features × timesteps × labels)? Please provide wall-clock time and memory for training, and how these scale with number of features and categorical cardinality.

What exact loss/objective is used for the multinomial diffusion classifiers (e.g., cross-entropy on x0 logits vs. a VLB-style objective)? How are timestep conditionings fed to XGBoost models?

In XGenB-AR, how are B_j (number of bins) and hierarchical depth H selected per feature? How sensitive are fidelity and MLE to these hyperparameters, especially with small per-bin sample sizes?

Could you report absolute metric scores (not just ranks) and include confidence intervals/significance tests across seeds for key datasets to quantify effect sizes?

**Strengths And Weaknesses:**

**Pros:**

- The mixed-type diffusion design that leverages multinomial diffusion for categoricals avoids one-hot encoding and aligns well with tree split behavior.

- The autoregressive modeling with hierarchical XGBoost classifiers for numerical conditionals and per-bin empirical quantile de-quantization is a thoughtful engineering of inductive biases to tabular data characteristics.

- The clustering-based scheme to cap high categorical cardinality is a practical contribution that balances scalability, fidelity, and reduced privacy risk for rare categories.

- The paper articulates constraints inherent to tree learners (single-output models, no minibatching) and designs architectures around them (per-feature models, DDIM for fewer steps, fixed-order AR).


**Cons:**

- The diffusion variant requires training separate per-feature, per-timestep models (and per-label for categoricals), plus data “extension” over K noise levels; even with DDIM, this can be heavy and the exact scale of K, T, and number of models is not fully quantified.

- The AR model’s de-quantization via empirical per-bin quantile functions may overfit when bins are small and can leak memorized local structure; more robust smoothing or DP-aware de-quantization is not explored.

- Fixed feature ordering simplifies training but can degrade modeling of long-range dependencies; attempts with MI-based orders are reported as inconclusive, but no systematic analysis is provided.

---

> ### Author Rebuttal · Authors · 2026-03-27
>
> Thank you for your comments. Below we address specific questions. Sometimes we will shorten the excerpts of your comments with three dots (…) for brevity.
>
> > the exact scale of K, T, and number of models is not fully quantified
>
> > what are the typical values of T and K, and how many models are trained in total per dataset (features × timesteps × labels)
>
> The scales of K and T are given in Appendix F: T=50 and K=100.
> The number of models can be directly inferred from Section 3.1, but also from your own statement – which means this information **is contained in the paper**: we train feature-, timestep-, and label-wise models: T $\cdot$ D $\cdot$ C models.
>
> > The diffusion variant requires training … this can be heavy
>
> We thoroughly address this issue in the paper. We design an entirely **new model architecture** (XGenB-AR) to solve these scaling issues. We then show that XGenB-AR effectively and efficiently scales to large datasets.
>
> > The AR model’s de-quantization … DP-aware de-quantization is not explored
>
> Small bins are **not applicable**: we use a rank-based binning which ensures equal-size bins. This only yields small bins if the dataset itself is very small, but in that case, we suggest XGenB-DF as more appropriate anyway. Additionally, we **do** investigate more robust smoothing in Appendix E: we show that EQF smoothing is not more prone to overfitting (DCR score) than uniform smoothing.
>
> We don't consider DP. We aim to learn the true generative process (which DP distorts) and provide a fair comparison to other baselines; DP-aware de-quantization is therefore out-of-scope.
>
> > Fixed feature ordering simplifies … no systematic analysis is provided
>
> Fixed ordering degrades modelling of long-range dependencies in **deep neural nets** especially (e.g., vanishing-gradients). As we use feature-wise XGBoost models and not neural networks, this does **not** apply to our method. Error accumulation during sampling may be a remaining issue, but seems a weaker argument given the relatively low number of features in tabular data.
>
> The feature-order analysis did not meaningfully improve results (null-result). We provide many more interesting ablations in this paper. We would rather keep this as an open issue to be explored in the future. We hope you understand.
>
> > Please provide wall-clock time and memory for training, and how these scale with number of features and categorical cardinality
>
> Wall-clock time is given graphically in Figure 1, and in absolute numbers in Table 17 (training) and Table 18 (inference). Appendix H indicates that we used 64GB RAM. Scaling can be computed from the provided wall-clock times and the dataset characteristics given in Table 3.
>
> > What exact loss/objective is used for the multinomial diffusion classifiers
>
> Cross-entropy on x0 logits.
>
> > How are timestep conditionings fed to XGBoost models
>
> See **Section 3.1**: we train separate models per-timestep to force conditioning on timestep information in XGBoost.
>
> > In XGenB-AR, how are B_j (number of bins) and hierarchical depth H selected per feature? How sensitive are fidelity and MLE to these hyperparameters, especially with small per-bin sample sizes?
>
> See **Section 3.2**: $B_j$ directly follows from H ($B_j=2^H$). These are fixed at H=5 as indicated in Appendix F. Regarding the effect of increasing/decreasing H we can add the following clarification in the text: *“Increasing H increases bin granularity and therefore improves multivariate fidelity, and any measures which depend on that, such as MLE. It does, however, negatively influence runtime, as we train $2^H$ binary XGB classifiers per feature.”*
>
> > report absolute metric scores (not just ranks) and include confidence intervals/significance tests
>
> Absolute metric scores were **already included in Appendix I**. We referred to them in Section 4.1.
>
> For an analysis on significance tests, please see our response to **reviewer CKoH**.
>
> ---
> Below we relate our rebuttal to some of the assessment dimensions:
>
> **Presentation**: many of the "Cons" and "Key Questions" regarded details which were, in fact, already adequately present in the paper. We explicitly referred to them in the main text. We therefore feel that an update in the score for "presentation" is warranted.
>
> **Significance**: the statistical significance analysis (see response to **reviewer CKoH**) shows that **i)** XGenB-DF significantly improves fidelity standards, and **ii)** XGenB-AR improves the quality-cost frontier. Furthermore, we address a broad range of use cases in this work: both small and large datasets, modelling mixed-type and high-cardinality features, tunable fidelity/privacy tradeoff, efficient training under modest compute, and easily accessible code. We therefore believe this work has all the ingredients to be highly impactful in the field of tabular synthesis, and a "significance" score of 2 seems low; especially given our newly provided significance analysis.
>
> Thanks again for your thorough review.

---

> > ### Author Rebuttal · Reviewer_CuKG · 2026-04-02
> >
> > The rebuttal clarifies several implementation details and partially addresses the de-quantization concern, but it does not materially change my overall assessment. The scalability cost of training many separate models still seems substantial; the feature-ordering issue is not systematically analyzed, and the AR sensitivity discussion remains limited.

---

> > > ### Author Response · Authors · 2026-04-02
> > >
> > > Thank you for your response.
> > >
> > > We would like to re-emphasize that we have provided many clarifications which should expectedly have an impact on the score for "presentation". Many details on which you were previously unsure were already readily available in the paper, as our rebuttal pointed out. We are interested to hear why this has not impacted your evaluation of the "presentation" score, and which concerns still prevent you from updating said score. If you do feel that we have addressed some of these, we humbly request that you consider updating this score.
> > >
> > > > The scalability cost of training many separate models still seems substantial
> > >
> > > The rebuttal clarified that **this is not relevant** in our work. XGenB-DF, which is the only architecture which trains many separate models, is **not used to scale to larger datasets**; scalability is therefore irrelevant. Our paper designs and implements XGenB-AR instead of XGenB-DF to scale to larger datasets. XGenB-AR **does not train many models**, and scales very well to larger datasets, as shown in Figure 1. We feel that this concern is therefore thoroughly addressed; we are interested to hear what exactly your remaining concern is regarding scalability, given this response. Or otherwise, if this concern is addressed, to reflect this in your score.
> > >
> > > > the AR sensitivity discussion remains limited
> > >
> > > We assume that with "AR sensitivity", you mean the potential degradation of long-range dependencies. We addressed this thoroughly, and from your response it is unclear what your remaining concern is. Do you not believe that this is mainly an issue in backprop-based models (vanishing gradients)? Or, do you not believe that error accumulation during AR-sampling is not a main problem for tabular data with a limited number of features? We hope you will re-evaluate our discussion on this from the rebuttal, as we feel it properly explains why this is not a significant issue.
> > >
> > >
> > > Thanks for reviewing our work. We hope you will take the time to re-evaluate our discussion, as we feel that we've addressed many previous concerns. Finally, we are interested to hear what substantial concerns remain which prevent acceptance of this work.

---

### Official Review · Reviewer_4WDR · 2026-03-11

**Soundness:** 2
**Presentation:** 1
**Significance:** 2
**Originality:** 2
**Overall Recommendation:** 2
**Confidence:** 4

**Summary:**

The paper proposes XGEnboost, which uses XGBoost as backbone of tabular synthesis method to introduce more tabular friendly inductive biases and reduces compute cost of table synthesis. The paper includes two models: XGenB-df is a diffusion model where the xgboost is used for score function estimator. It used separate Gaussian diffusion and multinomial diffusion processes to model numerical vs category variables. The XGenB-AR is an autoregressive predictor using a hierarchy of XGBoost model to fit the features sequentially via teacher forcing, while generate features autogressively during test time. The XGenB-DF is designed for smaller datasets while XGenB-AR for larger datasets.

The paper conducted benchmark on two datasets: a “Small Benchmark” and a “Big Benchmark” which the authors presented as containing smaller tables and larger tables. Both outperforming strong deep learning baselines such as TabDDPM/CTGAn with much lower training cost. The DF and AR version outperformed baselines on small and big dataset respectively. The appendix also contains clear ablations on dropout, hierarchical classification, category merging, and dequantization strategy illustrating contribution of each modules.

**Compliance With Llm Reviewing Policy:**

Affirmed.

**Final Justification:**

During the discussion phase, the authors responded to my follow-up questions:

(1) Presentation clarity: The authors acknowledged the existing clarity issues and committed to revising the introduction in future versions. This is appreciated.

(2) Baseline inclusion: The response does not address the core concern. Robust benchmarking requires including both impactful and strong baselines; these are not substitutes for each other. The authors' justification for omitting TabDiff: that they preferred comparing against more widely used methods, reflects an editorial preference rather than a technical argument. TabDiff is a directly comparable mixed-type diffusion model that has been published at a top-tier venue for roughly a year and is itself widely cited. Its omission leaves the central empirical claims inadequately supported.

I therefore maintain my rating.

**Key Questions For Authors:**

+ Is there a more principled decision boundary between XGenB-DF and XGenB-AR other than scaling??

+ Is the use of π(0) in Section 3.2 a typo, or is some initialization procedure missing from the presentation?

+ Can the authors provide stronger reasong for excluding TabSyn from the small benchmark?

+ Any theoretical or empirical motivation that diffusion model is stronger for small data while AR is stronger for bigger ones?

**Limitations:**

yes

**Strengths And Weaknesses:**

## Strength

+ The concept is interesting and meaningful. Using tree-based model such as XGboost has strong potential in enhancing table synthesis performance with knownly good inductive biases for tabular features while speeding up computation,  making industrial scale deployment more realistic.

+ The benchmark and ablation study are strong. Different important aspects of tabular synthesis such as fidelity/utlity/privacy are tested, and a wide range of methods are covered.

+ The idea of replacing NN back bone in diffusion model with tree-based estimator is interesting, and appear genuinely novel and worthy of in-depth discussion.


## Weakness

+ The presentation and framing is confusing, the introduction argues deep models are expensive, yet XGenB-DF is still a diffusion model. With the introduction paragraph jumping between points, it is hard to grasp the key method of replacing NN backbone.

+ Notation appear inconsistent; For example, the autoregressive factorization is defined over π(1),…,π(d) but later references π(0).

+ The small/large dataset split is not fully justified; no principled criterion (row count, feature count, cardinality) is given for choosing between XGenB-DF and XGenB-AR. Some of the "small dataset" actually has more row counts than some of the "big" ones.

+ Baseline comparisons are not fully convincing: TabSyn is excluded from the small benchmark by author judgment that they are not suitable for small data, and several baseline hyperparameters are borrowed or runtime-capped rather than uniformly tuned, entangling modeling quality with engineering constraints. More recent SoTA models such as TabDiff are also not included. Even if the author believes they will not be powerful baselines, excluding these strong baselines will will severely impact the strength of this benchmarking.

+  The claim that XGenB-DF is more suitable for small dataset is not well justified, as diffusion models are known to excel on generation of big and diverse datset in domains such as image diffusion.

---

> ### Author Rebuttal · Authors · 2026-03-27
>
> Thanks for your comments. Below we address specific questions. Sometimes we shorten the excerpts of your comments with three dots (…) for brevity.
>
> > The presentation and framing is confusing, … the key method of replacing NN backbone.
>
> > The claim that XGenB-DF is more suitable … such as image diffusion.
>
> > Any theoretical or empirical motivation … for bigger ones?
>
> We would like to point out that the paper doesn't make such arguments. The introduction does **not** use **training efficiency** as the main argument for using an XGB diffusion backbone. We only consider XGenB-DF for small datasets. Efficiency gains are much less important here, as absolute runtimes are still low. Our main argument focuses on XGB’s **inductive biases and overall strong performance** on mixed-type tabular data.
>
> Training efficiency is only considered for XGenB-AR, which we use for larger datasets, where efficiency gains are much more important.
>
> Our claims regarding XGenB-DF suitability for smaller datasets are **purely due to scaling issues**. Nowhere do we claim that diffusion models are better **quality-wise** on small datasets. First, we mention that XGB-based diffusion models require extending the training set size (Design Constraints). Then, we mention that this is permissible for small datasets (Synthesizing Small Datasets), but **not** permissible for large datasets due to scaling issues (Synthesizing Large Datasets).
>
> > Notation appear inconsistent; … but later references π(0).
>
> > Is the use of π(0) in Section 3.2 a typo … missing from the presentation?
>
> Thanks for pointing this out, this is a **typo**.
>
> > TabSyn is excluded … not suitable for small data
>
> > Can the authors provide stronger reasong for excluding TabSyn from the small benchmark?
>
> TabSyn is extremely heavy compared to other baselines under its default/public set-up: TabSyn has 10M parameters vs. TabDDPM <1M in the Small Benchmark. There is no light-weight default readily available. We use FD, FF, and UT instead of TabSyn as these are SoTA **on this specific benchmark**, and thus seem more useful.
>
> We can add a more principled justification to the text:
>
> *“TabSyn’s default set-up requires training two highly parameterized deep generative models (VAE and diffusion model), without a more lightweight set-up available from the original or other work. Less costly baselines exist which have already shown to be state-of-the-art on exactly this benchmark of small datasets, namely FD, FF, and UT, and therefore include these instead of TabSyn.”*
>
> > several baseline hyperparameters … modeling quality with engineering constraints.
>
> We follow standard practice by keeping baseline implementations faithful to the original instead of tuning parameters. See the following work published at similar venues: Xu et al. (2019), Watson et al. (2023), Jolicoeur-Martineau et al. (2024), Zhang et al. (2024), Mueller et al. (2025), Shi et al. (2025). The field expects a level of robustness to the hyperparameters, and tuning them is very expensive for generative models.
>
>
> > More recent SoTA models such as TabDiff are also not included.
>
> We covered the major model families with strong representatives rather than being exhaustive within every family. We include two of the most impactful diffusion baselines, TabDDPM and TabSyn. That said, omitting TabDiff is a limitation and we can make this explicit in the text. We’re interested to hear whether including TabDiff is a necessity, considering we include 9 strong baselines from a variety of modelling families, which is more than similar works.
>
> > The small/large dataset split is not fully justified; … row counts than some of the "big" ones.
>
> The split criterion of small/large datasets depends greatly on computational resources, so a general principle is not directly possible. With many resources, you can run the models on large datasets and have reasonable runtime. In practice, users should experiment whether XGenB-DF can be reasonably run for their scenario. This is the standard practice in this field; we are not aware of any work which provides a principled criterion for which dataset sizes their model can be run – this **always** depends on resources.
>
> The overlap in dataset size between the benchmarks are due to using existing benchmarks from literature. We do this to avoid cherrypicking datasets, which we see as a big issue.
>
>
> > Is there a more principled decision boundary … other than scaling??
>
> XGenB-DF more accurately models multivariate dependencies than XGenB-AR, as the latter loses granularity due to binning. However, XGenB-AR replicates marginal distributions very accurately. If marginals are your main concern, you can use XGenB-AR at much lower training cost. For more accurate multivariate dependencies, use XGenB-DF. We also mention this in **Section 4.1.**
>
> Thanks for your thorough review. We are interested to hear whether we’ve addressed some of your concerns, and whether you’re willing to reflect this in your score.

---

> > ### Author Rebuttal · Reviewer_4WDR · 2026-04-03
> >
> > The authors have partially addressed my concerns. I appreciate the clarification on the notation typo, and the nuance on when to prefer XGenB-DF versus XGenB-AR. However, two major issues remain insufficiently resolved.
> >
> > First, regarding presentation: while the clarification on core motivation is appreciated, the writing requires substantial improvement.  The authors state in their rebuttal that "the introduction does not use training efficiency as the main argument for using an XGB diffusion backbone," yet the introduction's second sentence reads "most current state-of-the-art methods rely on deep neural networks, which require modern computing resources (GPUs)," immediately followed by the claim that "tree ensembles are often considered to be more suitable function approximators." The paper then proceeds to propose XGenB-DF, itself a diffusion model. When the opening paragraph interleaves compute-cost with inductive-bias motivation without clearly scoping which argument applies to which proposed model, it is hard to disentangle the logic by readers themselves.
> >
> > Second, regarding baselines: the argument that nine baselines from diverse families compensates for omitting TabDiff is not convincing. Baseline breadth does not substitute for including the strongest known competitor within the paper's own model family. TabDiff is a recent, high-performing mixed-type diffusion model directly comparable to XGenB-DF, and its absence weakens the central empirical claims.
> >
> > I encourage the authors to prioritize a clearer introduction structure and a more complete baseline comparison in any revision.

---

> > > ### Author Response · Authors · 2026-04-03
> > >
> > > Thanks for your acknowledgement.
> > >
> > > Firstly, we can understand that some parts of the introduction may be confusing in its current state. We should have more clearly stated the differences in justification for using XGB for small and large datasets. We are glad that our response has clarified it somewhat. We will revise it according to our rebuttal; hopefully this is satisfactory.
> > >
> > > Secondly, we can understand your viewpoint regarding the baseline comparisons. This is, from our perspective, always a bit of a contentious point in these types of ML studies: do we choose the most *impactful* or the *strongest / most recent* baselines. Naturally, TabDDPM and TabSyn have been much more impactful in this field. However, TabDiff is indeed more recent, and likely stronger on most datasets, as it make various improvements. We chose the more impactful methods as baseline approaches, as these are more widely used, and it was more interesting to us to see how our approach fairs.
> > >
> > >
> > > Thanks again for your thorough review. If you feel that we have substantially clarified some of your concerns, we humbly request that you reconsider your scores.

---

### Decision · Program_Chairs · 2026-04-30

**Decision:**

Reject

**Comment:**

The paper introduces XGEnboost, a framework that utilizes XGBoost to build both diffusion (XGenB-df) and autoregressive (XGenB-AR) models for tabular data synthesis. The reviewers acknowledge the method’s potential in reducing computational costs and introducing tabular-friendly inductive biases, as well as the thoroughness of the ablations provided in the appendix.

However, the meta-review recommends rejection due to unresolved concerns regarding presentation and empirical validation. While the authors committed to improving the clarity of the introduction, they did not adequately address the omission of critical baselines. Specifically, the reviewers noted that excluding directly comparable state-of-the-art models like TabDiff—a widely cited diffusion model for mixed-type data—undermines the paper's central claims of superiority. The justification for omitting such strong baselines based on "editorial preference" was deemed a technical shortcoming rather than a valid research choice. Consequently, the benchmarking is considered insufficient to prove the method's competitiveness in the current research landscape.